# Understanding wind-driven melt of patchy snow cover

Luuk D. van der Valk[1,2], Adriaan J. Teuling[2], Luc Girod[3], Norbert Pirk[3], Robin Stoffer[1], and Chiel C. van Heerwaarden[1]

[1]Meteorology and Air Quality Group, Wageningen University & Research, Wageningen, the Netherlands
[2]Hydrology and Quantitative Water Management Group, Wageningen University & Research, Wageningen, the Netherlands
[3]Department of Geosciences, University of Oslo, Oslo, Norway

**Correspondence:** Luuk van der Valk, now at Department of Water Management, Delft University of Technology
(l.d.vandervalk@tudelft.nl)

**Abstract.** The simplified representation of snow processes in most large-scale hydrological and climate models is known to introduce considerable uncertainty in the predictions and projections of water availability. During the critical snowmelt period, the main challenge in snow modeling is that net radiation is spatially highly variable for a patchy snow cover, resulting in large horizontal differences in temperatures and heat fluxes. When a wind blows over such a system, these differences can drive advection of sensible and latent heat from the snow free areas to the snow patches, potentially enhancing the melt rates at the leading edge and increasing the variability of subgrid melt rates. To get more insight in these processes, we examine the melt along the upwind and downwind edges of a 50 meter long snow patch in the Finseelvi catchment, Norway, and try to explain the observed behaviour with idealized simulations of heat fluxes and air movement over patchy snow. The melt of the snow patch was monitored from 11 June until 15 June 2019 by making use of height maps obtained through the photogrammetric Structure-from-Motion principle. A vertical melt of $23 \pm 2.0$ cm was observed at the upwind edge over the course of the field campaign, whereas the downwind edge melted only $3 \pm 0.4$ cm. When comparing this with meteorological measurements, we estimate the turbulent heat fluxes to be responsible for 60 to 80% of the upwind melt of which a significant part is caused by the latent heat flux. The melt at the downwind edge approximately matches the melt occurring due to net radiation. To better understand the dominant processes, we represented this behaviour in idealized direct numerical simulations, which are based on the measurements on a single snow patch by Harder et al. (2017) and resemble a flat patchy snow cover with typical snow patch sizes of 15, 30 and 60 m. Using these simulations, we found that the reduction of the vertical temperature gradient over the snow patch was the main cause for the reductions in vertical sensible heat flux over distance from the leading edge, independent of typical snow patch size. Moreover, we observed that the sensible heat fluxes at the leading edge and the decay over distance were independent of snow patch size as well, which resulted in a 15% and 25% reduction in average snowmelt for respectively a doubling and quadrupling of typical snow patch size. These findings lay out pathways to include the effect of highly variable turbulent heat fluxes based on the typical snow patch size in large-scale hydrological and climate models to improve snowmelt modelling.

# 1 Introduction

Snow plays a crucial role for much of the world's population, through the important water and climate services that it provides.
Snowmelt is crucial for more than one sixth of the world population for agricultural purposes or human consumption (Barnett et al., 2005). Also, snow is a natural way to store water during the cold months, which is released during spring and summer when the water demands are higher (e.g. Viviroli et al., 2007; Berghuijs et al., 2014). Changes in this snow cover, due to shifts in precipitation and temperature, have amongst others an impact on society (Golombek et al., 2012; Grünewald et al., 2018; Sturm et al., 2017) and ecosystems (Groffman et al., 2001; Wheeler et al., 2016). Strong changes in snow cover have been observed over the past decades in many regions, including Europe (Fontrodona Bach et al., 2018) and the United States (Mote et al., 2018). To be able to assess the effects of changes in the snow cover, a more thorough understanding of the influence of snow processes on discharge is needed (Berghuijs et al., 2014) and uncertainties in hydrological models due to snow storage and melt need to be overcome (Melsen et al., 2018). Additional uncertainties are introduced by the transformation of a continuous snow cover into a patchy snow cover, as the snow patches and associated processes are challenging to be correctly captured by even relatively advanced hydrological and climate models (e.g. Lejeune et al., 2007; Liston, 2004; Loth and Graf, 1998; Mott et al., 2011, 2015). Uncertainty related to the modeling of snow processes might even lead to disagreement in the sign of projected mean streamflow changes (Melsen et al., 2018). To reduce the modelling uncertainty of patchy snow covers, a deeper understanding of the snowmelt processes under spatial variability is needed.

The main processes causing snow to melt are different for a patchy snow cover than a continuous snow cover. For a continuous snow cover, the surface energy balance, being responsible for snowmelt, is dominated by radiation (e.g. Male and Granger, 1981). For this type of cover, the turbulent heat fluxes are mainly driven by large-scale air mass movement affecting the ambient air temperature or moisture and wind speed, which could cause these fluxes to be significant during brief periods (Anderson et al., 2010). However, over longer periods, such as weeks, air temperature and moisture gradients near the surface are generally too low to generate significant turbulent heat fluxes (Hock, 2005). When the snow cover turns patchy, the net radiation becomes spatially highly variable, due to variations in surface albedo and emissivity. This spatial variability can act on orders of meters, such that a highly heterogeneous surface arises of relatively warm (and possibly wet) snow free area adjacent to a relatively cold (and drier) snow patch. When a wind blows over this horizontally heterogeneous surface, internal boundary layers form downwind of the transitions, due to changes in the surface conditions (e.g. Garratt, 1990). The heterogeneity of these internal boundary layers induces a system in which the turbulent heat fluxes can highly vary spatially, partly due to the advection of sensible and latent heat (Essery et al., 2006; Mott et al., 2013; Harder et al., 2017). These systems are often described as separate growing boundary layers following a power law as function of the fetch (e.g. Granger et al., 2002, 2006). For the stable internal boundary layers, i.e. over snow patches, the air close to the snow surface can decouple from the warmer air above, either due to large temperature differences between both or through cold-air pooling, eventually limiting the exchange of sensible and latent heat from the atmosphere towards the snow (Fujita et al., 2010; Mott et al., 2016). Moreover, the influence of the turbulent heat fluxes on the total amount of melt increases with decreasing snow cover fractions (Harder et al., 2019; Marsh et al., 1999; Schlögl et al., 2018a). It has been suggested that this process can be responsible for up to 50%

of the total snowmelt (e.g. Harder et al., 2017; Mott et al., 2011). Additionally, similar processes have been found to potentially significantly contribute to snowmelt on ice fields (e.g. Mott et al., 2019) and glaciers (Sauter and Galos, 2016; Bonekamp et al., 2020; Mott et al., 2020). This stresses the potentially significant contribution of lateral transport of sensible and latent heat to

a melting patchy snow cover, though it opens the question what the hydrological relevance on larger scales is.

In spite of its potential importance, the lateral transport of sensible and latent heat is a rather underrepresented process in hydrological or even more complex atmospheric snow surface models, as these focus mainly on vertical melt processes. Traditionally, simple models of snowmelt in hydrological models use the so-called temperature-index (TI), which performs generally well with few computational costs. However, the performance of these models decreases significantly with increasing

temporal resolution, adding a spatial component to the model, application beyond the period or domain of calibration (e.g. climate change) or during rain-on-snow events (e.g. Hock, 2003). The spatial component is especially important in mountainous regions due to topographical effects like cold-air pooling and shading, which are not taken into account with the temperature-index. Furthermore, the exclusion of wind speed in temperature-index models can cause a bias for the amount of modelled snowmelt, especially when turbulence-driven snowmelt becomes increasingly important, for example for patchy snow covers

(e.g Kumar et al., 2013). Even when using complex atmospheric snow surface models, such as Alpine3D (Lehning et al., 2006) with ARPS (Xue et al., 2001), local-scale advection associated with subgrid variation of snow and bare ground is excluded. Few models, like Alpine3D and CHM (Vionnet et al., 2021), do include wind-driven processes like snow redistribution and turbulent heat fluxes, but most often parameterize the subgrid turbulent fluxes with the average temperature or moisture at the surface and lowest atmospheric layer per grid cell and do not account for subgrid spatially varying melt fluxes. As potential solutions,

Harder et al. (2019) propose a simple model to include local-scale advection of the sensible and latent heat to snowmelt using scaling laws, while Essery et al. (2006) develops a more complex approach based on integration of the energy equation as suggested by Granger et al. (2002) using mixing length theory (Weismann, 1977). Considering the subgrid heterogeneity of the melting fluxes would significantly improve snow cover and discharge predictions (DeBeer and Pomeroy, 2017; Mott et al., 2017; Pohl and Marsh, 2006). Still, implementing the subgrid turbulent fluxes using a parametrization is difficult, mainly

due to the small spatial scales of the local-scale advection and the interplay between snow cover area and melt rates (Harder et al., 2019), as well as the process is less well understood on a catchment scale (Mott et al., 2018; Pohl and Marsh, 2006). As a consequence, more field observations should be performed to study its importance in various environmental settings. Additionally, these should be combined with other modelling approaches that can serve as a tool to improve our understanding of the process on small and larger scales. Such progress will eventually enable the implementation of lateral transport of the

sensible and latent heat.

Dedicated observations are needed to quantify the importance of the sensible and latent heat advection on snowmelt. Several field measurements focused on the meteorological surface characteristics, such as the development of an internal boundary layer (IBL) as a consequence of the heterogeneous snow cover (Mott et al., 2017) or the influence of topographical depression on cold-air pooling and subsequent snowmelt (Fujita et al., 2010). Other field measurement campaigns estimated the turbulent

heat fluxes and the accompanied mechanisms for a single isolated snow patch (e.g. Granger et al., 2006; Harder et al., 2017). Though, all of these experiments focus on relatively brief periods of time with a maximum timespan of a day, during which

the conditions for local-scale advection of sensible and latent heat were often ideal. For small areas in complex terrain and longer periods, Schlögl et al. (2018b) related measured snowmelt to turbulence, while Marsh et al. (1997) estimated the role of advection of the sensible heat by comparing estimated sensible heat fluxes with and without advection. Yet, estimates of a multiple-day contribution of the turbulent heat fluxes to the melt of a single snow patch are lacking, but could provide additional insights in the role of these fluxes on longer timescales.

To observe the melt of a snow patch over the course of multiple days, high spatial variability of melt rates complicates the observations. Single point measurements might not represent the region of interest, especially for seasonal snow covers (Sturm and Benson, 2004). This advocates the use of spatial field observations at a high temporal and spatial resolution for patchy snow covers. Ground-based methods fulfilling these requirements are amongst others terrestrial laser scanning (TLS) (Egli et al., 2012; Grünewald et al., 2010; Hojatimalekshah et al., 2021; Schlögl et al., 2018b) or georectification of oblique time-lapse photography (Härer et al., 2013). Additionally, aerial platforms, such as aerial laser scanning (ALS), either manned (e.g. Deems et al., 2013; Painter et al., 2016) or unmanned (e.g. Harder et al., 2020; Jacobs et al., 2021) can be used. From both positions, application of Structure-from-Motion (SfM) photogrammetry is possible as a relatively cheap method to monitor snowmelt, of which the usage has already been explored in the 1960s (e.g Brandenberger, 1959; Hamilton, 1965), but which was sidelined due to technical constraints. Recently, due to the technical development increasing the accuracy with lower computational costs, the SfM photogrammetry has been used to successfully study seasonal snow covers with low-cost imagery equipment and software for analysis (e.g. Bühler et al., 2016; Filhol et al., 2019; Girod et al., 2017; Nolan et al., 2015). Therefore, this method offers a promising way to monitor snowmelt with low costs and reasonable accuracy.

To eliminate parametrization uncertainties, a relatively new type of simulations, Direct Numerical Simulations (DNS), can be applied to model local-scale advection, as it resolves all the relevant spatial and temporal scales of turbulence. This simulation type has already proven its value in the field of fluid dynamics (Moin and Mahesh, 1998) and allows to extract very detailed information from the turbulent flow, enhancing our process-based understanding of local-scale advection. As a consequence of the high resolutions, these simulations do not need any turbulence parametrizations based on stability corrections and the Monin-Obukhov assumptions, in contrast to numerical atmospheric boundary layer models (Liston, 1995; Marsh et al., 1999) or Large-Eddy simulations (Mott et al., 2015; Sauter and Galos, 2016). These methods assume horizontal homogeneity and constant turbulent fluxes throughout the surface layer, which is violated for a patchy snow cover and, as such, introduce a large uncertainty (e.g. Mott et al., 2018; Schlögl et al., 2018a). Bonekamp et al. (2020) successfully showed the potential of DNS to simulate snow and ice melt in complex terrain. Through several sensitivity tests, they assessed the influence of surface properties on the micrometeorology and subsequent effect on the turbulent heat fluxes. Moreover, the simulations are used to further enhance our insight in the fundamental processes of melting ice cliffs. Thus, DNS show high potential to be used as a tool to further enhance our understanding of local-scale advection and eventual implementation of the process in larger models through the derivation of new parametrizations accounting for varying surface and meteorological characteristics.

Whereas previous studies pioneered in examining the lateral advection of sensible and latent heat most often for a single snow patch, it remains in question how this process is affected by the typical spatial and temporal scales within a catchment. Moreover, new modelling attempts need to be undertaken to increase our understanding of the wind-driven processes occurring

near the surface of melting patchy snow covers. Therefore in this study, we aim to assess the role of horizontal advection of the sensible and latent heat on snowmelt for a snow patch in a real world case and idealized environment. In the real world case, we will identify the role of the locally advected sensible and latent heat on a melting snow patch in the Finseelvi catchment through studying the vertical turbulent heat fluxes with SfM photogrammetry observations over the course of multiple days. The resulting snowmelt is compared to local meteorological measurements to put the snowmelt in perspective and extract the role of the turbulent fluxes on this melt. Subsequently, we try to uncover the behaviour of the vertical sensible heat fluxes on snowmelt, including the local-scale advection of sensible heat, in an idealized environment with DNS, allowing to extract detailed information on wind blowing over a small flat domain with a patchy snow cover. This allows us to illustrate the performance of DNS as a tool to understand the real world behaviour and try to explain this with idealized simulations. To do so, we perform these simulations with the Computational Fluid Dynamics (CFD) code MicroHH. We use the measurements of Harder et al. (2017) on a single snow patch as a basis for designing our numerical experiments, and choose to focus on the sensible heat flux. These measurements are done in close to idealized settings on a flat surface. Subsequently, we investigate the influence of enlarging snow patches on the vertical sensible heat fluxes into the snow and the implications for snowmelt modelling.

## 2 Snowmelt observations

The snowmelt observations are done in the Finseelvi catchment near Finse, Norway (17.6 km$^2$; 60.60 N, 7.51 E). The catchment is a snowmelt-dominated headwater basin with the discharge outlet located at an altitude of approximately 1340 meter a.s.l. (Fig. 1). The catchment has a relatively smooth topography, which decreases the spatial complexity of the turbulent heat fluxes over snow patches (e.g. Mott et al., 2011). Combining this with a high contribution of the turbulent heat fluxes to snowmelt, which is generally the case for maritime climates (Conway and Cullen, 2013; Sicart et al., 2008), makes the Finseelvi catchment a suitable location for assessing the role of the vertical turbulent heat fluxes on melting patchy snow covers. In the same region hydrological studies like Filhol et al. (2019) applied SfM timelapse photogrammetry for a patchy snow cover and Harding (1986) estimated the exchanges of mass and energy for a melting continuous snow cover. Moreover, at a distance of approximately 2.5 km to the southeast from the outlet of the catchment, meteorological measurements are taken (Fig. 1). The meteorological data is retrieved from a meteorological flux tower at a temporal resolution of 30 minutes and includes, amongst others, temperature, precipitation, wind speed and direction, incoming radiation and relative humidity.

We assessed the role of the vertical turbulent heat fluxes on a melting single snow patch within the Finseelvi catchment. From this snow patch (Fig. 1), daily height maps are obtained through photogrammetry over the course of 11 June until 15 June 2019 for the upwind and downwind edge of the snow patch. As the dominating wind direction experienced at the snow patch resembled the measured wind direction at the meteorological tower, which was constant throughout the field campaign (Table 4 in Sect. 4), we assume that the upwind and downwind location of the snow patch are approximately constant. The length of the snow patch is approximately 50 m with a maximum snow depth in the regions of interest estimated to be in the order of 0.5 m. Selection of the location was based on the following criteria: a relatively flat surface and the absence of

complex topographical features nearby, which could complicate the incoming radiation by, for instance, partial shading. The height maps enable us to derive the amount of snowmelt during these 5 days for this single snow patch and assess the role of the vertical turbulent heat fluxes. To do so, the meteorological data is averaged over the period between the photogrammetry observations and compared with the daily melt observations. Also, three snow samples are taken to determine the snow density, such that the measured height changes can be converted to a volume. The samples are taken by digging a small snow pit and collecting 100 mL samples at 5, 25 and 45 cm below the snow surface on 14 June. The snowpit is dug in snow adjacent to the snow patch, such that the measurements are similar to the snow patch, but do not affect the photogrammetry observations. We are aware that taking these samples only on a single day and location does not reflect the potentially complex temporal and spatial dynamics of the snow density. However, we assume the variations occurring on these scales to be relatively small compared to other uncertainties introduced by our analysis for computing contribution estimates of the vertical turbulent heat fluxes to the snowmelt.

The height maps are obtained by applying the photogrammetric principle of Structure from Motion (SfM) using MicMac (Rupnik et al., 2017). A total of 1087 pictures (610 for the upwind edge, 477 for the downwind edge) were taken from various positions for both edges spread out over the 5 days, using a Xiaomi A2 smartphone camera. By using a ground-based camera, we are not able to obtain height maps covering the entire snow patch, which does prohibits a detailed analysis of the snowmelt. Yet, using this method, we illustrate that with a simple and cheap method, it is still possible to come up with relatively decent snowmelt estimates. The method is based on Filhol et al. (2019), who studied the melt of a relatively large snowfield with 3 time-lapse cameras over the course of an ablation season. In this study, MicMac is used to initially determine the camera positions and orientations. Subsequently, tie points appearing on multiple images for all days are obtained, such that the orientation of the camera positions can be determined. Considering images from all days during the tie point retrieval allows the eventual height models per day to be coregistered from the start. Eventually, each timestep is processed using the MicMac implementation of multi-view stereopsis to create point clouds for each day, which are interpolated into orthoimages and digital elevation models (DEM). The orthoimages are in RGB colors, allowing to distinguish snow from bare ground and determine the amount of snowmelt for snow-covered pixels ($0.04 \times 0.04$ m) in the DEM.

Both types of grids are available for each of the 5 days and for both locations, being the upwind and downwind edge, such that we have 20 grids in total. For both locations, the following post-processing is done after obtaining the grids:

1. For all grids, remove isolated groups of cells which are smaller than 0.05 m$^2$

2. For all grids, apply a median filter of $5 \times 5$ pixels to diminish the influence of noise located within the areas of interest, but maintain the sharp transitions between snow and snow free surfaces.

3. Compute the median height of the bare ground cells per day. The cells selected for this computation should be covered by all grids and already be bare ground on the first day.

4. Compute correction heights through comparing the daily median heights of the bare ground with the median height of the first day (step 3).

5. Remove bare ground cells out of DEM, based on orthoimages of the same day, such that only snow-covered cells remain for each day.

6. Apply correction height (step 4) on snow covered DEM (step 5) for each day.

7. For snow covered grid cells that are present on each day (step 5), we calculate height differences between the DEMs of the first day and other days (step 6). We remove absolute height differences larger than 50 cm, as larger values are highly unlikely to occur.

The resulting height differences over time correspond to 6.7 m$^2$ and 30.7 m$^2$ for respectively the upwind and downwind edge. We chose to solely use grid cells that are continuously covered by snow and have a recorded height change on each day, to reduce the chance of cells being random scatter. Additionally, this method does not include cells with relatively shallow snow depths of which the recorded melt could be affected by the presence of the bare ground below the snow.

We are aware that these filters have an effect on the number of analyzed grid cells and could cause an underestimation of the amount of snowmelt, especially on the upwind edge due to the varying locations of snow covered grid cells and the retreating snow line (Figure A1). For the downwind edge, the approximately constant location of the snow covered grid cells combined with the little retreat at this edge, causes this area to be significantly larger. As a disadvantage of the sizes of the covered areas, we decided to treat the edge as "point" and not consider the spatial distribution of the recorded melt.

To determine the snowmelt based on the net radiation, the measured incoming radiation is combined with estimates of the outgoing radiation based with the help of common snow characteristics. The outgoing shortwave radiation is calculated by assuming a snow albedo between 0.6 and 0.8. These values are based on Harding (1986), who reports albedos of approximately 0.8 for the same region in May. We use both of these values in the following calculations to account for the uncertainty in the albedo, due to spatial and temporal variations, and other potential uncertainties in the shortwave component. For the longwave radiation component, we assume it to be an appropriate estimate for the larger region. Furthermore, we assume that the snowpack is continuously ripe throughout research period, given that air temperature was continuously above freezing point and the largest discharge peak had already taken place 1.5 months before the observation period. This allows to compute the outgoing longwave radiation according to the Law of Stefan-Boltzmann (i.e. $LW^{\uparrow} = \epsilon \sigma T_{sn}^4$). It is assumed that the emissivity $\epsilon$ is close to unity (Dozier and Warren, 1982) and the snow surface temperature is 273.15 K, such that the outgoing longwave radiation is approximately 315 W m$^{-2}$. This results in the following equation for net radiation:

$$R_{net} = (1 - \alpha)SW^{\downarrow} + LW^{\downarrow} - \sigma T_{sn}^4, \tag{1}$$

in which $R_{net}$ (W m$^{-2}$) is the net radiation, $\alpha$ (−) the snow albedo and $\sigma$ (W m$^{-2}$ K$^{-4}$) the constant of Stefan-Boltzmann. The net radiation is subsequently recalculated to the melt of the snow expressed as height change, by combining this with the

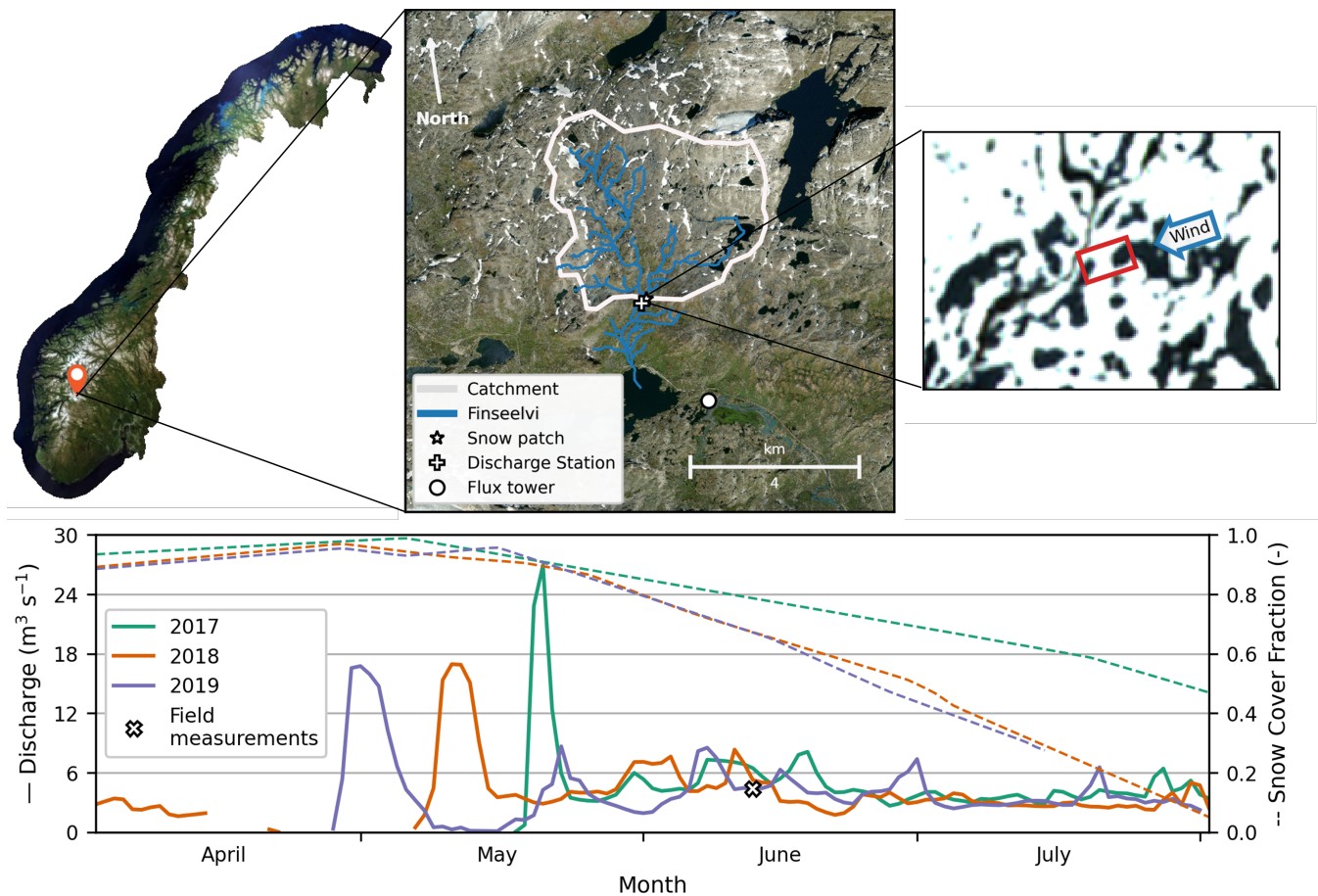

**Figure 1. Overview of the Finseelvi research area with the location of the snow patch and daily averaged discharge and SCF data of the catchment.** The map of Norway is obtained from https://norgeibilder.no/, the catchment area and the streams are respectively obtained through the University of Oslo and the Norwegian Water Resources and Energy Directorate (http://nedlasting.nve.no/gis/). The zoomed image is made by the Sentinel-2 satellite on 13 June 2018 with the snow patch (white areas are snow covered) and local wind direction indicated as experienced during the field campaign, which resembled the measured direction at the meteorological tower.

snow density and constant for latent heat of fusion, which is 334 KJ kg$^{-1}$. This can be recalculated, such that eventually the radiation-driven melt is expressed as height change:

$$M_R = \frac{R_{net} * \Delta t}{\rho_{sn} * L_f},$$

(2)

in which $M_R$ (m) is the snowmelt due to radiation expressed as height change, $\Delta t$ (s) the time between the photogrammetry observations, $\rho_{sn}$ (kg m$^{-3}$) the snow density and $L_f$ (J kg$^{-1}$) the constant for latent heat of fusion. Subsequently, we assume that the total melt is caused by radiation and the vertical turbulent heat fluxes (e.g. Plüss and Mazzoni, 1994), such that the

difference between the total observed melt and the computed radiation-driven melt can be attributed to the these turbulent heat fluxes.

To obtain an indication of the influence of the latent heat on the melt, relative humidity is recalculated to the vapor pressure difference between the air and snow surface ($e - e_{sn}$) and, subsequently, the specific humidity difference ($q - q_{sn}$). To calculate $e - e_{sn}$, we assume the vapor pressure of the snow to be the saturated vapor pressure of air at 0°C, which is 0.613 kPa. We are aware that the used relative humidity (measured at the meteorological tower) probably does not reflect the exact conditions at the snow patch. Therefore, we only use these computed values as an indication of the contribution of the latent heat on the

melt.

When comparing the measured meteorological conditions with the snowmelt, especially the net radiation (Eq. 1 and 2), the spatial variability of these conditions should be considered. For this comparison, we apply the measured values without any additional computations other than time-averaging, introducing additional uncertainty. For example, the snow patch is located at the bottom of a north-south orientated side valley, which causes shading at sunrise and sunset. The most prominent mountains

to the east and west of the snow patch are approximately 150 to 200 m higher at 1 to 1.5 km distance from the snow patch. The meteorological measurements are done in an east-west orientated main valley, such that less shading occurs at sunrise and sunset.

## 3 Idealised system

### 3.1 System description

We used an idealised system (Fig. 2) to study the turbulent heat fluxes in detail and understand the behaviour observed in the field. As this is one of the first studies using DNS to investigate the role of the vertical turbulent heat fluxes and local heat advection in these systems, we focus on the sensible heat flux, even though MicroHH allows to include the latent heat flux (e.g. Bonekamp et al., 2020). Instead of using our own measurements, the idealised system is based on the measurements on a single snow patch done by Harder et al. (2017), due to the availability of relatively high resolution measurements and similarity

to an idealized system in which the contribution of the local-scale advection of the sensible and latent heat to the total melt is relatively large. We are aware that this complicates the comparison between our observations and simulations. The prescribed conditions within simulations of the idealised system are based on the observations of Harder et al. (2017) and obtained through a dimensional analysis (elaborated on in Sect. 3.2). We assume our idealised system to consist of an on average near-neutral atmosphere above a patchy surface with heterogeneous properties and can be described by the following variables:

$$(\nu, \kappa, u, \delta, \lambda_{elem}, \lambda_{sn}, b_{sn}, b_{bg}).$$    (3)

The viscosity $\nu$ (m$^2$ s$^{-1}$), thermal diffusivity $\kappa$ (m$^2$ s$^{-1}$) and wind speed $u$ (m s$^{-1}$) describe the properties of the neutral atmosphere. $\delta$ (m) is the height of the atmospheric surface layer, $\lambda_{elem}$ (m) represents the average size of one snow patch

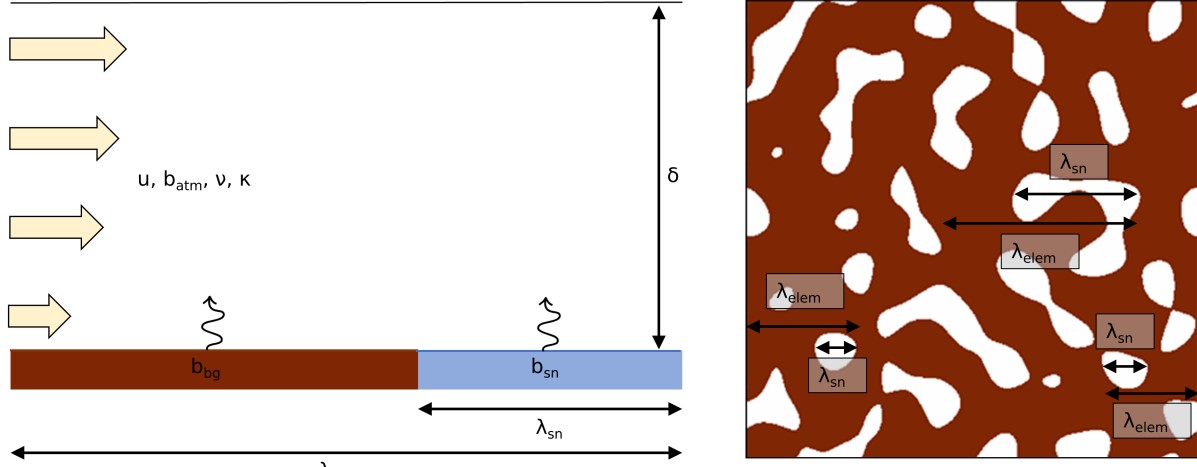

**Figure 2. Conceptual situation sketches.** Sketches of one snow patch and the adjacent bare ground, i.e. an element, (left) and an exemplary horizontal domain with indicated element and snow patches (right). The parameters represent viscosity $\nu$, thermal diffusivity $\kappa$, wind speed $u$, height of the atmospheric surface layer $\delta$, average size of one snow patch element $\lambda_{elem}$, consisting of the length of the snow patch $\lambda_{sn}$ and the adjacent bare ground ($\lambda_{elem} \equiv \lambda_{bg} + \lambda_{sn}$), and the buoyancy of the snow $b_{sn}$ and bare ground $b_{bg}$.

element, consisting of the snow patch itself ($\lambda_{sn}$) and the adjacent bare ground ($\lambda_{elem} \equiv \lambda_{bg} + \lambda_{sn}$). The buoyancy of the surface layer over the snow ($b_{sn}$) and the bare ground ($b_{bg}$) are defined as

$$b \equiv \frac{g}{\theta_{atm}}(\theta - \theta_{atm}), \tag{4}$$


in which $g$ (m s$^{-2}$) is the gravitational acceleration, $\theta_{atm}$ (K) is the temperature of the atmosphere and $\theta$ can be any temperature to be recalculated to the buoyancy $b$ in this equation. This definition causes the temperature dimension to cancel out and m s$^{-2}$ to remain as unit. In the simulations, we assumed for each simulation that initially the simulated atmosphere is well-mixed, such that the temperature of the atmosphere is constant over height (similar to Harder et al., 2017). Furthermore,

we assumed that the horizontal extent is orders of magnitudes larger than the elements, such that this does not have an influence on the physics in the model.

To assess the impact of the snow patch size, we varied $\lambda_{sn}$ in our simulations. Specifically, we doubled and quadrupled $\lambda_{sn}$ compared to a reference simulation with, similar to Harder et al. (2017), 15m-long snow patches. Therefore, in the remainder of this paper we will denote the reference simulation with P15m, and the simulations with a doubled and quadrupled snow

patch size as P30m and P60m respectively. The patches in the P60m simulation (and some larger patches in the P30m) allow to compare with our own observations and study the dominant processes. Although, the meteorology in the simulations is not based on the field observations, it allows for a qualitative analysis of the processes.

Furthermore, an additional simulation is performed in which the system is the same as in the P15m simulation, except that stability effects are excluded. This allows to identify the influences of stability on the vertical sensible heat fluxes, as well

as the dominating turbulent character. In this simulation, temperature does not influence buoyancy of the air. To refer to this simulation in figures, we append "NB", i.e. the abbreviation for "No Buoyancy", to the name, i.e. P15m-NB.

## 3.2 Dimensional analysis

### Parameter derivation

To create a system physically similar to the measurements done by Harder et al. (2017) within our modelling domain, a

dimensional analysis according to Buckingham's Pi Theorem (Buckingham, 1914) is used. The modelling domain in which we fit these measurements is based on to the atmospheric turbulent channel flow from Moser et al. (1999).

All of the descriptive variables (Sect. 3.1) are nondimensionalized by using $\delta$ and $u$ as repeating variables, since these cover both the primary dimensions, respectively [L] and [L T$^{-1}$], and are constant throughout all the numerical experiments (Sect. 3.4). Thus, this results in six dimensionless groups, which can be combined into:

$$\left( \frac{\nu}{\kappa}, \frac{u\delta}{\nu}, \frac{\lambda_{sn}^2}{\lambda_{elem}^2}, \frac{\delta}{\lambda_{elem}}, -\frac{b_{sn}\delta}{u^2}, -\frac{b_{bg}\delta}{u^2} \right). \tag{5}$$

The nondimensional parameters can be interpreted as follows:

$\frac{\nu}{\kappa}$ is the Prandtl number (Pr). This is assumed to be constant throughout this study, such that the flow over the patchy surface always has the same characteristics and does not influence the outcome of the simulations. In this study, the number is set to one, instead of 0.71, which is the atmospheric value. This deviation has negligible impacts on the flow and allows

for simpler scaling, when analyzing the simulations (Kawamura et al., 1998).

$\frac{u\delta}{\nu}$ is the Reynolds number (Re). For the same reason as the Prandtl number, this parameter is taken constant throughout the study. During this study, the same Reynolds number as Moser et al. (1999) is taken, being $1.10 \cdot 10^4$. Therefore, we impose a wind speed of 0.11 m s$^{-1}$, a 1 meter vertical extent and a viscosity of $1.0 \cdot 10^{-5}$ m$^2$ s$^{-1}$.

$\frac{\lambda_{sn}^2}{\lambda_{elem}^2}$ is the SCF and varies between zero and one. If this parameter is one, the field is completely snow covered, whereas

values close to zero represent low snow cover fractions. For all simulations, this dimensionless group is set to a value of 0.25 similar to Schlögl et al. (2018a), as Harder et al. (2017) does not provide any information about this.

$\frac{\delta}{\lambda_{elem}}$ is a measure for the size of a surface element compared to the height of the system. This nondimensional number will decrease, if the typical element size increases. If the SCF is kept constant, changes in this variable will affect the typical snow patch size, meaning that this variable can be interpreted as a measure for the relative snow patch size.

$-\frac{b_{sn}\delta}{u^2}$ is the bulk Richardson number above the snow patches ($Ri_{sn}$). Positive and negative values indicate respectively stability and instability, whereas the absolute values specify the magnitude of the (in)stability.

$-\frac{b_{bg}\delta}{u^2}$ is the bulk Richardson number above the bare ground ($Ri_{bg}$). For $Ri_{bg}$ the same principles hold as for $Ri_{sn}$.

We ensure that the increases in snow patch size $\lambda_{sn}$ necessary for our analysis, only affect the relative element size $\frac{\delta}{\lambda_{elem}}$. We do this by changing $\lambda_{sn}$ in tandem $\lambda_{elem}$, such that the snow cover fraction, $\frac{\lambda_{sn}^2}{\lambda_{elem}^2}$, remains constant. This allows us to study the impact of increasing snow patch size separately from the snow cover fraction.

**Parameter estimation**

To determine the values of the defined dimensionless parameters, first the measurements of Harder et al. (2017) are used to calculate these values (Table 1). Note that not all variables used in the dimensionless parameters are known, and are therefore estimated based on other literature, such as Schlögl et al. (2018a) (Table 2). Subsequently, the variables coming from Moser et al. (1999) are filled into the nondimensional parameters, such that the remaining dimensionless parameters can be calculated for all simulations.

It should be brought up that for the P60m simulation, the nondimensional number $\frac{\delta}{\lambda_{elem}}$ is rather low and could potentially have affected the outcomes. During this simulation the number was 0.83, whereas the number was originally designed to be approximately 1 or higher, such that the relative snow patch size would not be too large. The implications could be that the largest turbulent structures arising, due to the snow patches, do not have enough space to develop and are therefore affected by the horizontal domain of the simulation. However, the results show no clear deviations from the expectation, and thus we assume that the influence of this value of the nondimensional number is relatively minor to non-existent.

**Table 1. Dimensionless parameter values.** Overview of the applied values for the dimensionless parameters during the simulations.

|         | Pr   | Re                 | SCF  | $\delta/\lambda_{elem}$ | $Ri_{sn}$ | $Ri_{bg}$ |
|---------|------|--------------------|------|------------------------|-----------|-----------|
| **P15m**    | 1.00 | $1.10 \cdot 10^4$ | 0.25 | 3.33                   | 0.66      | 0.00      |
| **P15m-NB** | 1.00 | $1.10 \cdot 10^4$ | 0.25 | 3.33                   | 0.66[1]   | 0.00[1]   |
| **P30m**    | 1.00 | $1.10 \cdot 10^4$ | 0.25 | 1.67                   | 0.66      | 0.00      |
| **P60m**    | 1.00 | $1.10 \cdot 10^4$ | 0.25 | 0.83                   | 0.66      | 0.00      |

[1]These are in fact not real Richardson numbers as buoyancy does not affect the flow in the
P15m-NB simulation.

### 3.3 Rescaling the model results to reality

After nondimensionalizing the system for the simulations, the outcomes of the simulations are rescaled back to realistic scales to compare with observations. In order to do so, the non-dimensional numbers are used, as these numbers characterize the dominant processes in the considered idealized system.

For rescaling the surface heat fluxes back to reality, the $Ri_{sn}$ affects the rescaling factor being used. The number describes whether shear-driven turbulence or buoyancy-driven turbulence or a balance between the two is more dominant in the system,

**Table 2. Variable values applied in the simulations.** Overview of the values used for obtaining the dimensionless parameters in the simulations. * are values that are estimated based on Schlögl et al. (2018a).

| Variables | Harder et al. (2017) | Moser et al. (1999) | Simulations |
|---|---|---|---|
| $\nu \, (m^2 \, s^{-1})$ | - | $1.0 \cdot 10^{-5}$ | $1.0 \cdot 10^{-5}$ |
| $\kappa \, (m^2 \, s^{-1})$ | - | $1.0 \cdot 10^{-5}$ | $1.0 \cdot 10^{-5}$ |
| $u \, (m \, s^{-1})$ | 6.4 | 0.11 | 0.11 |
| $\delta \, (m)$ | 100* | 1.00 | 1.00 |
| $\lambda_{elem} \, (m)$ | 30* | - | 0.30, 0.60, 1.20 |
| $\lambda_{sn} \, (m)$ | 15 | - | 0.15, 0.30, 0.60 |
| $b_{sn} \, (m \, s^{-2})$ | $-0.28$ | - | $-8.0 \cdot 10^{-3}$ |
| $b_{bg} \, (m \, s^{-2})$ | 0.0 | - | 0.0 |

and thus also which process predominantly affects the surface buoyancy flux scaling. We assume shear-driven turbulence to be dominant near the surface, such that the rescaling equation becomes

$$B_0 = u b_{sn}, \tag{6}$$

in which $B_0$ is the typical buoyancy surface flux (m$^2$ s$^{-3}$), $b_{sn}$ the surface buoyancy of snow (m s$^{-2}$) and $u$ is the wind speed (m s$^{-1}$). This equation is derived from the original equation for the surface buoyancy flux implemented in the model

$$B_0 = -\kappa \frac{\partial b}{\partial z}\bigg|_{sn} \approx -\kappa \frac{\delta b}{\delta z}\bigg|_{sn},$$
$$\sim \kappa \frac{b_{sn}}{\delta_v}, \tag{7}$$

in which $\delta_v$ is the depth of the viscous sublayer (m), which is assumed to be in the order of magnitude of $\delta z$, whereas $b_{sn}$ is assumed to be a measure for $\delta b$. For $\delta_v$, we use the viscous sublayer instead of the boundary layer, as the latter is approximately neutral, such that buoyancy is constantly zero throughout the layer except near the surface.

Equation 8 follows from a definition of the height of the viscous sublayer $\delta_v$, for conditions in which shear dominates, such that the $\delta_v$ is determined by wind and viscosity. This allows to set up a relation between these three variables

$$\delta_v = f(u, \nu),$$
$$\sim \frac{\nu}{u}. \tag{8}$$

When implementing this into Eq. 7, to make Eq. 6

$$B_0 \sim \kappa \frac{b_{sn}}{\left(\frac{\nu}{u}\right)} = \kappa u \frac{b_{sn}}{\nu} = u b_{sn},$$ (9)

in which $\kappa$ and $\nu$ cancel out, since $Pr$ (i.e. $\frac{\nu}{\kappa}$) is unity in this study. Subsequently, this equation can be used to rescale the simulated surface buoyancy fluxes back to realistic values, according to

$$\frac{B_{0,sim}}{B_{0,real}} = \frac{u_{sim} b_{sn,sim}}{u_{real} b_{sn,real}}.$$ (10)

From Table 2 it follows that $b_{sn,sim} = -0.008$ m s$^{-2}$, $b_{sn,real} = -0.28$ m s$^{-2}$, $u_{sim} = 0.11$ m s$^{-1}$ and $u_{real} = 6.4$ m s$^{-2}$, thus the equation reduces to

$$\frac{B_{0,sim}}{B_{0,real}} = \frac{0.11 \times -0.008}{6.4 \times -0.28} = 4.9 \times 10^{-4}.$$ (11)

Thus, the simulated values for the surface buoyancy fluxes are divided by $4.9 \times 10^{-4}$ to compute the realistic values. Subsequently, the outcome for the realistic surface heat fluxes can be transformed to W m$^{-2}$ according to

$$H_{real} = \rho c_p \frac{\theta_0}{g} B_{real},$$ (12)

in which $H$ is the sensible heat flux into the surface (W m$^{-2}$), $\rho$ the density of the air (kg m$^{-3}$), $c_p$ the specific heat capacity (J kg$^{-1}$ K$^{-1}$), $\theta_0$ the reference temperature, being 273 K in this case and $g$ the gravitational acceleration (m s$^{-2}$).

For rescaling simulated time back to reality, the bulk Richardson number above snow ($Ri_{sn} = -\frac{b_{sn}\delta}{u^2}$) is used for obtaining a timescale. From this number the following timescale is derived

$$\frac{\delta}{u} = t_{adv}$$ (13)

in which $t_{adv}$ (s) resembles an advective timescale.

When calculating the ratios between the simulated and measured timescale by making use of the values in Table 2, the following ratio between the simulated and realistic timescale arises

$$\frac{t_{adv,sim}}{t_{adv,meas}} = \frac{9.09}{15.63} = 0.58,$$ (14)

Thus, the simulated time is divided by 0.58 to compute the realistic values.

## 3.4 Model description and set-up

In this study, the model simulations are performed using the MicroHH 2.0 code, which is primarily made for DNS of atmospheric flows over complex surfaces by van Heerwaarden et al. (2017). When solving the conservation equations for mass, momentum and energy, MicroHH makes use of the Boussinesq approximation, such that the evolution of the system for velocity vector $u_i$, buoyancy $b$ and volume is described by

$$\frac{\partial u_i}{\partial t} + \frac{\partial u_j u_i}{\partial x_j} = -\frac{\partial \pi}{\partial x_i} + \delta_{i3} b + \nu \frac{\partial^2 u_i}{\partial x_j^2},$$
$$\frac{\partial b}{\partial t} + \frac{\partial u_j b}{\partial x_j} = \kappa \frac{\partial^2 b}{\partial x_j^2},$$
$$\frac{\partial u_j}{\partial x_j} = 0, \tag{15}$$

in which $\pi$ is a modified pressure (van Heerwaarden and Mellado, 2016). Moreover, MicroHH uses periodic boundary conditions in the horizontal directions, which implies that we simulate a wind blowing over an infinite snow field. In the following parts of the article, when reporting vertical sensible heat fluxes into the surface, these are recalculated from the surface buoyancy flux $B$ (m$^2$ s$^{-3}$) computed in the model equation according to

$$B = -\kappa \frac{\partial b}{\partial z}\bigg|_{surface}, \tag{16}$$

which can be recalculated to realistic sensible heat fluxes following the steps in Sect. 3.2. The horizontally advected sensible heat is computed similar to Harder et al. (2017) (Eq. 2), being

$$H_{adv} = \int_{z=0m}^{z=2m} \rho c_p \bar{u} \frac{\partial \bar{\theta}}{\partial x} dz, \tag{17}$$

in which z (m) is the elevation above the surface, $\rho$ (kg m$^{-3}$) the air density, $c_p$ (1005 J kg$^{-1}$ K$^{-1}$) the specific heat capacity of air, x (m) the distance from the leading edge of the snow patch. Next to integrating over 2 m profile height, we also integrate over 4 meter height.

The starting point for designing the numerical experiments is the atmospheric turbulent channel flow with a reduced Reynolds number ($Re$) designed by Moser et al. (1999). This simulation is also used as spin-up, during which we have no buoyancy effects included in the flow. The shear Reynolds number $Re_\tau$ obtained from the measurements performed by Harder et al. (2017) is relatively high compared to Moser et al. (1999); $\sim 6 * 10^6$ vs. 590, but the results of Moser et al. (1999) suggest that the bulk statistics, i.e. mean and variances, at least the initial neutral channel flow is hardly affected by this difference in $Re_\tau$. This channel flow is simulated in MicroHH at a resolution of $384 \times 192 \times 128$ grid points, for a domain size of $2\pi$ m $\times \pi$ m $\times 2$ m. The flow is forced in the x-direction by imposing an average wind speed, which is in this case 0.11 m s$^{-1}$. At the bottom and top boundary, no slip and no penetration conditions are applied to the velocities (i.e. the flow velocity at

the boundary is zero). Overall, van Heerwaarden et al. (2017) showed that MicroHH is well able to reproduce this turbulent channel flow.

Initially, the turbulent channel flow from Moser et al. (1999) used as spin-up is simulated until 1800 seconds, such that the
turbulent channel flow has well developed. Subsequently for each simulation, which take 900 seconds, this turbulent channel flow is adapted, such that an atmospheric flow over a patchy snow surface is obtained. On the bottom boundary, a pattern of surface buoyancies depending on the simulation is prescribed, such that the surface characteristics determined during the dimensional analysis are fulfilled. For snow and bare ground, the surface buoyancy is respectively $-8 \times 10^{-3}$ m s$^{-2}$ and 0.0 m s$^{-2}$ (i.e. 273 K and 280.9 K in reality), which is elaborated on in Sect. 3.2. The implemented surface for the P15m and
P15m-NB (P30m, P60m) contains snow patches of on average 0.15 m (0.30 m, 0.60 m) and an average element length of 0.30 m (0.60 m, 1.20 m) (Table 2; multiply with 100 in Fig. 3). These surfaces enable the model to solve the periodic boundary conditions, because we ensured that the patches at the opposing walls fit together and flow that leaves the system on one side, continues over the same snow patch when it re-enters the system on the opposite wall. These patterns are created by generating noise in the Fourier space around specific wavelengths, prescribed in the form of 2D power spectra. When transforming these
back to physical space using the Inverse Fast Fourier Transform, a 2D field with dominant patterns is obtained. For a more elaborate explanation on generation see Appendix B.

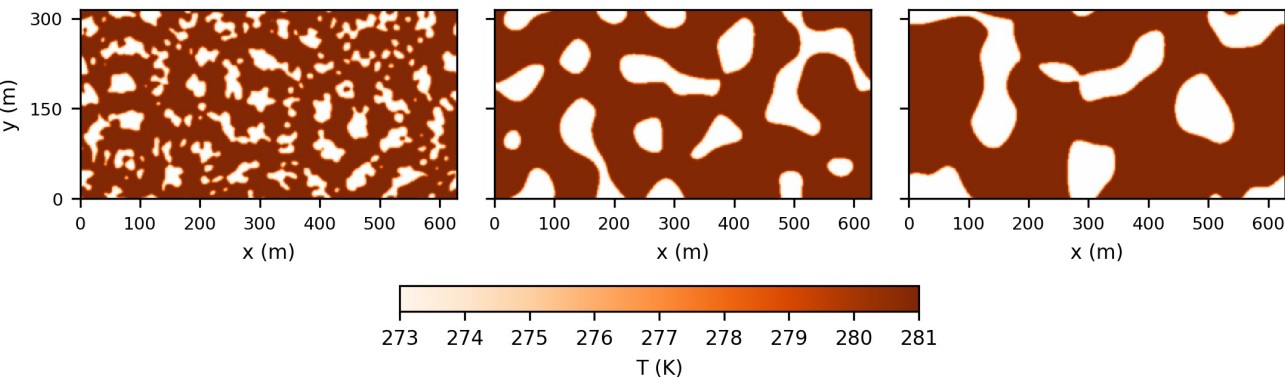

**Figure 3.** Generated realistic surface temperatures for the P15m and P15m-NB (left), P30m (middle), P60m (right) simulations.

## 4 Field observations

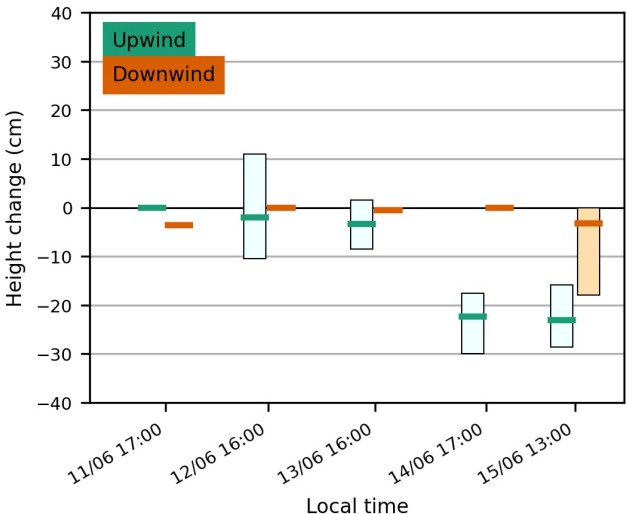

**Figure 4. Boxplot of height changes for the upwind and downwind edge of a snow patch recorded with SfM photogrammetry from 11 June 2019 until 15 June 2019.** For the upwind edge the 11 June is taken as reference date. For the downwind edge, the first measurement on 11 June is not used as reference, due to a low coverage of grid cells. The melt is estimated with an error of $\pm 2.0$ and $\pm 0.4$ cm, respectively for the upwind and downwind edge.

At the upwind edge of the snow patch, the snow surface decreased approximately 23 cm, whereas for the downwind edge this decrease is approximately 3 cm (Fig. 4). For the upwind edge, the height change is relative to 11 June 17:00 LT. For the downwind edge, the height change is computed relative to 12 June 16:00 LT, due to a low coverage of bare ground pixels on 11 June (Fig. A1), which are used for computing the vertical correction to align all height maps (Table 3). For the upwind edge the melt relative to 12 June 16:00 LT is estimated to be 21 cm.

The average standard error after applying the vertical correction for the bare ground grid cells, that are present throughout the all height maps, is for the upwind edge 1.38 cm and for the downwind edge 0.29 cm. Propagation of these errors ($\sqrt{2\sigma_{\Delta\bar{h}}^2}$), due to relating the height changes between two height models, allows to estimate the melt with an error of $\pm 2.0$ and $\pm 0.4$ cm, respectively for the upwind and downwind edge. Usually, these standard errors are calculated over other grid cells than those used for the vertical correction, such that these points are independent. In this study, however, the standard error and vertical correction are computed for all bare ground grid cells present throughout the research period, since the relatively small spatial scale causes all grid cells to be related. Considering the errors, for the upwind edge significant melt was recorded on 13 June and onwards compared to 11 June as reference. For the downwind edge, significant melt was recorded on 13 and 15 June taking 12 June as reference. For the third period from 13 to 14 June an increase in snow height relative to the previous day was recorded, but this increase was not significant due to overlapping error estimates.

**Table 3. Vertical shift correction (cm) applied on the individual height models per day for the upwind and downwind location.** * is only computed with the bare ground grid cells present in this height map.

|          | 11-06  | 12-06  | 13-06  | 14-06  | 15-06  |
|----------|--------|--------|--------|--------|--------|
| **Upwind**   | 0.00   | −0.69  | 0.71   | −0.32  | −1.11  |
| **Downwind** | −3.57* | 0.00   | −0.48  | −0.05  | −0.01  |

## 4.1 Snowmelt related to meteorology

Relating the meteorological circumstances (Table 4) to the measured snowmelt, allows to estimate the contribution of the vertical turbulent heat fluxes to the total amount of snowmelt. To do so, the snow density measurements are used. The snow densities at 5, 25 and 45 cm below the snow surface are respectively 556 kg m$^{-3}$, 551 kg m$^{-3}$ and 610 kg m$^{-3}$. Following from these densities, the snow density is approximately constant near the surface, whereas further down the snow is more compressed or stores water. Even though these densities are relatively high, we consider the values to be realistic, based on the largest discharge peak taking place 1.5 months before the observation period (Fig. 1) combined with the notice that the snowpack was relatively wet while being in the field.

**Table 4. Average meteorological measurements and calculated variables in between the photogrammetry observations.** $T_{2m}$ is the air temperature measured at 2 meter, $u_{10m}$ and $u_{dir}$ are respectively the wind speed and wind direction in degrees from the north measured at 10 meter, $P_r$ is the summed precipitation during the period, $SW^{\downarrow}$ and $LW^{\downarrow}$ are respectively the incoming shortwave and longwave radiation, $RH_{2m}$ the measured relative humidity at 2 meter and $P_a$ the air pressure. The net radiation ($R_{net}$), subsequent melt ($M_R$) and specific humidity difference ($q - q_{sn}$) are computed based on a combination of the measured variables. The ranges in $R_{net}$ and $M_R$ are caused by applying two values for the albedo, i.e. 0.6 and 0.8 to account for uncertainties in the shortwave radiation component (see Sect. 2).

| Period in LT | Measured variables | | | | | | | | Calculated variables | | |
|---|---|---|---|---|---|---|---|---|---|---|---|
| | $T_{2m}$ (min−max) | $u_{10m}$ | $u_{dir}$ | $P_r$ | $SW^{\downarrow}$ | $LW^{\downarrow}$ | $RH_{2m}$ | $P_a$ | $R_{net}$ | $M_R$ | $q - q_{sn}$ |
| | (°C) | (m s$^{-1}$) | (°) | (mm) | (W m$^{-2}$) | (W m$^{-2}$) | (%) | (kPa) | (W m$^{-2}$) | (cm) | (g kg$^{-1}$) |
| **11/06 17:00 − 12/06 16:00** | 5.5 (3.8 − 10.2) | 7.4 | 121 | 4.5 | 53 | 327 | 82 | 88.3 | 23 − 33 | 1.0 − 1.5 | 0.92 |
| **12/06 16:00 − 13/06 16:00** | 3.8 (2.5 − 7.5) | 5.1 | 136 | 9.0 | 83 | 329 | 94 | 87.4 | 31 − 47 | 1.4 − 2.2 | 1.00 |
| **13/06 16:00 − 14/06 17:00** | 6.7 (4.8 − 9.1) | 7.4 | 121 | 0.6 | 159 | 309 | 83 | 87.8 | 36 − 58 | 1.3 − 2.8 | 1.42 |
| **14/06 17:00 − 15/06 13:00** | 7.6 (1.9 − 14.2) | 2.5 | 157 | 0.3 | 305 | 285 | 75 | 88.1 | 31 − 92 | 1.2 − 3.6 | 1.20 |

The minimally estimated melt due to net radiation from 12 June until 15 June is 3.9 cm (Table 4), which is 0.5 cm larger than the melt estimate including the error for the downwind edge. As the snow patch was approximately 50 meter in length, the turbulent heat fluxes into the snow have likely reduced to negligible values at the downwind edge (e.g. when extrapolating the measurements of Harder et al., 2017), such that this mismatch is most likely caused by uncertainties in the net radiation estimates. When assuming this radiation to be an appropriate estimate and homogeneously spread over the patch, the estimated contribution of the vertical turbulent heat fluxes at the upwind edge is 13.0 to 18.2 ± 2.0 cm. For this, we also assume that the residual of the difference between the observed snowmelt and radiation-driven snowmelt is caused by these turbulent heat

fluxes (e.g Plüss and Mazzoni, 1994). When using Eq. 2, to compute the average vertical turbulent heat fluxes during the period, we estimate these fluxes at the upwind edge to be between 73 and $102 \pm 11$ W m$^{-2}$. These are in the same order as found by other studies in somewhat similar conditions, such as Mott et al. (2018), Olyphant and Isard (1988) and Harding (1986). Mott et al. (2018) reported for typical weather situations in alpine areas a sensible heat flux into the snow of up to 50 W m$^{-2}$, whereas the latent heat was negligible or even contributed to the cooling of the snow. Olyphant and Isard (1988) report downward turbulent heat fluxes of over 100 W m$^{-2}$ for a large single snow patch. Harding (1986) estimated the full energy balance for 15 days in May of a relatively homogeneous snow cover near Finse and reported a sensible heat flux of approximately 20 W m$^{-2}$ and an on-average-negligible latent heat flux. For our observations, however, it is expected the latent heat flux had also a significant contribution due to the high relative humidity and subsequent difference in specific humidity between the air and snow of at least 0.9 g kg$^{-1}$ (Table 4) during the field campaign, which is common for Finseelvi as it is located in a maritime climate. Our results show similar moisture gradients as Harder et al. (2017), who found a significant influence of the latent heat on snowmelt, such that we expect this to be a significant part of our estimated turbulence-driven melt. Overall, by comparing the overall melt with the melt driven by the radiation and taking the residual as turbulence-driven melt, we estimate the contribution of the turbulent heat fluxes to the snowmelt to be roughly 60 to 80 % for the upwind edge of the snow patch. Extrapolating this to the entire catchment, applying the relations reported by Schlögl et al. (2018b) for two test sites in the Alps, we estimate the contribution of the fluxes to the total melt to be maximally in the order of 10 %, under the assumption that the entire catchment behaves similar.

For the entire snow patch, it is likely that the incoming radiation at the snow patch is overestimated, due to topography. The incoming solar radiation at the snow patch is blocked by mountains at low solar angles in the east and west, which does not apply to the location of the meteorological observations. Moreover, the incoming longwave radiation reduces with height, as was found in the Alps by Marty et al. (2002), such that the 150 m higher located snow patch has received less incoming longwave radiation. Lastly, other characteristics influencing the net radiation, such as varying snow albedo and slightly different slopes for the upwind and downwind edge, are also identified as potential influences on the results.

# 5    Model simulations

## 5.1    System characteristics

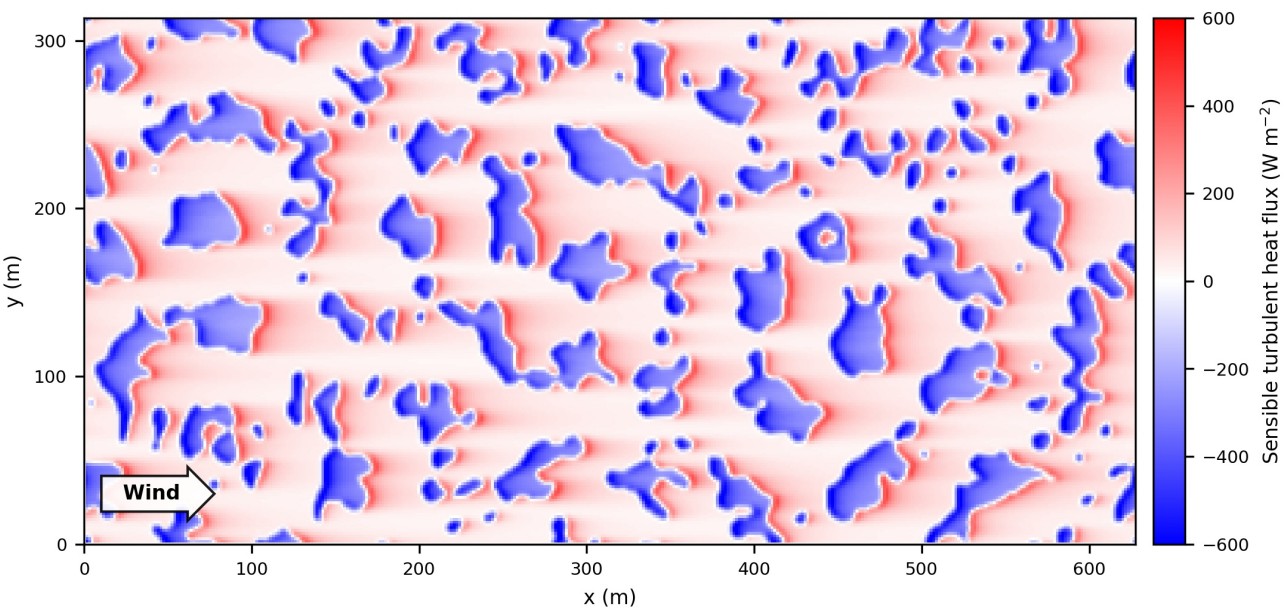

**Figure 5. Time-averaged surface sensible turbulent heat fluxes.** Sensible turbulent heat fluxes in the P15m simulation averaged from 2000 seconds until 2700 seconds. Negative values indicate a downward flux, i.e. over snow patches, whereas positive values resemble an upward flux, i.e. over bare ground.

In the idealized simulations, the time-averaged sensible turbulent heat fluxes resemble the implemented surface pattern of snow patches (Fig. 5). There is a negative surface flux on the snow patches, whereas at the bare ground the surface flux is positive. For each single snow patch, there is a clear pattern arising somewhat similar to the observations. The leading edge of the snow patches shows the highest fluxes towards the surface, $\sim -500$ W m$^{-2}$, and decreases downwind of the leading edge until the end of the patch. Subsequently, at the trailing edge of the snow patch and leading edge of the bare ground, the sensible heat flux changes sign, as the bare ground is relatively warm compared to the colder air coming from above the snow patch, resulting in $\sim 300$ W m$^{-2}$. The air warms when flowing over the bare ground, such that when the air arrives at the next snow patch, it is relatively warm compared to the cold snow patch, causing a high downward flux. These fluxes at the leading edge of the snow patches are relatively high, mostly due to the ideal circumstances for wind-driven melt, including local-scale advection of sensible heat (i.e. high wind speeds and relatively large temperature differences). Compared to our own observations, the sensible heat fluxes at the leading edge are much larger than the observed combined turbulent heat fluxes. This is in line with

our expectations because of the inclusion of not only ideal circumstances for local-scale advection of the sensible and latent heat during the field campaign.

## 5.2 Total snowmelt

When comparing the average sensible heat fluxes for all the snow patches in the domain, clear differences arise between all simulations (Fig. 6). The highest sensible heat fluxes towards the snow (i.e. most negative) are found in the simulation without buoyancy effects. This also causes the total sensible heat flux for this simulation to be significantly lower than the other simulations. Furthermore, increasing snow patch size reduces the heat fluxes into the snow patches. The heat fluxes of the simulation with doubled snow patch size (P30m) decrease with approximately 15% relative to the P15m simulation. For the simulation with quadrupled snow patch size (P60m), the heat fluxes reduce with approximately 25%. This is in contrast to the results of Schlögl et al. (2018a), who reports a minor influence of snow patch size on the amount of melt. Our findings are more in line with the results of Marsh et al. (1999), who based there work on a 2D Boundary Layer Model with a regular tiled surface pattern. Potentially, the differences with Schlögl et al. (2018a) are caused by the disability of ARPS to fully resolve the leading edge effect, due to a too coarse resolution to resolve the thin internal stable boundary layer formed over snow patches and the violation of the Monin-Obukhov assumptions. Both of these limitations do not apply for DNS.

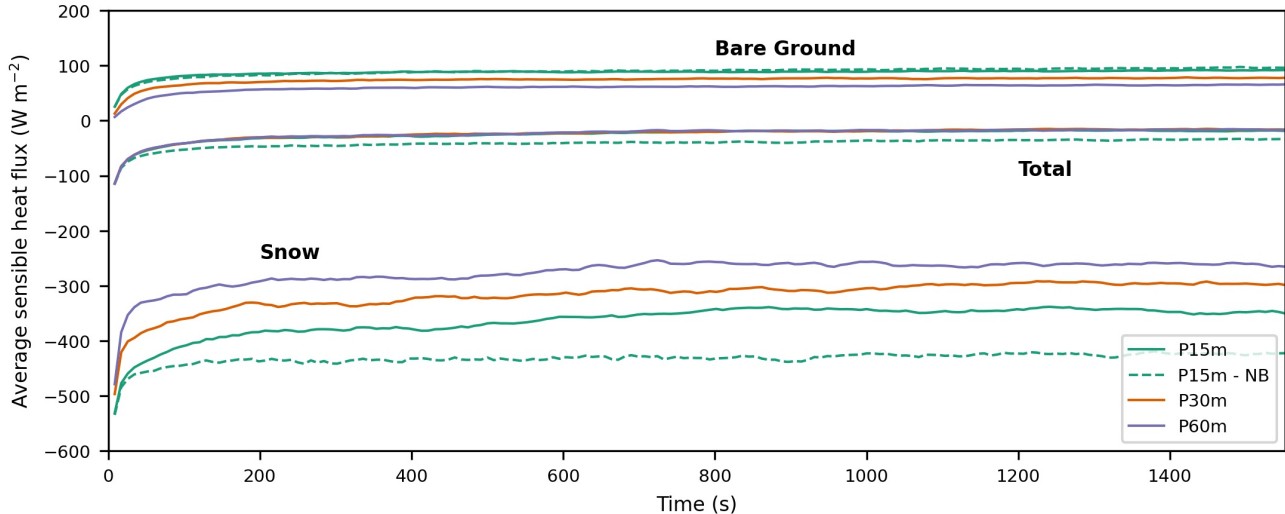

**Figure 6. Domain averaged sensible heat fluxes for different surfaces.** A time series of the averaged surface sensible heat fluxes for the bare ground, snow and total surface after introduction of the temperature differences at the surface.

The total heat fluxes for the simulations with 15, 30 and 60 m snow patches coincide approximately, as the differences arising at the snow surface are compensated for at the bare ground surface. So, although the total fluxes are equal, the snowmelt does vary with snow patch size. The simulation without buoyancy effects (P15m-NB) has a significantly reduced total heat flux

compared to the original simulation (P15m). This is caused by similar averaged heat fluxes for the bare ground for both simulations, whereas this is not the case above the snow patches. This suggests that stability does have little effect on the fluxes above the bare ground, whereas above the snow the surface heat fluxes are affected by stability.

Moreover, the largest adjustment of the sensible heat fluxes after initiation of the simulations is done after less than 200
seconds. However, on a longer term a minor trend is still present for each simulation. For this study, we assume that after the largest adjustment the dominant processes are well developed and suffice to understand the system. We expect that eventually the total summed surface sensible heat fluxes will go to zero, due to the infinite blowing over the snow cover without any other heat fluxes than those originating from the snow patches and bare ground. However, as the volume of the channel is relatively large compared to the heat fluxes, it takes relatively long before the whole system has cooled to reach an equilibrium.

**5.3 Surface fluxes for individual patches**

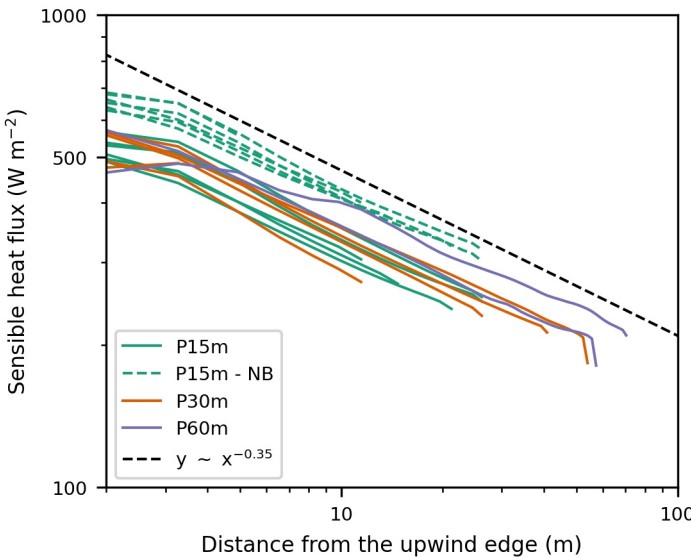

**Figure 7. Time-averaged surface sensible heat fluxes over single patches of snow.** The surface sensible heat fluxes for individual patches with snow as a function of distance from the leading edge on a log-log scale at y = 1.1 m. The dashed black line is an implemented trendline to show the approximate power law decay of the fluxes over distance. The fluxes of the snow are multiplied with −1, such that these values can also be plotted on logarithmic axes.

The surface fluxes show for all individual snow patches a similar behaviour for each simulation over distance from the leading edge, but the differences above snow patches occur due to varying fluxes at the leading edge (Fig. 7). The linear behaviour of

the surface fluxes as a function of the distance from the upwind edge on logarithmic axes implies that the fluxes decay over distance from the leading edge according to a power law. The power laws take on the following approximate forms:

$$H_{sn}(x) \equiv C_{sn} x^{-0.35}, \tag{18}$$

in which $H_{sn}$ (W m$^{-2}$) is the sensible heat flux, $x$ (m) is the distance from the leading edge and $C$ (W m$^{-1.65}$) is a constant representing the initial conditions at the leading edge of each patch.

Our simulated vertical sensible heat fluxes into the snow are approximately 500 W m$^{-2}$ and at the upwind edge and 200-300 W m$^{-2}$ at the downwind edge. In comparison to our field observations, at both edges these sensible heat fluxes are relatively high. At the upwind edge, the simulated sensible heat fluxes are approximately 5 times larger than the derived contribution of the combined turbulent heat fluxes to the measured snowmelt. We reckon that the simulated values are large, though it should be noted that the simulations are based on highly ideal conditions for turbulence-driven melt and local-scale advection of sensible and latent heat, whereas the conditions during the measurements were not ideal (e.g. nighttime melt is included). At the downwind edge, the measurements suggest an approximately negligible contribution of the vertical turbulent heat fluxes to the snowmelt, whereas the simulations show at comparable snow patches a significant contribution of the sensible heat flux ($\sim$ 200-300 W m$^{-2}$). Thus, the simulated decay of the sensible heat flux seems to be an underestimation in comparison to field observations. We expect that the comparable behaviour of the sensible heat fluxes between patches found within the idealized system, also is occurring within the Finseelvi catchment for patches within similar local conditions.

Figure 7 shows that the length of snow patches is the main cause of less snowmelt for larger patches, which was also found by Marsh et al. (1999). The power laws are approximately the same for each patch and simulation, whereas $C_{sn}$ is the same for each simulation except the simulation without buoyancy effects. This behaviour explains why larger snow patches reduce the average surface fluxes into the snow patches.

Striking is the behaviour of the simulation without buoyancy effects, as the decay of the fluxes for this simulation are similar to the decay of the fluxes for the other simulations, suggesting that stability has little influence on the decay of the surface heat fluxes, i.e. shear turbulence dominates. However, this simulation has a relatively lower (higher absolute flux in Fig. 7) initial surface flux above the snow, indicating an effect of the stability on the leading edge conditions. Thus, the differences in total snowmelt (Fig. 6) between the P15m and P15m-NB simulations solely occur due to these differences at the leading edge.

The decay of the sensible heat fluxes are a consequence of the decreasing temperature gradients in the IBL (Fig. 8a). Wind speed, another important component affecting the sensible heat flux, remains constant over a snow patch (Fig. 8b). Moreover, surface roughness is the same for the entire domain. Strikingly, the average temperature within the IBL remains constant and does not depend on the height of this IBL, as the shape of the vertical temperature profile remains constant while the flow proceeds over the snow patch (Fig. 8a inset). Yet, it should be noted that the reported values for the height of the IBL are relatively high compared to Harder et al. (2017).

Compared to Harder et al. (2017), our estimates of the horizontally advected sensible heat are relatively high. The measurements done by Harder et al. (2017) show values slightly above 400 W m$^{-2}$ for the first 3.6 m (Fig. 8c). In our simulations the

horizontal advection of sensible heat decreases in the first 3.6 m from 577 W m$^{-2}$ to approximately 400 W m$^{-2}$. For the following 4.8 m, Harder et al. (2017) reported an average reduction in horizontally advected sensible heat to approximately 20% of values found for the first 3.6 m. In the comparable simulation with a similar dominant snow patch pattern, this reduction is not found. Yet, further downwind the advected sensible heat does reduce to values in the same order of magnitude. It should be

noted that, due to setting the integration height at 2 m in Eq. 17, not all changes in the vertical temperature profile over distance from the leading edge are included, such that the horizontally advected sensible heat is underestimated. When considering a 4 m integration height, the horizontally advected sensible heat is approximately equal to the vertical sensible turbulent heat fluxes at the snow surface, especially for the first half of the patch, implying also a power of -0.35 for the decay of the horizontally advected sensible heat. This also illustrates the major contribution of the horizontally advected sensible heat to the sensible

heat flux into the snow. We expect a similar role of the advected heat in our field observations, as in all directions and great distances from the observed snow patch, there was a patchy snow cover present, causing approximate equilibrium conditions. In comparison to a similar relationship obtained by Granger et al. (2002) (based on Weismann, 1977) through boundary-layer integration, our advected heat decays less. Granger et al. (2002) come up with a similar power law, but their powers maximally reach -0.47, depending on surface temperatures of snow and bare ground, friction velocity and roughness length.

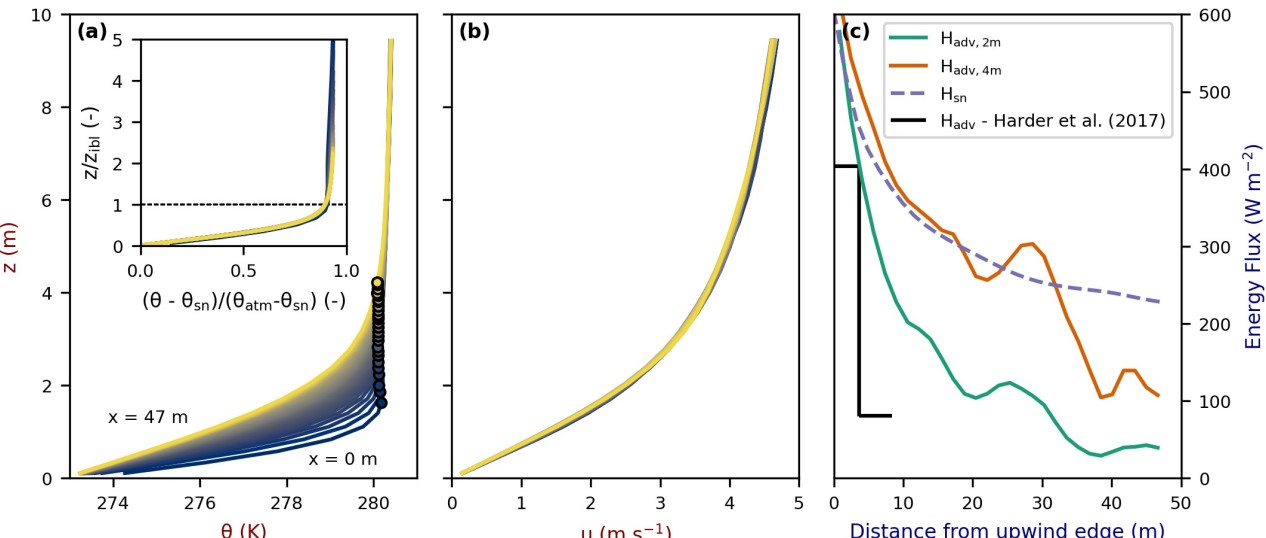

**Figure 8. Time-averaged vertical temperature (a) and wind profiles (b), and horizontally advected sensible energy $H_{adv}$ (at 2 and 4 meter integration height) and vertical sensible heat flux at the surface $H_{sn}$ (c) along the snow patch at y = 83 m and x = 150 m in the P30m simulation.** The markers in the temperature profile resemble the IBL height, which is computed as the first value (seen from the bottom) of the gradient of the temperature over height below 0.1 K m$^{-1}$. The first 2 upwind grid cells and 2 downwind grid cells have been removed, as these are located in the transitions region from bare ground to snow and vice versa. The labels with 'x =' show the location of vertical profile related to the downwind distance of the leading edge, whereas the line colors are based on the distance from the leading edge and go from purple to red. In the inset graph, the vertical temperature profile has been normalised with height of the IBL and prescribed temperature of the surface and atmosphere.The advected energy as a function of distance from the leading edge are based on Equation 17.

## 6 Discussion

This study aims to assess the role of local-scale advection of the sensible and latent heat fluxes on snowmelt. In order to do so, complementary field observations and simulations are performed. On small spatial scales, the largest melt differences due to the combined turbulent heat fluxes occur at the opposing edges of snow patches. Our results show that the upwind edge of a single snow patch in the Finseelvi catchment, which is approximately 50 m in length, melted $23 \pm 2.0$ cm over the course of 5 days, whereas the downwind edge melted just $3 \pm 0.4$ cm in 4 days. As the snow patch was approximately 50 meter, the vertical turbulent heat fluxes have likely reduced to negligible values at the downwind edge, due to the leading edge effect, such that the main cause of melt at this edge is the net radiation. The simulations allow to extract detailed information on the atmospheric flow and are used as a tool to provide insight in the evolution of the fluxes and temperature over the patchy snow cover. The sensible heat fluxes reduce over distance from the upwind edge following a constant power law, which likely depends on the meteorological circumstances. In the simulations, this results in a reduction of 15% and 25% for respectively a doubling and quadrupling in snow patch size. The simulations reveal that the reducing sensible heat fluxes over distance from the leading edge are caused by the reducing temperature gradients, pointing out the major role of the horizontally advected sensible heat, which we expect to behave similarly in our field observations. Other important factors on the turbulent heat fluxes, i.e. wind speed and surface roughness, are constant over distance from the leading edge in the simulations.

Though, the simulations lack the surface roughness differences for the snow and bare ground and topographical variations. Including the transition from a rough (bare ground) to smooth (snow) surface, likely would diminish the IBL growth due to increased turbulence levels and enhance the vertical sensible heat fluxes as a consequence of the larger temperature gradients in the IBL (Garratt, 1990). It should be noted that this mostly holds for shear dominated turbulence, whereas at lower wind speeds the influence of thermal turbulence on the IBL should also be considered. Moreover, the common formation of snow patches in topographical depressions causes atmospheric decoupling and reduced vertical turbulent heat fluxes at low and moderate wind speeds, especially downstream of the upwind edge (Mott et al., 2016). In the Finseelvi catchment snow patches have formed to some extent in these depressions, while this does not hold for Harder et al. (2017), Also, the choice of basing the numerical experiments on Harder et al. (2017) instead of our own observations, complicates an exact comparison. Lastly, the exclusion of external forcings, such as radiation or large-scale advection, causes the simulations to approach equilibrium, due to the compensation between the sensible heat fluxes at the snow patches and the bare ground. This is advantageously for investigating the behaviour of the system in relation to snow patch size, but makes it more difficult to compare other characteristics (e.g. temperature) with for example Schlögl et al. (2018a), who did include external forcings, such as incoming radiation. Overall, these mechanisms greatly increase the uncertainty and make us decide not to directly compare between the simulations and observation.

On a catchment scale, these simulation results imply that differences in snowmelt, within a highly idealized catchment, solely occur due to snow patch length. The sensible heat fluxes into the snow at the upwind edge of the patches are independent of snow patch size and show the same decay over distance from the leading edge, such that systems with typically larger patches have on average reduced sensible heat fluxes into the snow. The major cause of these fluxes seems to be the horizontal

advection of sensible heat. It should be noted that the latent heat flux, which is not considered in these simulations, can also play a significant role for the amount of snowmelt (Harder et al., 2017, 2019). For this flux, we expect similar mechanisms in our simulations based on the observations of Harder et al. (2017). Variations in surface roughness and topography mostly create micrometeorological circumstances which differ substantially from the average circumstances within a catchment, for example through shading or a slope-induced drainage flow, and thus also affect snowmelt. We expect that the important role of the horizontally advected sensible heat and the identical behaviour of the vertical sensible heat flux between patches, both found in the simulations, is also applicable to our field observations, given the probable larger-scale approximate equilibrium of the atmosphere. Though anomalies in this behaviour can be found due to varying micrometeorological conditions. Overall, our results imply that the performance of snowmelt predictions would improve when also considering snow patch size distributions. Information on and usage of these distributions can be obtained with various methods, ranging from relatively simple methods, for example scaling laws (e.g. Harder et al., 2019), to more complex methods, for example by assimilating various satellite retrievals (e.g Aalstad et al., 2018).

The melt estimates obtained with the SfM photogrammetry are in line with our expectations based on rough visual estimates from during the field campaign, whereas the estimated errors are relatively small. The errors are in the same order of magnitude as found for high-accuracy snow depth estimates obtained from time-lapse photography (e.g. Dong and Menzel, 2017; Garvelmann et al., 2013), though, it should be noted that these studies focus on somewhat larger areas. Overall, this illustrates that the influence of the vertical turbulent heat fluxes on melting snow patches is widespread and can even be observed with relatively simple and cheap methods. One of the potential limitations to this study, is the choice of solely using grid cells that are continuously covered causing the amount of available grid cells to reduce drastically and underestimate the snowmelt, especially at the upwind edge due to the retreating snow line. Advantageously, this shrinks the chance of grid cells being random scatter, which could result in unrealistic height changes. Moreover, the weather conditions on multiple days during the field campaign complicated the identification of tie points, due to a limited amount of light (Cimoli et al., 2017; Nolan et al., 2015). Lastly, the small angle between the camera positions and the horizontal snow surface was uncommon, as often the method is applied with camera positions at a higher incidence angle (e.g. with drone imagery). Overall, this caused not all objects to be captured from multiple perspectives, again complicating tie point identification. A possible solution to these limitations, could be to add passive control points in the snow to create more tie points for the software to connect. For more precise radiation, and thus melt estimates, this study could have benefited from radiation modelling using high-resolution terrain information (cf e.g. Silantyeva et al., 2020). Future studies could also make use of Lidar scanners, which have recently gone into mass production and hence seen the corresponding drop in cost.

A potential weakness of our simulations is the application of a low $Re_\tau$ compared to the observations of Harder et al. (2017) (i.e. 590 vs $\sim 6 \times 10^6$). This saves computational costs, which are relatively high for DNS, and was done based on the results of Moser et al. (1999), who showed large differences between $Re_\tau = 180$ and $Re_\tau = 395$, whereas $Re_\tau = 590$ has similar bulk quantities and variances as the latter simulation. Adjusting the $Re_\tau$ possibly has affected the surface momentum fluxes, of which $\frac{U}{u_\tau}$ is a measure, and thus also the heat fluxes. Furthermore, the low $Re$ has likely caused the fluxes in the IBL to be predominantly diffusive, whereas in reality the turbulent fluxes are more probable to dominate. As the typical diffusive

timescales are larger than the typical turbulent timescales, the typical time- and length-scales of the processes in the IBL are relatively large compared to reality, such that one of the processes affected by this, could be the decay of the vertical sensible heat fluxes over distance from the leading edge of a snow patch. To uncover whether the sensible heat fluxes and the decay are dependent of the $Re_\tau$, we recommend to perform simulations with higher $Re_\tau$. Increasing $Re$, will reduce the scale of the smallest eddies, i.e. the Kolmogorov length scale (Pope, 2000), such that an enhancement of the resolution possibly also is required. Next to the influence of using an idealised system, this makes that our formulated relationship between the sensible heat flux at the surface of the snow patches and the distance from the upwind edge (i.e. $H_{sn} \sim x^{-0.35}$) should be taken with caution. However, the method does illustrate the use of DNS to come up with potentially useful relationships. Future studies would need to look further into the above described behaviour, especially when more comparable high-resolution data is available.

Moreover, we identify some inaccuracies during the nondimensional scaling of the wind speed and temperature difference between the snow and atmosphere. As Harder et al. (2017) reported a wind speed of 6.4 m s$^{-1}$, this value is also considered in our dimensional analysis and related to 0.11 m s$^{-1}$, which was the average wind speed over the whole channel in the case of Moser et al. (1999). However, the reported wind speed was measured at 1.8 meter above the ground, thus implying that the average wind speed for the whole air column under consideration would be higher. Consequently this affects the leading edge effect, due to increased wind shear and, thus, also the fluxes towards the surface. Also, the temperature difference between the atmosphere and snow has been possibly overestimated, causing an increased sensible heat flux. The graphs presented by Harder et al. (2017) show the temperatures of bare ground and atmosphere to be constant near the surface, being 6.4 °C. However, for the dimensional analysis, the atmospheric temperature mentioned by Harder et al. (2017), i.e. 7.9 °C, has been used. Overall, these differences in assumptions between the simulations and the field observations make a one-to-one comparison difficult. Yet, the general behaviour found in the simulations is similar to previous literature, i.e. temperature profiles and melting patterns (e.g. Harder et al., 2017; Mott et al., 2016), and shows the potential of DNS as a modelling tool to understand the melting of a patchy snow cover. Especially, considering that DNS does not violate the assumptions for the Monin-Obukhov bulk formulations and is able to resolve the leading edge effect, in contrast to modelling studies with coarser spatial resolutions, which could lead to major errors (Schlögl et al., 2017). As such, this type of simulations is expected to provide a more realistic behaviour of the leading edge effect on the vertical turbulent heat fluxes, especially when combining with case-specific boundary conditions.

In the studied system, the influence of stability seems to be negligible for the relative decay of the surface fluxes over distance from the leading edge, since the sensible heat fluxes in the simulations with and without buoyancy effects show the same decay. However, the snowmelt in the simulation without buoyancy effects is still higher as the absolute sensible heat fluxes at the leading edge are highest for this simulation. We expect, that the decay is similar due to the relative high wind speeds compared to the temperature difference between the snow and bare ground; 6.4 m s$^{-1}$ and 8 K respectively. Overall, this causes the shear induced turbulence to dominate over the buoyancy induced turbulence. As multiple studies suggest the role of stability on snowmelt (e.g. Dadic et al., 2013; Essery et al., 2006) stable regions (i.e. snow patches) could have a much larger impact on the amount of wind-driven snowmelt, especially when reducing the turbulence towards the edge of collapsing.

Therefore, it would be interesting to reduce the wind speed and increase the temperature difference between the snow and bare ground, to identify which $Ri_{sn}$ is needed for stability to become a more important factor on the sensible heat fluxes.

## 7    Conclusions

In this study, we examined the melt of a 50 meter long snow patch in the Finseelvi catchment, Norway, and investigated the observed melt with highly idealized simulations. The melt estimates, obtained with relatively simple and cheap structure-from-motion photogrammetry, for the upwind and downwind edge of the snow patch are feasible and the estimated errors are in line with previous studies. The combined influence of the sensible and latent heat flux on the snowmelt at the upwind edge is estimated to be between 60 and 80%, while for the entire catchment this contribution would come down to be maximally in the order of 10% based on previous studies. This estimate is based on the difference between the recorded melt and net radiation of the snow patch determined with measurements of a meteorological tower near the catchment. This shows that under specific circumstances the local advection of the sensible and latent heat can be of major importance on the snowmelt of a patchy snow cover, expressing the necessity of a sound implementation of this process when modelling snowmelt.

In the idealized simulations, based on measurements done by Harder et al. (2017) on a single 15 m snow patch on a flat surface, the sensible heat fluxes reduce over distance from the leading edge following a constant power law. These reductions are caused by the cooling of the air above the snow patch while wind speed and surface roughness are constant over the snow patch. Other simulations, in which the typical snow patch length is doubled and quadrupled, show exactly the same behaviour over snow patches, such that larger snow patches receive on average less sensible heat. Domain averaged sensible heat fluxes even reduced with 15% and 25% for respectively a doubling and quadrupling of the typical snow patch size. Overall, this implies that the sensible (and likely also the untested latent) heat fluxes have a lower influence on the snowmelt in catchments with typically larger snow patches.

When comparing the simulated behaviour to the observed melt in the field, the observed vertical turbulent heat fluxes at the upwind edge are in the same order of magnitude as the simulations, especially when considering the inclusion of the diurnal cycle in these estimates. Moreover, based on the simulations, it is expected that the behaviour found in the simulations also explains the found reductions in the field. Though it should be noted that the decay of sensible heat fluxes over distance from the leading edge measured by Harder et al. (2017) was higher than the simulated decay, for which some potential causes are identified, such as a too low Reynolds number and inaccuracies in the nondimensionalisation.

Yet, the idealized simulations have shown the potential of direct numerical simulations when simulating a patchy snow cover, especially compared to the errors that have been found for other simulation types. All performed simulations show the ability to simulate the leading edge effect, and also clear IBLs form over snow patches. Whereas studies making use of LES, such as Schlögl et al. (2018a), report large errors compared with measurements. For our study, the flow characteristics are similar to controlled and field measurements. Next to the measurements done by Harder et al. (2017) in the field, the measurements of Mott et al. (2016) in a wind tunnel also show similar shapes for the temperature profiles above snow patches. Some characteristics vary compared to observations, such as the height of our IBLs compared to Harder et al. (2017), but the

general outcome seems promising for future research. Overall, the simulations allow to extract very detailed information on the atmospheric behaviour above a snow patch and can be used as a tool for a better understanding of melting patchy snow covers.

680 **Appendix A: Orthoimages**

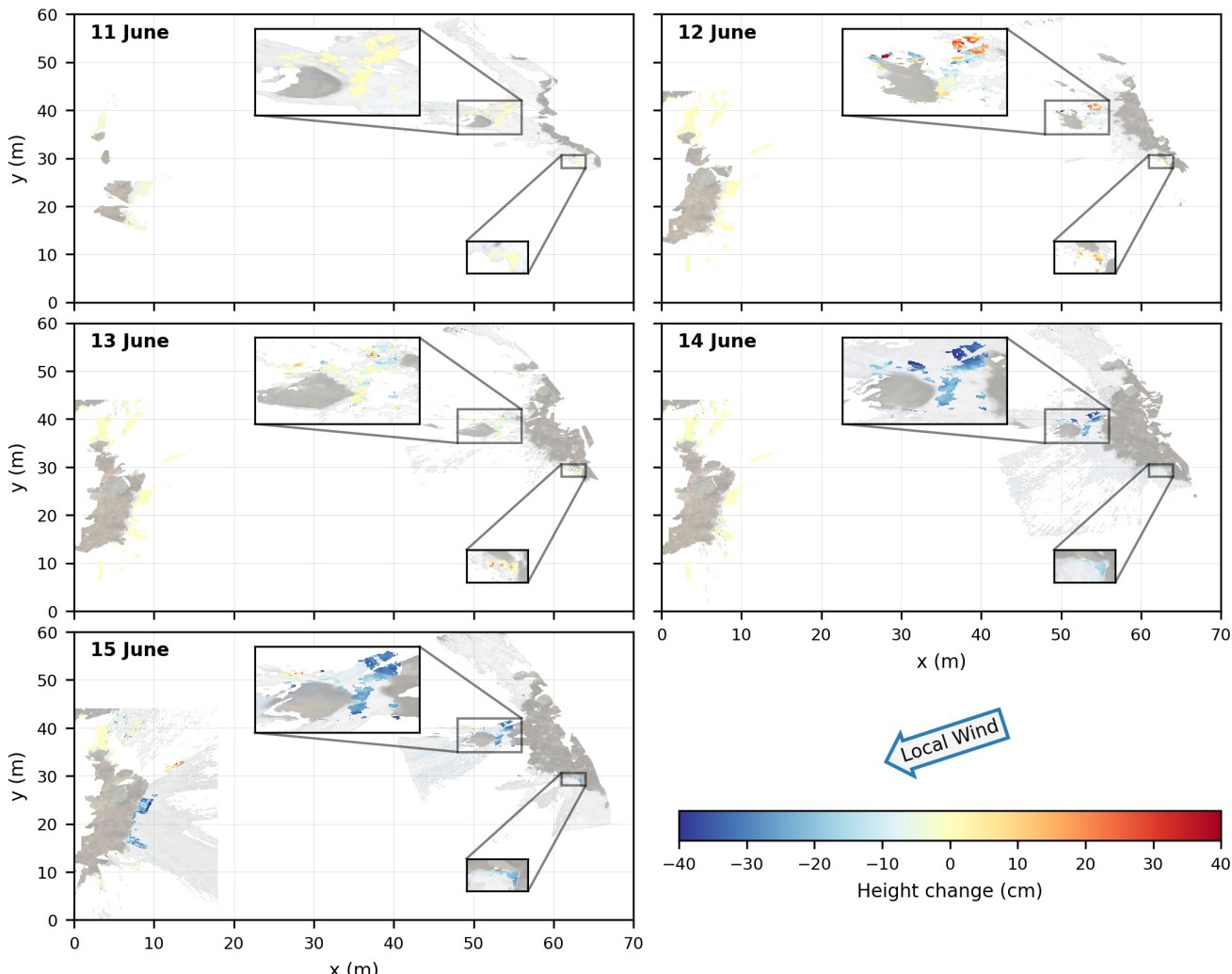

**Figure A1. The resulting height change maps used for Figure 4 plotted over the real-color orthoimages, which are used for distinguishing snow from bare ground, after removing isolated groups of cells and median filtering.** The insets show zoomed areas of the upwind edge for more detail.

## Appendix B: Surface generation

To create the surfaces for the simulations, noise is generated in the Fourier space, such that seemingly random patterns with a specified wavelength arise. These wavelengths are prescribed in the form of 2D power spectra. This method is applied, such that the patches at the opposing walls fit together and flow that leaves the system on one side, continues over the same snow patch when it re-enters the system on the opposite wall. This enables the model to solve the periodic boundary conditions.

Initially, a field with random phases between 0 and $2\pi$ is generated. These phases are applied in Euler's law (i.e. $e^{i\varphi} = \cos\varphi + i\sin\varphi$), such that the phases are described in exponential form. The phases are multiplied with the desired magnitude per phase, such that the Fourier space is generated ($z = |z|e^{i\varphi}$ and Figures B1 - B3). Eventually, a 2D field with dominant patterns is obtained by returning to physical space by using the Inverse Fast Fourier Transform on the Fourier space. Also, to avoid numerical instabilities in MicroHH a Gaussian filter is applied on the surface with a standard deviation of 1 grid cell. When this filter overlaps the edges of the domain, the values at the opposing edge are applied.

The specified spectra for all generated surfaces consist of two broad peaks (Figures B1 - B3). The peak with the lowest wavenumber has the higher factor and, thus, gives the dominant structures to the snow patch distribution. The peak with the higher wavenumbers (i.e. 3 times the average wavenumber of the main peak) has a lower factor, such that within the larger structures some smaller fluctuations occur, giving the patches a more realistic appearance. For the surface in the P30m and P60m simulations, the wavenumber of the main peak has been reduced with a factor 2 and 4 respectively compared to the surface in the P15m and P15m-NB simulations. This implies an average length of the snow patches in reality of 15 m, 30 m and 60 m for the P15m (and P15m-NB), P30m and P60m simulations respectively.

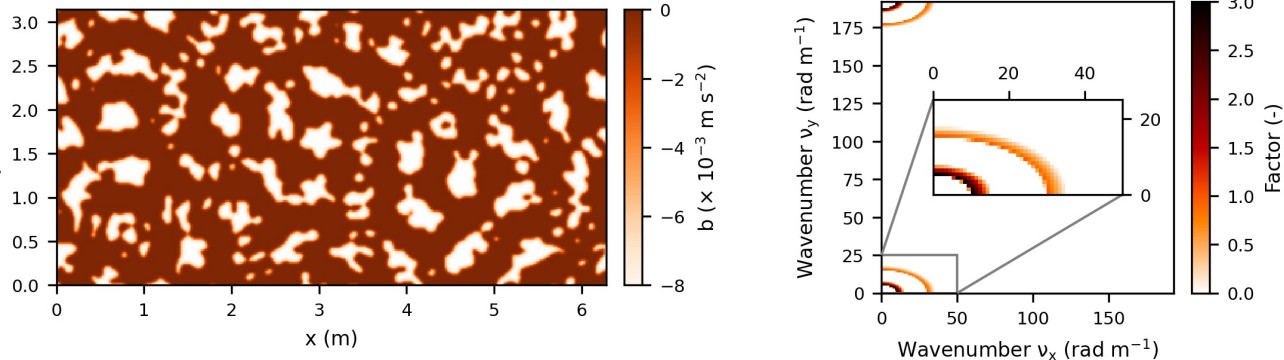

**Figure B1. Generated surface temperature and the applied Fourier space in absolute form for the P15m and P15m-NB simulations.** The generated surface temperature (left) obtained by applying the Inverse Fast Fourier Transform on the Fourier space (right).

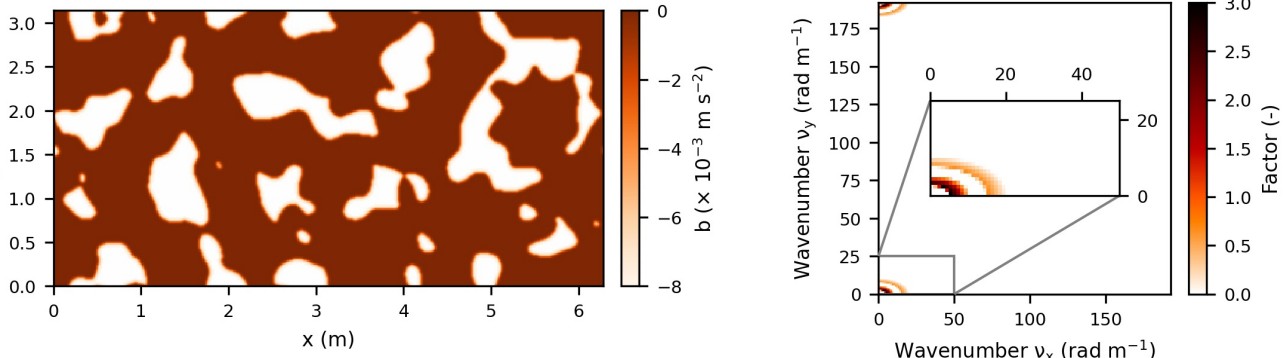

**Figure B2. Generated surface temperature and the applied Fourier space in absolute form for the P30m simulation.** The generated surface temperature (left) obtained by applying the Inverse Fast Fourier Transform on the Fourier space (right).

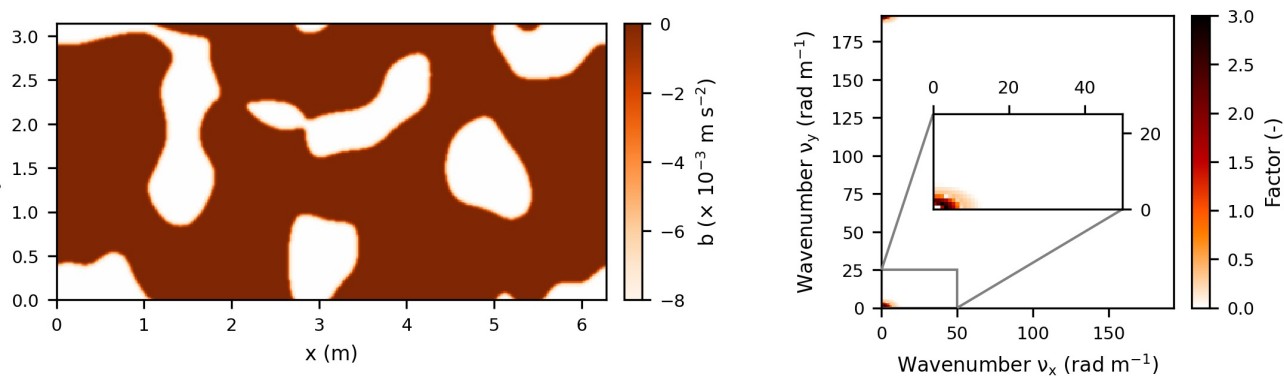

**Figure B3. Generated surface temperature and the applied Fourier space in absolute form for the P60m simulation.** The generated surface temperature (left) obtained by applying the Inverse Fast Fourier Transform on the Fourier space (right).

*Code and data availability.* For the snowmelt observations, the used images and a brief description of the photogrammetry workflow are available at https://doi.org/10.5281/zenodo.4704873 (van der Valk et al., 2021a). For the numerical experiments, exemplary input files and model output can be found at https://doi.org/10.5281/zenodo.4705288 (van der Valk et al., 2021b).

*Author contributions.* LvdV carried out the research under supervision of AJT, NP, RS, and CvH. LvdV designed the numerical experiment together with CvH and RS, and LvdV designed the field experiment together with AJT and NP. LG helped with performing the photogrammetry analysis. LvdV prepared the manuscript with contributions from all co-authors.

*Competing interests.*  The authors declare that they have no conflict of interest.

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
