# Peer review of "Understanding wind-driven melt of patchy snow cover"

_The Cryosphere, 2021_

## Author Comment (AC1)

Dear Referee,

We would like to thank you for taking the time to review our paper and for all your constructive suggestions, which will help to improve the quality of the paper. For now, we would like to answer to your major comments (and some larger specific comments). Our response to the comments appears in *italic*. We will take the remaining detailed comments into account when preparing a revised version.

1.  **Literature context**
    I get the impression that the authors may be newer to the snow field and so some of the statements in the introduction (see comments) need revising. Broadly there is a tremendous amount of research on processed based- energy balance snowmelt modelling that needs a clearer summary/context for this work. In terms of local-scale advection work there is a much more limited amount but can be found back to the 1970's. This needs a more complete treatment to solidify the context of this contribution as well as to distinguish these contributions from previous one. (See comments for examples).

    *We will include a more elaborate discussion of the amount of research done regarding this topic. Thank you for your suggestions.*
* * *
2.  **Validation with SFM**
    I'm not exactly clear on the process by which the snow depth difference are calculated via SfM on the observed patch in order to compute expected ranges of turbulent flux contributions and model validation. Are we only considering snow depth difference for grid cells that were completely uncovered during melt (so we can have a bare ground reference) or are we also considering cells that were not fully uncovered that would have also decreased in snow elevation due to melt? What areas do these number represent? Do you observed a decrease in melt from leading edge? Many dynamics can be examined with a spatial dataset but I'm not clear on how this is also processed/what it represents and so would welcome a lot of clarification and perhaps a figure to describe this process.

    *We agree that the text describing the SfM, and especially the post-processing of the DEM and orthoimages can be better formulated. Below we added a short description on how we used the DEMs and orthoimages, which we will also include in the revised version of the manuscript.*

    *We have 2 types of grids (DEM and orthoimages) per day (5 days) per location (upwind & downwind). So, we have 20 grids in total. The size of the grid cells is 0.04 m x 0.04 m.*
    *For both locations, the following post-processing is done after obtaining the grids:*
    1.  *Remove isolated groups of cells which are smaller than 0.05 $m^2$ (All grids)*
    2.  *A median filter of 5 × 5 pixels is applied to diminish the influence of noise located within the areas of interest, but maintain the sharp transitions between snow and snow free surfaces (All grids)*
    3.  *Compute the median height of bare ground cells per day. The conditions for the selecting the cells used during this computation:*
        a.  *Bare ground on first day*
        b.  *Covered by all grids*
    4.  *Compute correction heights (Table 3) through comparing the daily median heights of the bare ground (step 3) with the median height of first day*
        *E.g. median height bare ground cells of day 2 − median height bare ground cells of day 1 = correction value found in Table 3*

5. *Remove bare ground cells out of DEM based on orthoimages of the same day → snow covered cells remain*
6. *Apply correction height (step 4) on snow covered DEM (step 5)*
7. *For snow covered grid cells that are present on each day (based on step 5), we calculate height differences between the DEMs of first day and other days (both obtained in step 6)*

*The resulting height differences over time correspond to 6.7 $m^2$ and 30.7 $m^2$ for respectively the upwind and downwind edge. This might seem in contrast to what would be expected based on Figure A1 (in which the upwind edge shows a larger coverage). However, we chose to solely use grid cells that are continuously covered by snow and have a recorded height change on each day, to reduce the chance of cells being random scatter. As additional advantage this method does not include cells with relatively shallow snow depths of which the recorded melt could be affected by the presence of the bare ground below the snow and also be in the same order of magnitude as our melt error estimates. Our choice for these filters is supported by the fact that when loosening these filters, the size of the boxplots increases drastically, also to unrealistic values and variations in snow surface height, such as large increases over the course of these 5 days.*

*We are aware that this has an effect on the number of analyzed grid cells, especially on the upwind edge due to the varying locations of snow covered grid cells or the retreating snow line (Figure A1). For the downwind edge, the approximately constant location of the snow covered grid cells combined with the little retreat at this edge, causes this area to be significantly larger. Even though these resulting areas are relatively small, we are convinced that the obtained height changes obtained are decent estimates, also based on our error estimates.*

*Unfortunately, as a disadvantage of the size of the upwind area consisting of multiple separate smaller areas, we decided to treat the edge as "point" and not look further into the spatial distribution of the recorded melt (e.g. is there a decay in the melt?). The smaller areas are too far apart to do so.*
* * *
3. **Latent heat flux decay**
There is an extensive treatment of the sensible heat flux decay with patch length while this is not discussed with respect to latent heat flux. Can this also be included or is there are reason it is not included. The heavily cited Harder 2017 paper suggest that latent heat is also an important contributor or at times compensatory (Harder 2018) and so would be very interested in seeing if some of those dynamics could be captured in this modelling scheme.

*MicroHH does allow to include the latent heat flux (e.g. Bonenkamp et al., 2019), however we chose to only consider the sensible heat flux to explore the potential of using DNS for studying this kind of system.*

*We will elaborate on this when revising our paper, such that our intentions are better formulated.*
* * *
4. **Implications**
There are some interesting dynamics explained but I'm not exactly clear on how those could be implemented in larger scale snowmelt prediction. There is a scaling relationship articulation for sensible heat over a patch length. Is this considered to be a parameterization that could be used in basin scale snowmelt prediction models.

*Indeed, a scaling relationship would make it possible to parametrize these type of processes. However, we are not certain that our relationship is appropriate for implementation, due to some limitations of our methods, such as the use of an idealized system. Future studies would need to look further into this relationship, especially when more data is available. In a revised version, we will better emphasize our concerns about the applicability of the formulated relationship.*
* * *
Line 138-140: when were these samples taken with respect to the observation interval as snow density is dynamic over melt? Was a snow tube used? Snow pit? How were 100ml samples collected? Did the melt period have a consistently ripe snowpack?

Line 160-165: How deep was the snowpack and do you have any information to say that the snowpack was ripe at the start of the melt. Were the cold content requirements satisfied at the start of the period and so all energy could be assumed to be related to melt.

Line 334-336: It seems SWE and density are being used interchangeably here which is not correct. Can this be cleared up? These are pretty high densities. Any observations from field notes about water saturation or other structural attributes. What was the overall snow depth variability? Can you report the SWE of the snow patch?

*The samples were taken by digging a small snow pit and collecting 100 ml samples at 5, 25 and 45 cm below the surface at June 14 (4$^{th}$ day in the field). We are aware that taking these samples only on a single day does not reflect the potentially complex temporal dynamics of the snow density. However, we assume the variations occurring on these temporal scales to be relatively small compared to other uncertainties introduced to our method for computing contribution estimates of the turbulent heat fluxes to the snowmelt.*

*We agree that the measured snow densities are relatively high. Yet, we do think that these densities are realistic and represent a continuously ripe snowpack, given the fact that largest discharge peak had taken place already 1.5 month before the fieldwork (Figure 1) and the air temperature never decreased to freezing point during the campaign (Table 4). Additionally, during the campaign it was noted that the snow pack was relatively wet. In a revised version of the manuscript, we will add these considerations regarding the magnitude of the observed snow densities.*

*Lastly, indeed in L334-336 the interchangeable use of the SWE and snow density is incorrect, and will be adapted in a revised version.*
* * *
Line 398-403: Granger et al., 2002 and Weisman, 1977 propose similar power law relationship to describe a sensible heat flux. Perhaps worth contrasting this formulation and the meaning of your terms with those papers?

*Thank you for this suggestion, we are currently looking into this.*

---

## Author Comment (AC2)

Dear Referee,

We would like to thank you for taking the time to review our paper and for all your constructive suggestions, which will help to improve the quality of the paper. For now, we would like to answer to your major comments (and some larger specific comments). Our response to the comments appears in *italic*. We will take the remaining detailed comments into account when preparing a revised version.

1. The authors describe a process they term advection of turbulent heat flux and reference studies discussing local advection of sensible heat as described in Mott et al. (2018) and also Harder et al., (2017). It is not clear to me to which term the authors are really relating to as it seems to me that they mix up advection of sensible heat with the vertical turbulent sensible heat flux. The ambiguity becomes particularly clear when the authors compare modelled sensible heat fluxes with estimated advected sensible heat as presented in Harder et al (2017). I recommend to include equations where they clearly state at which terms they are looking at and how these are calculated. Equations for advection of sensible heat are presented in Harder et al. (2017) and Mott et al. (2020).

   *In a revised version, we will describe more clearly which process we are considering. Also, we will include equations stating which set of equations is used by the model (e.g. van Heerwaarden and Mellado., 2016; equation 12) and which terms we are considering.*

   *Regarding the comparison with the results of Harder et al. (2017), we do think that the performed dimensional analysis for setting up the DNS is consistent with the system presented by Harder et al. (2017). However, we do realize that Figure 7 needs some revision. We will adapt Figure 7, such that we can distinguish the advected energy from the currently presented turbulent sensible heat fluxes at the surface. For this comparison, we will use equation 2 from Harder et al. (2017), which will also be referred to in our revised version.*

   *van Heerwaarden, C. C., & Mellado, J. P. (2016). Growth and Decay of a Convective Boundary Layer over a Surface with a Constant Temperature, Journal of the Atmospheric Sciences, 73(5), 2165-2177, https://journals.ametsoc.org/view/journals/atsc/73/5/jas-d-15-0315.1.xml*
* * *
2. The Introduction of the process and its relevance could be extended to allow the readers an easier access to the very complex interplay of near-surface boundary layer processes that become important over patchy snow covers. I think that the manuscript would particularly benefit from a more detailed background (also including "older" studies) on wind-driven heat exchange processes, the development of internal boundary layers (.e.g. Granger et al., 2002; Essery et al., 2006) and the local advection of sensible heat (e.g. Marsh et al., 1999).

   *We will include a more elaborate discussion of the amount of research done regarding the topic. Thank you for your suggestions.*
* * *
3. The connection between the experimental and the numerical part of the manuscript is not totally clear to me. For the experimental part, the study would particularly benefit from a more detailed

analysis on the spatial aspect of the process, i.e. analysis of fetch distance related snow melt and advection estimates. What is the added values of the experimental part?

*We use the field observations to illustrate that the processes can be important and even can be observed with relatively simple and cheap methods on relatively short timescales. Of course, especially the importance has also been shown by previous studies.*

*Also, we try to discuss which processes play a role for the melt we observed in the field with the help of the simulations. Additionally, these simulations show the potential of DNS to be used for studying this kind of system. As disadvantage, these simulations are in an idealized environment and do not include any complex interactions, for example between topography and atmosphere, which probably are playing a role in the field. Therefore, in the discussion we try to uncover which processes are missing in the simulations and how these could affect our understanding of what is going on at the observed snow patch.*

*We do realize that this has not been formulated elaborately enough and will add this to a revised version.*
* * *
4.  Why are such extreme boundary conditions used for the DNS leading to unrealistically high calculated turbulent heat fluxes? In my view, more representative meteorological boundary conditions (i.e. matching up with the conditions at the observed snow path) would provide more meaningful conclusions. Also, Schlögl et al. (2018a) did a similar modeling study using ARPS. Please set your results more in context of this recent study. What are the benefits of using DNS? How do the results compare? What do we learn? How can we represent the process in larger-scale models?

*We used the conditions reported by Harder et al. (2017) for 30 March 2015. Indeed these conditions are relatively extreme, whereas the usage of the meteorological conditions at our observed snow patch would allow for a better comparison. However, due to the absence of accurate local meteorological measurements at the snow patch, we decided to use the data reported by Harder et al. (2017), with the advantage that their system is relatively more similar to an ideal system. In a revised version of the manuscript, we will treat these choices and consequences more elaborately.*

*Additionally, we will include a discussion on how our study relates to Schlögl et al. (2018), also treating the benefits and drawbacks of DNS. Among these are the advantage that DNS does not use the Monin-Obukhov similarity theory, of which the horizontal homogeneity assumption is violated for a patchy snow cover, but also the potential influence of the applied boundary conditions and relatively low Reynolds number on surface fluxes.*
* * *
L134: Why did you not measure the spatial distribution of snow ablation over the entire snow patch? How did you determine the local wind direction? Also, was the wind fetch always constant through the measurement time period?

*Indeed, having a photogrammetry product covering the entire snow patch would be ideal for this study and allow for a more detailed analysis of the snowmelt. However, this would require other equipment than what was available. Still, with the equipment at hand, we try to illustrate that with relative simple*

*and cheap methods, it is possible to come up with relatively decent snowmelt estimates. In a revision, we will explain this more elaborately.*

*The reported values for the wind direction (Table 3) are obtained from the meteorological flux tower. Through experiencing the local wind direction at the field site, we determined that this local wind direction resembled the wind direction at the flux tower. We are aware that these numbers include uncertainty, but still are illustrative for the wind direction at the snow patch. When revising, we will emphasize that the reported wind direction numbers are only an indication, and not necessarily the exact numbers at the field site.*
* * *
L139: why did you not measure SWE for the entire snowpack? Doing so at different sites with different snow depths would allow a more precise information on SWE of the snowpack at the snow patch. Was the snow pack already isothermal at the start of the measurement campaign?

*We are aware that taking these samples only on a single location and only on one day does not reflect the potentially complex spatial (and temporal) dynamics of the snow density and SWE. However, we assume the variations occurring on these spatial and temporal scales to be relatively small compared to other uncertainties introduced to our method for computing contribution estimates of the turbulent heat fluxes to the snowmelt. Moreover, we do think that these densities are realistic estimates and represent a continuously ripe snowpack, given the fact that largest discharge peak had taken place already 1.5 month before the fieldwork (Figure 1) and the air temperature never decreased to freezing point during the campaign (Table 4). Additionally, during the campaign it was noted that the snow pack was relatively wet. In a revised version of the manuscript, we will articulate these considerations.*
* * *
L 161: what do you mean by assuming a snow albedo between 0.6 and 0.8? changed the value in time? Can you provide a reference for choosing those numbers? The albedo value has an extreme effect on your energy balance calculation and your estimated contribution from turbulent heat fluxes.

*We agree that it is not clear how we used these albedos. We will express this more clearly in a revised version. These albedos are both used in the computations, because we don't know the exact albedo of the snow patch, let alone spatial and temporal variations. Moreover, with this range we try to account for other uncertainties we have in the shortwave radiation component. This also is the main cause for the ranges in our eventual estimates.*

*The values are based on Harding (1986), who did measurements in the same region, in approximately the same time of year and reports albedos varying around 0.8 in May. We will add this reference in a revised version.*
* * *
L 318: how do you define up-wind and downwind edge? is it the first grid cell? How do you deal with grid cells which become snow-free during the observation day? The daily-melt rate will be underestimated if you also consider pixels which become snow-free during a measurement day. Would be interesting to see a snow ablation rate curve depending on fetch distance.

*We agree that the text describing the SfM, and especially the post-processing of the DEM and orthoimages can be better formulated. We will include a more elaborate explanation in the revised version of the manuscript.*

*To answer your comments, we have grids for two locations, i.e. the upwind and downwind edge of the same snow patch. So, when referring to either the upwind or downwind edge, we mean the location of the grid (Figure A1).*

*Through the filtering process (which we will state more clearly in the revision), we only consider grid cells that are continuously covered by snow and have a recorded height change on each day, to reduce the chance of cells being random scatter. Indeed, as additional advantage this method does not include cells with relatively shallow snow depths of which the recorded melt could be affected by the presence of the bare ground. Our choice for these filters is supported by the fact that when loosening these filters, the size of the boxplots increases drastically, also to unrealistic values and variations in snow surface height, such as large increases over the course of these 5 days.*

*The resulting height differences over time correspond to 6.7 $m^2$ and 30.7 $m^2$ for respectively the upwind and downwind edge. We are aware that these areas are limited by our filtering choices, especially on the upwind edge due to the varying locations of snow covered grid cells or the retreating snow line (Figure A1). For the downwind edge, the approximately constant location of the snow covered grid cells combined with the little retreat at this edge, causes this area to be significantly larger. Even though these resulting areas are relatively small, we are convinced that the obtained height changes obtained are decent estimates, also based on our error estimates.*

*Unfortunately, as a disadvantage of the size of the upwind area consisting of multiple separate smaller areas, we decided to treat the edge as "point" and not look further into the spatial distribution of the recorded melt (e.g. how is the melt related to fetch distance?). The smaller areas are too far apart to do so.*

\---------------------------

L336 and table 4: Please provide more precise explanation on your estimate ranges. Please also state whether any spatial interpolation is done to the meteorological variables or not.

L343: I assume that you are taking the difference of snow melt due to radiation (equation2) and the actual snow melt to estimate the contribution of the turbulent heat flux. Please add more information how you exactly calculate the turbulent heat flux (latent and sensible turbulent heat flux?)

*Indeed, our explanation on the computations used to come up with our estimate ranges can be clarified. In a revised version, we will include a more precise explanation on these computations.*

*For the meteorological variables, we have not applied any spatial interpolation. We are aware that these number do not exactly represent the local circumstances at the observed snow patch. However, the shortwave radiation is treated with the potential uncertainties and the longwave radiation is assumed to be an appropriate estimate for the larger region. For both, we agree that we have not dealt with all potential uncertainties, which we also try to discuss in Section 4.1. Yet, we will more clearly define these uncertainties in a revised version.*

\---------------------------

L353/354: and how does this compare to the contribution at the downwind edge? As mentioned earlier it would be extremely interesting to have a fetch distance related estimate of the contribution of turbulent heat fluxes (sensible and latent). Also, if you provide a number of 60-80% contribution at the upwind edge – what does this exactly mean? Over which area? As known from other studies, the contribution strongly changes with fetch distance. These high numbers of 60-80% might be very

misleading looking at the relevance for the catchment scale snow melt. It would be very interesting to see an analysis on the contribution of heat advection to total snow melt for varying snow patch sizes and snow cover fractions. Furthermore, the relative contrition of heat advection to total snow melt strongly depends on the spatial variability of snow depths as snowpacks with a high spatial variability of end of season snow depths are typically characterized by a longer time period of the patchy snow cover stage and therefore a higher importance of the heat advection process. A more detailed discussion would allow a better comparison to the study of Schlögl et al., 2018a. Please relate to results of Schlögl et al. (2018), who tried to put the local scale estimations into the catchment scale context to draw conclusions for its relevance.

*As we explained in a previous comment, we treat the observed height change at the both edges as "point" data, due to the small coverage area. Indeed, if we had better coverage of the areas, an analysis of the spatial distribution of the melt would be very interesting and provide insight into the role of the turbulent heat fluxes.*

*Regarding the estimated contribution of the turbulent heat fluxes to the snowmelt at the upwind edge, we will articulate more clearly how this melt does relate to the downwind edge in the revision and also state how these numbers relate to snowmelt on catchment scales. For this perspective, we will also relate to the results of Schlögl et al. (2018).*
* * *
Section 4.1: These estimations include many uncertainties (snow density differences depending on snow height, differences in shortwave radiation between snow patch and actualmeasurement location due to terrain shading, albedo). The high number of turbulent heat fluxes at the surface do not tell us how much of this turbulent heat flux originates from the higher air temperatures at the upwind edge caused by the local advection of sensible heat. Regarding the uncertainty in the net shortwave radiation the authors should consider doing radiation modelling for the area for the respective time period including high-resolution terrain information.

*We agree that there many uncertainties in computing these estimates. We therefore specifically chose a relatively large range in albedo to cover the uncertainties in shortwave radiation, and we include these uncertainties in our subsequently computed melt estimates. So we can still be confident that the numbers hold and support our conclusions.*

*We also agree that performing radiation modelling combined with high-resolution terrain information is relevant for snowmelt runoff simulations. There are planned studies looking specifically into this issue for our study region (cf. e.g. Silantyeva et al., 2020), but it would be out of scope to consider this in the study we present here.*

*Both of these points, we will discuss this more elaborately in a revised version.*

*Regarding the contribution of the higher air temperatures and moisture content at the upwind edge caused by the local advection of turbulent heat, we do assume that the atmosphere has adapted itself to the patchy snow cover and is approaching equilibrium. In all directions and great distances from the observed snow patch, there was a patchy snow cover present. Based on this we do assume that our estimate of the total turbulent heat flux is dominated by the local advection of sensible and latent heat to come up with our estimates. Yet, we are aware that these estimates can be affected by the large scale atmospheric conditions. We will add these considerations to a revised version and also relate this to our revision of Figure 7.*

*Silantyeva, O., Burkhart, J. F., Bhattarai, B. C., Skavhaug, O., and Helset, S.: Operational hydrology in highly steep areas: evaluation of tin-based toolchain, EGU General Assembly 2020, Online, 4–8 May 2020, EGU2020-8172, https://doi.org/10.5194/egusphere-egu2020-8172, 2020*

---

## Author Response (AR1)

**Reply Referee #1**

**Dear Referee,**

**We would like to thank you for taking the time to review our paper and for all your constructive suggestions, which will help to improve the quality of the paper. We reply to your comments below. Our response to the comments appears in bold and revised text as** *italic***.**

Major Comments:

1. Literature context: I get the impression that the authors may be newer to the snow field and so some of the statements in the introduction (see comments) need revising. Broadly there is a tremendous amount of research on processed based- energy balance snowmelt modelling that needs a clearer summary/context for this work. In terms of local-scale advection work there is a much more limited amount but can be found back to the 1970's. This needs a more complete treatment to solidify the context of this contribution as well as to distinguish these contributions from previous one. (See comments for examples).

   **A. Based on your comments and the comments of the other reviewer we have elaborated the introduction. See our replies on your comments on L61-63 L63-64, L83-90**

   **Additionally, in the results and discussion we include a more elaborate discussion on the use of DNS and a comparison with the studies of Schlögl et al. (2018) and Marsh et al., (1999), who studied the role of snow patch size on snowmelt:**

   **Results:**

[revised manuscript text omitted]

**In the following paragraph we stress the advantages of DNS:**

*….Overall, these differences in assumptions between the simulations and the field observations make a one-to-one comparison difficult. Yet, the general behaviour found in the simulations is similar to previous literature, i.e. temperature profiles and melting patterns (e.g. Harder et al., 2017; Mott et al., 2016), and shows the potential of DNS as a modelling tool to understand the melting of a patchy snow cover. Especially, considering that DNS does not violate the assumptions for the Monin-Obukhov bulk formulations and is able to resolve the leading-edge effect, in contrast to modelling studies with coarser spatial resolutions, which could lead to major errors (Schlögl et al., 2017). As such, the simulations are expected to provide a more realistic behaviour of the turbulent heat fluxes.*
* * *
2. Validation with SFM: I'm not exactly clear on the process by which the snow depth difference are calculated via SfM on the observed patch in order to compute expected ranges of turbulent flux contributions and model validation. Are we only considering snow depth difference for grid cells that were completely uncovered during melt (so we can have a bare ground reference) or are we also considering cells that were not fully uncovered that would have also decreased in snow elevation due to melt? What areas do these number represent? Do you observed a decrease in melt from leading edge? Many dynamics can be examined with a spatial dataset but I'm not clear on how this is also processed/what it represents and so would welcome a lot of clarification and perhaps a figure to describe this process.

**A. We agree that this is unclear. As indicated in our preliminary response, we have adapted this. We have changed these lines by including a stepwise explanation on this methodology:**

*Both types of grids are available for each of the 5 days and for both locations, being the upwind and downwind edge, such that we have 20 grids in total. For both locations, the following post-processing is done after obtaining the grids:*

1. *For all grids, remove isolated groups of cells which are smaller than 0.05 $m^2$*
2. *For all grids, apply a median filter of 5 x 5 pixels to diminish the influence of noise located within the areas of interest, but maintain the sharp transitions between snow and snow free surfaces.*
3. *Compute the median height of the bare ground cells per day. The cells selected for this computation should be covered by all grids and already be bare ground on the first day.*
4. *Compute correction heights through comparing the daily median heights of the bare ground with the median height of the first day (step 3).*
5. *Remove bare ground cells out of DEM, based on orthoimages of the same day, such that only snow-covered cells remain for each day.*
6. *Apply correction height (step 4) on snow covered DEM (step 5) for each day.*
7. *For snow covered grid cells that are present on each day (step 5), we calculate height differences between the DEMs of the first day and other days (step 6). We remove absolute height differences larger than 50 cm, as larger values are highly unlikely to occur.*

*The resulting height differences over time correspond to 6.7 m2 and 30.7 m2 for respectively the upwind and downwind edge. We chose to solely use grid cells that are continuously covered by snow and have a recorded height change on each day, to reduce the chance of cells being random scatter. Additionally, this method does not include cells with relatively shallow snow depths of which the recorded melt could be affected by the presence of the bare ground below the snow.*

*We are aware that these filters have an effect on the number of analyzed grid cells and could cause an underestimation of the amount of snowmelt, especially on the upwind edge due to the varying locations of snow covered grid cells and the retreating snow line (Figure A1). For the downwind edge, the approximately constant location of the snow covered grid cells combined with the little retreat at this edge, causes this area to be significantly larger. As a disadvantage of the sizes of the covered areas, we decided to treat the edge as "point" and not consider the spatial distribution of the recorded melt.*

**Also we have adapted Figure A1 to give an overview of the resulting height differences. We have changed this figure by including the eventually used height changes, in accordance with figure 4, and aligning the edges of the patches. The new figure looks like:**

[Figure]

***Figure A1.*** *The resulting height change maps used for Figure 4 plotted over the real-color orthoimages, which are used for distinguishing snow from bare ground, after removing isolated groups of cells and median filtering. The insets show zoomed areas of the upwind edge for more detail.*
* * *
3. Latent heat flux decay: There is an extensive treatment of the sensible heat flux decay with patch length while this is not discussed with respect to latent heat flux. Can this also be included or is there are reason it is not included. The heavily cited Harder 2017 paper suggest that latent heat is also an important contributor or at times compensatory (Harder 2018) and so would be very interested in seeing if some of those dynamics could be captured in this modelling scheme.

**A. We agree that the latent heat flux can be an important contributor to melt. MicroHH does allow to include the latent heat flux (e.g. Bonekamp et al., 2020), however we chose to only consider the sensible heat flux to explore the potential of using DNS for studying this kind of system.**

**We have added some sentences to the manuscript, stating our choices:**

**At the start of section 3.1:**

*We used an idealised system (Fig. 2) to study the turbulent heat fluxes in detail and understand the behaviour observed in the field. As this is one of the first studies using DNS to investigate the role of the turbulent heat fluxes in these systems, we focus on the sensible heat flux, even though MicroHH allows to include the latent heat flux (Bonekamp et al., 2020). Instead of using our own measurements ….*

**At the end of the introduction:**

*This allows us to illustrate the performance of DNS as a tool to understand the real world behaviour and try to explain this with idealized simulations. To do so, we perform these simulations with the Computational Fluid Dynamics (CFD) code MicroHH. We use the measurements of Harder et al. (2017) on a single snow patch as a basis for designing our numerical experiments, and choose to focus on the sensible heat flux. These measurements are done in close to idealized settings on a flat surface. Subsequently, we investigate the influence of enlarging snow patches on the turbulent fluxes into the snow and the implications for snowmelt modelling.*

**Halfway in the discussion:**

*On a catchment scale, these simulation results imply that differences in snowmelt, within a highly idealized catchment, solely occur due to snow patch length. The sensible heat fluxes into the snow at the upwind edge of the patches are independent of snow patch size and show the same decay over distance from the leading edge, such that systems with typically larger patches have on average reduced sensible heat fluxes into the snow. The major cause of these fluxes seems to be the advection of sensible heat. It should be noted that the latent heat flux, which is not considered in these simulations, can also play a significant role for the amount of snowmelt (Harder et al., 2017, 2019). For this flux, we expect similar mechanisms in our simulations based on the observations of Harder et al. (2017). Variations in surface roughness and topography mostly create micrometeorological circumstances…*
* * *
4. Implications: there are some interesting dynamics explained but I'm not exactly clear on how those could be implemented in larger scale snowmelt prediction. There is a scaling relationship articulation for sensible heat over a patch length. Is this considered to be a parameterisation that could be used in basin scale snowmelt prediction models

**A. Indeed, a scaling relationship would make it possible to parametrize these type of processes. However, we are not certain that our relationship is appropriate for implementation, due to some limitations of our methods, such as the use of an idealized system and Reynolds number limitations. Future studies would need to look further into this relationship, especially when more data is available. We have added these considerations:**

*A potential flaw of our simulations is the application of a low $Re_\tau$ compared to the observations of Harder et al. (2017) (i.e. 590 vs $\sim 6 \times 10^6$). This was done based on the results of Moser et al. (1999), who showed large differences between $Re_\tau = 180$ and $Re_\tau = 395$, whereas $Re_\tau = 590$ has similar bulk quantities and variances as the latter simulation. Adjusting the $Re_\tau$ possibly has affected the surface momentum fluxes, of which $U/u_\tau$ is a measure, and thus also the heat fluxes. Furthermore, the low Re has likely caused the fluxes in the IBL to be predominantly diffusive, whereas in reality the turbulent fluxes are more likely to dominate. As the typical diffusive timescales are larger than the typical turbulent timescales, the typical time- and length-scales of the processes in the IBL are relatively large compared to reality, such that one of the processes affected by this, could be the decay of the sensible heat fluxes over distance from the leading edge of a snow patch. To uncover whether the sensible heat fluxes and the decay are dependent of the $Re_\tau$, we recommend to perform simulations with higher $Re_\tau$. Increasing Re, will reduce the scale of the smallest eddies, i.e. the Kolmogorov length scale (Pope, 2000), such that an enhancement of the resolution possibly also is required.* *Next to the influence of using an idealised system, this makes that our formulated relationship between the sensible heat flux at the surface of the snow patches and the distance from the upwind edge (i.e. $H_{sn} \sim x^{-0.35}$) should be taken with caution. Yet, the method does illustrate the use of DNS to come up with potentially useful relationships. Future studies would need to look further into the above described behaviour, especially when more comparable high-resolution data is available.*
* * *
Specific Comments:

Line 39-40: Snowmelt, especially over continuous snow cover, is governed by the surface energy balance (radiation AND turbulent exchange processes) with radiation being the dominant source (Male and Granger, 1981). Turbulent processes, for which air temperature is a proxy, can clearly be important (especially with advection as seen here). Commonly used empirically based temperature index models erroneously lead to the impression that snowmelt is related to air temperature but if we are to be focused on process interactions this statement is problematic.

**A. We agree that this statement is not correct. We combined this comment with the following and have adapted the text as follows:**

*The main processes causing snow to melt are different for a patchy snow cover than a continuous snow cover. For a continuous snow cover,  the surface energy balance, being responsible for snowmelt, is dominated by radiation (e.g. Male and Granger, 1981). For this type of cover, the turbulent heat fluxes are mainly  driven by large-scale air mass movement affecting the ambient air temperature or moisture and wind speed, which could cause these fluxes to be significant during brief periods (Anderson et al., 2010). However, over longer periods, such as weeks, air temperature and moisture gradients near the surface are generally too low to generate significant turbulent heat fluxes (Hock, 2005).*
* * *
Line 43-45: this contradicts the lines 39-43.  Distinction between the scales of advection are needed. (Shook and Gray, 1997)  Large scale air mass movement can drive turbulent exchange because otherwise the temperature and humidity gradients will tend to equilibrium as noted here.

**A. See our reply to your previous comment.**
* * *
Line 53-54:  TI models are empirical so another major criticisms is their applicability when applied outside of their calibration periods or domains – especially in prediction of future changes.  There are many other physically based snowmelt models out there besides Alpine3D (which is based on SNOWPACK), such as snobal (Marks et al., 1998), CROCUS (Vionnet et al., 2012) or in the multitude of land surface schemes.

**A. We agree. We have adapted this in combination with a remark of the other reviewer:**

*However, the performance of these models decrease_s_ significantly with increasing temporal resolution_,_ adding a spatial component to the model, application beyond the period or domain of calibration (e.g. climate change) or during rain-on-snow events  (e.g. Hock, 2003).*
* * *
Line 61. Few models parametrise lateral snow distribution processes fully/explicitly (CHM (Vionnet et al., 2021) and APLINE3D are the only 2 that come to mind that have actual process level physics involved)– most others are often based on simplified parameterisations.

**A. We have rephrased this to put more emphasis on these models:**

* Few models, like Alpine3D and CHM (Vionnet et al., 2021), do include wind-driven processes like snow redistribution and turbulent heat fluxes, but most often parameterize the subgrid turbulent fluxes with the average temperature or moisture at the surface and lowest atmospheric layer per grid cell and do not account for subgrid spatially varying melt fluxes.*
* * *
Line 61-63: (Harder et al., 2018) provides a simple snowmelt advection model to account for subgrid variability in melt.  (Marsh et al., 1999, 1997) provide an approach to account for sensible heat advection.

**A. Based on your comments (and the other reviewer), we have included additional text in this paragraph, which states various approaches of local-scale advection for larger-scale models.**

*….Even when using complex atmospheric snow surface models, such as Alpine3D (Lehning et al., 2006) with ARPS (Xue et al., 2001), local-scale advection associated with subgrid variation of snow and bare ground is excluded. Few models, like Alpine3D and CHM (Vionnet et al., 2021), do include wind-driven processes like snow redistribution and turbulent heat fluxes, but most often parameterize the subgrid turbulent fluxes with the average temperature or moisture at the surface and lowest atmospheric layer per grid cell and do not account for subgrid spatially varying melt fluxes. As potential solutions, Harder et al. (2019) propose a simple model to include local-scale advection of turbulent heat to snowmelt using scaling laws, while Essery et al. (2006) develops a more complex approach based on integration of the energy equation as suggested by Granger et al. (2002) using mixing length theory (Weismann, 1977). Considering the subgrid heterogeneity of the melting fluxes….*
* * *
Line 63-64:  (Harder et al., 2018) provides a advection modelling framework that makes an argument that in some situations upscaling with and including advection will not make any different to discharge predictions.  Ie things can get complicated when the snowmelt is increasing in rate but decreasing in area.

**A. We agree, that this is indeed important to formulate and included a sentence on this in the 2nd paragraph of the introduction, where we elaborated on the process (based on the other review). This paragraph now looks as follows:**

*The main processes causing snow to melt are different for a patchy snow cover than a continuous snow cover. For a continuous snow cover, the surface energy balance, being responsible for snowmelt, is dominated by radiation (e.g. Male and Granger, 1981). For this type of cover, the turbulent heat fluxes are mainly driven by large-scale air mass movement affecting the ambient air temperature or moisture and wind speed, which could cause these fluxes to be significant during brief periods (Anderson et al., 2010). However, over longer periods, such as weeks, air temperature and moisture gradients near the surface are generally too low to generate significant turbulent heat fluxes (Hock, 2005). When the snow cover becomes patchy, the net radiation becomes spatially highly variable, due to variations in surface albedo and emissivity. This spatial variability can act on orders of meters, such that a highly heterogeneous surface arises of relatively warm (and possibly wet) snow free area adjacent to a relatively cold (and drier) snow patch. When a wind blows over this horizontally heterogeneous surface, downwind of the transitions internal boundary layers form, due to changes in the surface conditions (Garratt, 1990, e.g.). The heterogeneity of these internal boundary layers induces a system in which the turbulent heat fluxes can highly vary spatially, partly due to the advection of turbulent heat (Essery et al., 2006; Mott et al., 2013; Harder et al., 2017). These systems are often described as separate growing boundary layers*

*following a power law (Granger et al., 2002, 2006; Harder et al., 2017). For the stable internal boundary layers, i.e. over snow patches, atmospheric decoupling of the air close to the snow surface from the warmer air above can occur, due to large temperature differences between both or cold-air pooling, limiting the exchange of sensible and latent heat from the atmosphere towards the snow (Fujita et al., 2010; Mott et al., 2016). Moreover, the influence of the influence of the turbulent heat fluxes on the total amount of melt increases with decreasing snow cover fractions (Harder et al., 2019; Marsh et al., 1999; Schlögl et al., 2018a).* It has been suggested that this process can be responsible for up to 50% of the total snowmelt (e.g. Harder et al., 2017; Mott et al., 2011). *Additionally, similar processes have been found to potentially significantly contribute to snowmelt on ice fields (e.g. Mott et al., 2019) and glaciers (Sauter and Galos, 2016; Bonekamp et al., 2020; Mott et al., 2020).* This stresses the potentially significant contribution of lateral transport of turbulent heat to a melting patchy snow cover, though it opens the question what the hydrological relevance on larger scales is.
* * *
Line 71-76: missing the advection work of Marsh found in the publications in 1997 and 1999.

**A. We added these papers to the introduction. Marsh et al., 1997 is used in this particular chapter, as it really focusses on observations:**

*Dedicated observations are needed to quantify the importance of turbulent heat advection on snowmelt. Several field measurements focused on the meteorological surface characteristics, such as the development of an internal boundary layer (IBL) as a consequence of the heterogeneous snow cover (Mott et al., 2017) or the influence of topographical depression on cold-air pooling and subsequent snowmelt (Fujita et al., 2010). Other field measurement campaigns estimated the turbulent heat fluxes and the accompanied mechanisms for a single isolated snow patch (e.g. Granger et al., 2006; Harder et al., 2017). Though, all of these experiments focus on relatively brief periods of time with a maximum timespan of a day, during which the conditions for local-scale advection of the turbulent heat fluxes were often ideal. For a small area in complex terrain and longer periods, (Schlögl et al., 2018b) related measured snowmelt to turbulence, while Marsh et al. (1997) estimated the role of advection of turbulent heat by comparing estimated sensible heat fluxes with and without advection. Yet, estimates of a multiple-day contribution of the turbulent heat fluxes to the melt of a single snow patch are lacking, but could provide additional insights in the role of these fluxes on longer timescales*

**Additionally, we refer to the work of Marsh et al., 1999 in the paragraph about modelling approaches and the more elaborate explanation of the process (for the latter see our reply to your previous comment):**

*To eliminate parametrization uncertainties, a relatively new type of simulations, Direct Numerical Simulations (DNS), can be applied to model local-scale advection, as it resolves all the relevant spatial and temporal scales of turbulence. This simulation type has already proven its value in the field of fluid dynamics (Moin and Mahesh, 1998) and allows to extract very detailed information from the turbulent flow, enhancing our process-based understanding of local-scale advection. As a consequence of the high resolutions, these simulations do not need any turbulence parametrizations based on stability corrections and the Monin-Obukhov assumptions, in contrast to numerical atmospheric boundary layer models (Liston, 1995; Marsh et al., 1999) or Large-Eddy*

*simulations (Mott et al., 2015; Sauter and Galos, 2016). These corrections and assumptions are violated for a patchy snow cover and, as such, introduce a large uncertainty (e.g. Mott et al., 2018; Schlögl et al., 2018a). Bonekamp et al. (2020) successfully….*
* * *
Line 80-81: "spatially highly variable character of melt rates can complicate the observations" - > "high spatial variability of melt rates complicate the observations"

**A. Agree, we changed the sentence as follows:**

*To observe the melt of a snow patch over the course of multiple days, high spatial variability of melt rates complicate the observations.*
* * *
Line 83-90: there is a tremendous body of work on snow remote sensing at high resolutions that far exceed the Offenbach and rittger references which are not the most appropriate to consider local scale advection dynamics. Perhaps recast this in terms of remote sensing that is suitable for advection (ie high temporal frequency <=daily, and spatial resolution <<10m). Some high resulting satellite products coming online now but really should focus on aerial platforms (ie ASO (Deems et al., 2013; Painter et al., 2016), drone based (you have many of the UAV-Sfm references but lidar applications are coming online now as well (Harder et al., 2020; Jacobs et al., 2021), and terrestrial laser scanning (Grünewald et al., 2010; Hojatimalekshah et al., 2020), and georectification of oblique time-lapse photography (Härer et al., 2013)

**A. We agree and indeed do think that recasting in tems of advection would be more suitable. We changed the paragraph as follows**

*To observe the melt of a snow patch over the course of multiple days, high spatial variability of melt rates complicates the observations. Point measurements might not represent the region of interest, especially for seasonal snow covers (Sturm and Benson, 2004). This advocates the use of spatial field observations at a high temporal and spatial resolution for patchy snow covers. Ground-based methods fulfilling these requirements are amongst others terrestrial laser scanning (TLS) (Egli et al., 2012; Grünewald et al., 2010; Hojatimalekshah et al., 2021; Schlögl et al., 2018) or georectification of oblique time-lapse photography (Härer et al., 2013). Additionally, aerial platforms, such as aerial laser scanning (ALS), either manned (e.g. Deems et al., 2013; Painter et al., 2016) or unmanned (e.g. Harder et al., 2020; Jacobs et al., 2021) can be used. For both positions, application of Structure-from-Motion (SfM) photogrammetry is possible as a relatively cheap method to monitor snowmelt, ofwhich the usage has already been explored in the 1960s (e.g Brandenberger, 1959; Hamilton, 1965), but which was sidelined due to technical constraints. Recently, due to the technical development increasing the accuracy with lower computational costs, the SfM photogrammetry has been used to successfully study seasonal snow covers with low-cost imagery equipment and software for analysis (e.g. Bühler et al., 2016; Filhol et al., 2019; Girod et al., 2017; Nolan et al., 2015). Therefore, this method offers a promising way to monitor snowmelt with low costs and reasonable accuracy.*
* * *
Line 129: of -> from

**A. Agree, changed as follows:**

*The meteorological data is retrieved from a meteorological flux tower at a temporal resolution of 30 minutes and includes, amongst others, temperature, precipitation, wind speed and direction, incoming radiation and relative humidity.*
* * *
Line 131-133: wind direction was constant you state. Can you provide a wind rose or some sort of metric to quantify this?

**A. We provide the wind direction in Table 4. We now added a reference to this table (and adapted the sentence based on suggestions from the other reviewer):**

*From this snow patch (Fig. 1), daily height maps are obtained through photogrammetry over the course of 11 June until 15 June 2019 for the upwind and downwind edge of the snow patch. As the dominating wind direction at the snow patch resembled the measured wind direction at the meteorological tower, which was constant throughout the field campaign (Table 4 in Sect. 4), we assume that the upwind and downwind location of the snow patch are approximately constant.*
* * *
Line 138-140: when were these samples taken with respect to the observation interval as snow density is dynamic over melt? Was a snow tube used? Snow pit? How were 100ml samples collected? Did the melt period have a consistently ripe snowpack?

**A. We have added information on the snowcover and the sampling and included our considerations. This paragraph has have been adapted as follows:**

*… The length of the snow patch is approximately 50 m with a maximum snow depth in the regions of interest estimated to be in the order of 0.5 m. Selection of the location was based on the following criteria: a relatively flat surface and the absence of complex topographical features nearby, which could complicate the incoming radiation by, for instance, partial shading. The height maps enable us to derive the amount of snowmelt during these 5 days for this single snow patch and assess the role of local-scale advection. To do so, the meteorological data is averaged over the period between the photogrammetry observations and compared with the daily melt observations. Also, three snow samples are taken to determine the snow water equivalent (SWE), such that the measured height changes can be converted to a volume. The samples are taken by digging a small snow pit and collecting 100 mL samples at 5, 25 and 45 cm below the snow surface on 14 June. The snowpit is dug in snow adjacent to the snow patch, such that the measurements are similar to the snow patch, but do not affect the photogrammetry observations. We are aware that taking these samples only on a single day and location does not reflect the potentially complex temporal and spatial dynamics of the snow density. However, we assume the variations occurring on these scales to be relatively small compared to other uncertainties introduced by our analysis for computing contribution estimates of the turbulent heat fluxes to the snowmelt.*

**Also we added a note on the assumption of the ripe snowpack 3 paragraphs later (line ….):**

*To determine the snowmelt based on the net radiation, the measured incoming radiation is combined with estimates of the outgoing radiation based with the help of common snow characteristics. The outgoing shortwave radiation is calculated by assuming a snow albedo between 0.6 and 0.8. These values are based on Harding (1986), who reports albedos of approximately 0.8 for the same region in May. We use both of these values in the following calculations to account for the uncertainty in the albedo, due to spatial and temporal variations, and other potential uncertainties in the calculations. Furthermore, we assume that the snowpack is continuously ripe throughout research period, given that air temperature was continuously above freezing point and the largest discharge peak had already taken place 1.5 months before the observation period.*
* * *
Line 153: desolated - > isolated?

**A. Agree, we indeed meant isolated, we have adapted desolated to isolated in the manuscript**
* * *
Line 153-159:  not exactly clear on this methodology.  Base on this and images in Figure A1 we are only looking at the edges and measuring surface change for where the snow melted and a bare ground surface appeared?  Related to figure A1 – how do the upwind and downwind edges line up – unclear as they are plotted in separate rows?

**A. See our answer to your major comment on the validation with SFM**
* * *
Line 160-165:  How deep was the snowpack and do you have any information to say that the snowpack was ripe at the start of the melt.  Were the cold content requirements satisfied at the start of the period and so all energy could be assumed to be related to melt.

**A. See our answer to your comment on lines 138-140**
* * *
Equation 2: I believe SWE should instead be snow density?

**A. Indeed. The new equation is:**

*The net radiation is subsequently recalculated to the melt of the snow expressed as height change, by combining this with the snow density and constant for latent heat of fusion, which is 334 KJ kg$^{-1}$. This can be recalculated, such that eventually the radiation-driven melt is expressed as height change:*

$$M_R = \frac{R_{net} * \Delta t}{\rho_{sn} * L_f}$$

*in which $M_R$ (m) is the snowmelt due to radiation expressed as height change, $\Delta t$ (s) the time between the photogrammetry observations,* $\rho_{sn}$ *(kg m$^{-3}$) the snow density and $L_f$ (J kg$^{-1}$) the constant for latent heat of fusion.*
* * *
Figure 3: how were the surface temperatures and gradients between snow and non-snow generated? I may be blind but can't seem to see this.

**A. We agree that this is not well explained in the main text. We have now adapted the paragraph as follows:**

*Initially, the turbulent channel flow from Moser et al. (1999) used as spin-up is simulated until 1800 seconds, such that the turbulent channel flow has well developed. Subsequently for each simulation, which take 900 seconds, this turbulent channel flow is adapted, such that an atmospheric flow over a patchy snow surface is obtained. On the bottom boundary, a pattern of surface buoyancies depending on the simulation is prescribed, such that the surface characteristics determined during the dimensional analysis are fulfilled. For snow and bare ground, the surface buoyancy is respectively $-8 \times 10^{-3}$ m s$^{-2}$ and 0.0 m s$^{-2}$ (i.e. 273 K and 280.9 K in reality), which is elaborated on in Sect. 3.2. The implemented surface for the P15m and P15m-NB (P30m, P60m) contains snow patches of on average 0.15 m (0.30 m, 0.60 m) and an average element length of 0.30m (0.60 m, 1.20 m) (Table 2; multiply with 100 in Fig. 3). These surfaces enable the model to solve the periodic boundary conditions, because we ensured that the patches at the opposing walls fit together and flow that leaves the system on one side, continues over the same snow patch when it re-enters the system on the opposite wall. These patterns are created by generating noise in the Fourier space around specific wavelengths, prescribed in the form of 2D power spectra. When transforming these back to physical space using the Inverse Fast Fourier Transform, a 2D field with dominant patterns is obtained. For a more elaborate explanation on generation see Appendix B.*
* * *
Figure 4: I'm not exactly clear on what area constituted an upwind or downwind edge and how that relates to a specific number/boxplot. There will be a gradient of change. Can this be clarified? This approach does not consider height changes if it that spot does not become snow free by June 15?

**A. See our answer to your major comment on the validation with SFM**
* * *
Line 334-336: It seems SWE and density are being used interchangeably here which is not correct. Can this be cleared up? These are pretty high densities. Any observations from field notes about water saturation or other structural attributes. What was the overall snow depth variability? Can you report the SWE of the snow patch?

**A. Indeed the interchangeable use of SWE and density is incorrect. Moreover, we added some considerations on the magnitude of the observed snow densities. This has been adapted as follows:**

*Relating the meteorological circumstances (Table 4) to the measured snowmelt, allows to estimate the contribution of the turbulent heat fluxes to the total amount of snowmelt. To do so, the snow density measurements are used. The* snow densities *at 5, 25 and 45 cm below the snow surface are respectively 556 kg m⁻³, 551 kg m⁻³ and 610 kg m⁻³. Following from these densities, the snow density is approximately constant near the surface, whereas further down the snow is more compressed or stores water. Even though the densities are relatively high, we consider the values to be realistic, based on the largest discharge peak taking place 1.5 months before the observation period (Figure 1) combined with the notice that the snowpack was relatively wet while being in the field.*
* * *
Line 353-354:  60-80% based on computing the overall melt energy needed and radiation melt and the turbulent portion as the residual of this energy balance?  If so can that be clarified?

**A. Indeed these percentages are based on comparing the overall melt with the radiation driven melt and taking the residual as turbulent driven melt. We changed this as follows:**

*Overall,* by comparing the overall melt with the melt driven by the radiation and taking the residual as turbulence-driven melt, *we estimate the contribution of the turbulent heat fluxes to the snowmelt to be roughly 60 to 80 \% for the upwind edge of the snow patch.*
* * *
Line 398-403:  Granger et al., 2002 and Weisman, 1977 propose similar power law relationship to describe a sensible heat flux.  Perhaps worth contrasting this formulation and the meaning of your terms with those papers?

**A. We agree that this would be a useful comparison. Based on comments of the other reviewer, we have adapted our figures and now present the advected energy at the end of the results section. Here we discuss our advection estimates. We also compare our found decay with the proposed relation by Granger et al. (2002). This looks as follows:**

*Compared to Harder et al. (2017), our estimates of the advected sensible energy are relatively high. The measurements done by Harder et al. (2017) show values slightly above 400 W m⁻2 for the first 3.6 m (Figure 7). In our simulations the advection of sensible heat decreases in the first 3.6 m from 577 W m⁻2 to approximately 400 W m⁻2. For the following 4.8 m, Harder et al. (2017) reported an average reduction in advected sensible heat to approximately 20% of values found for the first 3.6 m. In the comparable simulation with a similar dominant snow patch pattern, this reduction is not found. Yet, further downwind the advected sensible heat does reduce to values in the same order of magnitude. It should be noted that, due to setting the integration height at 2 m in Eq. 17, not all changes in the vertical temperature profile over distance from the leading edge are included, such that the advected sensible heat is underestimated. When considering a 4 m integration height, the advected sensible energy is approximately equal to the sensible turbulent heat fluxes at the snow surface, especially for the first half of the patch,* implying also a power of -0.35 for the decay of the advected sensible heat. This also illustrates the major contribution of the advected sensible heat to the sensible heat

*flux into the snow. We expect a similar role of the advected heat in our field observations, as in all directions and great distances from the observed snow patch, there was a patchy snow cover present, causing an approximate equilibrium condition. In comparison to a similar relationship obtained by Granger et al. (2002) (based on Weismann, 1977) through boundary-layer integration, our advected flux decays less. Granger et al. (2002) come up with a similar power law, but their powers maximally reach -0.47, depending on surface temperatures of snow and bare ground, friction velocity and roughness length.*

**In the discussion we formulate the limitations of our relationship. See also our reply to your major comment on the implications.**
* * *
Section 5.3: this section exclusively discusses sensible heat flux only. Your model also considers latent heat flux and observation are available from Harder et al 2017. Can this also be considered or is there are particular reason you did not bring latent heat flux into the results here?

**A. See our answer to your major comment regarding the latent heat flux.**
* * *
Line 459-461: same order of magnitude regardless of patch size on the upwind edge? Sentence seems not complete.

**A. This sentence can indeed be more clearly formulated. We have changed it:**

*The sensible heat fluxes into the snow at the upwind edge of the patches are independent of snow patch size and show the same decay over distance from the leading edge, such that systems with typically larger patches have on average reduced sensible heat fluxes into the snow.*
* * *
Line 462: "microclimates" - > micrometeorological? Would suggest that these are very dynamics occurrences unlike what is captured with the "climate" term.

**A. We agree with you. We have adapted this term:**

*Variations in surface roughness and topography mostly create micrometeorological circumstances which differ substantially from the average circumstances within a catchment*
* * *
Line 464-467: terrain absolutely plays a role with snow distributions but this is a rather simplistic explanation for very complex physical processes underpinning blowing snow redistribution. Topography, meteorology, surface characteristics all conspire to make any domain very complex in terms of snowpack distribution variability. I'd step this back and say that snow patch size distributions (if available) would improve snowmelt

predictions. There are many tools and statistical descriptions of snowpack's available to do so (see snow pack scaling laws in Harder et al., 2018 and the papers cited therein that consider fractal geometry for example).

**A. We agree that this is rather simplistically explained. Therefore, we have followed your suggestion and adapted our text as follows:**

*Overall, our results imply that the performance of snowmelt predictions would improve when also considering snow patch size distributions. Information on and usage of these distributions can be obtained with various methods, ranging from relatively simple methods, for example scaling laws (e.g. Harder et al., 2019), to more complex methods, for example by assimilating various satellite retrievals (e.g Aalstad et al., 2018).*
* * *
Line 467-468: "The melt estimates obtained with the SfM photogrammetry are in line with own expectations based on visual estimations, whereas the estimated errors are relatively small." Can you clarify what you mean with "in line with own expectations" - meaning is not apparent to me.

**A. We meant that based on our experience during the field campaign, we estimated the snow to melt in the same order of magnitude as obtained with the SfM photogrammetry.**

The melt estimates obtained with the SfM photogrammetry are in line with our expectations based on rough visual estimates from during the field campaign, whereas the estimated errors are relatively small.
* * *
Line 480-490: have you run any simulations with a higher Re in line with Harder 2017 so that you could make some more conclusive predictions of this RE- decay relationship?

**A. Unfortunately, we have not run any simulation with a higher Re. Based on the comments of the other reviewer, we added a more elaborate discussion on the advantages and drawbacks of DNS. Among these is the mentioning of the Reynolds number. See our reply to your first major comment.**
* * *
Line 664-665: please change to the non-discussion version of this paper

**A. Agree, using the discussion is incorrect.**

*Nolan, M., Larsen, C., and Sturm, M.: Mapping snow depth from manned aircraft on landscape scales at centimeter resolution using structure-from-motion photogrammetry, The Cryosphere, 9, 1445–1463, 2015.*

**Reply Referee #2**

**Dear Referee,**

**We would like to thank you for taking the time to review our paper and for all your constructive suggestions, which will help to improve the quality of the paper. We reply to your comments below. Our response to the comments appears in bold and revised text as** *italic***.**

General comment:

1. The authors describe a process they term advection of turbulent heat flux and reference studies discussing local advection of sensible heat as described in Mott et al. (2018) and also Harder et al., (2017). It is not clear to me to which term the authors are really relating to as it seems to me that they mix up advection of sensible heat with the vertical turbulent sensible heat flux. The ambiguity becomes particularly clear when the authors compare modelled sensible heat fluxes with estimated advected sensible heat as presented in Harder et al (2017). I recommend to include equations where they clearly state at which terms they are looking at and how these are calculated. Equations for advection of sensible heat are presented in Harder et al. (2017) and Mott et al. (2020).

**We have more clearly stated which terms we are looking at and also mentioned how these terms are calculated.**

**Last paragraph introduction:** *Whereas previous studies pioneered in examining the lateral exchange of turbulent heat fluxes most often for a single snow patch, it remains in question how this process is affected by the typical spatial and temporal scales within a catchment. Moreover, new modelling attempts need to be undertaken to increase our understanding of the wind-driven processes occurring near the surface of melting patchy snow covers. Therefore in this study, we aim to assess the role of lateral transport of the turbulent heat fluxes on snowmelt for a snow patch in a real world case and idealized environment. In the real world case, we will identify the role of* *the turbulent heat fluxes* *for a snow patch in the Finseelvi catchment with SfM photogrammetry observations to study the importance of this process over the course of multiple days. The resulting snowmelt is compared to local meteorological measurements to put the snowmelt in perspective and extract the role of the turbulent fluxes on this melt. Subsequently, we try to uncover the behaviour of* *the turbulent heat fluxes and* *local-scale advection of sensible heat in an idealized environment with DNS, allowing to extract detailed information on wind blowing over a small flat domain with a patchy snow cover. This allows us to illustrate the performance of DNS as a tool to understand the real world behaviour and try to explain this with idealized simulations. To do so, we perform these simulations with the CDF-code MicroHH. We use the measurements of Harder et al. (2017) on a single snow patch as a basis for designing our numerical experiments. These measurements are done in close to idealized settings on a flat surface. Subsequently, we investigate the influence of enlarging snow patches* *on the turbulent fluxes into the snow* *and the implications for snowmelt modelling.*

**Section 3.4:** *In this study, the model simulations are performed using the MicroHH 2.0 code, which is primarily made for DNS of atmospheric flows over complex surfaces by van Heerwaarden et al. (2017). When solving the conservation equations for mass, momentum and energy, MicroHH makes use of the Boussinesq approximation,* *such that*

*the evolution of the system for velocity vector $u_i$ , buoyancy $b$ and volume is described by*

$$\frac{\partial u_i}{\partial t} + \frac{\partial u_j u_i}{\partial x_j} = -\frac{\partial \pi}{\partial x_i} + \delta_{i3} b + \nu \frac{\partial^2 u_i}{\partial x_j^2},$$

$$\frac{\partial b}{\partial t} + \frac{\partial u_j b}{\partial x_j} = \kappa \frac{\partial^2 u_i}{\partial x_j^2},$$

$$\frac{\partial u_j}{\partial x_j} = 0,$$

*in which π is a modified pressure (van Heerwaarden and Mellado, 2016). Moreover, MicroHH uses periodic boundary conditions in the horizontal directions, which implies that we simulate a wind blowing over an infinite snow field. In the results, when reporting sensible heat fluxes, these are recalculted from the surface buoyancy flux B ($m^2$ $s^{-3}$) computed in the model equation according to*

$$B = -\kappa \frac{\partial b}{\partial z}\Big|_{surface},$$

*which can be recalculated to realistic sensible heat fluxes following the steps in Sect. 3.2. The advected energy is computed similar to Harder et al. (2017) (Eq. 2), being*

$$H_{adv} = \int_{z=0\,m}^{z=2\,m} \rho C_p \bar{u} \frac{\partial \theta}{\partial x} dz,$$

*in which z (m) is the elevation above the surface, ρ (kg $m^{-3}$) the air density, $C_p$ (1005 J $kg^{-1}$ $K^{-1}$) the specific heat capacity of air, x (m) the distance from the leading edge of the snow patch. Next to integrating over 2 m profile height, we also integrate over 4 meter height.*

**For our comparison with Harder et al. (2017), we have revised text, and figures 7 and 8. In the results we have changed the following:**

- **1$^{st}$ paragraph section 5.1: …..***These fluxes at the leading edge of the snow patches are relatively high, mostly due to the ideal circumstances for wind-driven melt, including local-scale advection of sensible heat (i.e. high wind speeds and relatively large temperature differences),* *. Compared to our own observations, the sensible heat fluxes at the leading edge are much larger than the observed turbulent fluxes, which is in line with our expectations because of the inclusion of not only ideal circumstances for local-scale advection.*
- **We have removed the results of Harder et al. (2017) from figure 7 and added these to figure 8, where now also provide the advected sensible heat for that specific snow patch. The revisions are as follows:**
    **Fig 7:**

[Figure]

**Figure 7. Time-averaged surface sensible heat fluxes over single patches of snow .** *The surface sensible heat fluxes for individual patches with snow as a function of distance from the leading edge on a log-log scale at y = 1.1 m. The dashed black line is an implemented trendline to show the approximate power law decay of the fluxes over distance. The fluxes of the snow are multiplied with -1, such that these values can also be plotted on logarithmic axes.*

**Fig 8:**
[Figure]

[Figure]

**Figure 8. Time-averaged vertical temperature (a) and wind profiles (b), and advected sensible energy $H_{adv}$ (at 2 and 4 meter integration height) and sensible heat flux $H_{sn}$ at the surface (c) along the snow patch at y = 83 m and x = 150 m in the P30m simulation.** *The markers in the temperature profile resemble the IBL height, which is computed as the first value (seen from the bottom) of the gradient of the temperature over height below 0.1 K m$^{-1}$. The first 2 upwind grid cells and 2 downwind grid cells have been removed, as these are located in the transitions region from bare ground to snow and vice versa. The labels with 'x =' show the location of vertical profile related to the downwind distance of the leading edge, whereas the line colors are based on the distance from the leading edge and go from purple to red. In the inset graph, the vertical temperature profile has been normalised with height of the IBL and prescribed temperature of the surface and atmosphere. The advected*

> *energy fluxes as a function of distance from the leading edge are calculated based on Eq. 17.*

- **We also have adapted the text in section 5.3. For the 2nd and 3rd paragraph, we have adapted the text as follows:**

   *Our simulated sensible heat fluxes are approximately 500 W m$^{-2}$ at the leading edge* and at the upwind edge and 200-300 W m-2 at the downwind edge. In comparison to our field observations, at both edges these sensible heat fluxes are relatively high.

  ~~and reduce to approximately 450 W m−2 during the first 3.6 m. These large sensible heat fluxes are due to the highly ideal conditions for local-scale advection of sensible heat, being high wind speed and large temperature differences between the atmosphere and snow. For the following 4.8 m, Harder et al. (2017) reported an average reduction in sensible heat flux to approximately 20% of values found for the first 3.6 m. In the comparable simulation with a similar dominant snow patch pattern, this reduction is not found. The sensible heat flux above the snow does never reduce to these fractions, but only halves compared to the maximum values for the surface sensible heat flux.~~

  *At the upwind edge, the simulated sensible heat fluxes are approximately 5 times larger than the derived total contribution of the turbulent heat fluxes to the measured snowmelt. We reckon that the simulated values are large, though it should be noted that the simulations are based on highly ideal conditions for local-scale advection of the turbulent fluxes, whereas the conditions during the measurements were not ideal (e.g. nighttime melt is included). At the downwind edge, the measurements suggest an approximately negligible contribution of the turbulent heat fluxes to the snowmelt, whereas the simulations show at comparable snow patches a significant contribution of the sensible heat flux ($\sim$ 200-300 W m$^{-2}$ ). Thus, the simulated decay of the sensible heat flux seems to be an underestimation in comparison to field observations.*

- **We have added a new final paragraph describing the new figure 8, which also relates our results to Harder et al. (2017):**

  *Compared to Harder et al. (2017), our estimates of the advected sensible energy are relatively high. The measurements done by Harder et al. (2017) show values slightly above 400 W m$^{-2}$ for the first 3.6 m (Figure 7). In our simulations the advection of sensible heat decreases in the first 3.6 m from 577 W m$^{-2}$ to approximately 400 W m$^{-2}$. For the following 4.8 m, Harder et al. (2017) reported an average reduction in advected sensible heat to approximately 20% of values found for the first 3.6 m. In the comparable simulation with a similar dominant snow patch pattern, this reduction is not found. Yet, further downwind the advected sensible heat does reduce to values in the same order of magnitude. It should be noted that, due to setting the integration height at 2 m in Equation 17, not all changes in the vertical temperature profile over distance from the leading edge are included, such that the advected sensible heat is underestimated. When considering a 4 m integration height, the advected sensible energy is approximately equal to the sensible turbulent heat fluxes at the snow surface,*

*especially for the first half of the patch, implying also a power of -0.35 for the decay of the advected sensible heat. This also illustrates the major contribution of the advected sensible heat to the sensible heat flux into the snow. We expect a similar role of the advected heat in our field observations, as in all directions and great distances from the observed snow patch, there was a patchy snow cover present, causing approximate equilibrium conditions. In comparison to a similar relationship obtained by Granger et al. (2002) (based on Weismann, 1977) through boundary-layer integration, our advected flux decays less. Granger et al. (2002) come up with a similar power law, but their powers maximally reach -0.47, depending on surface temperatures of snow and bare ground, friction velocity and roughness length.*
* * *
2. The Introduction of the process and its relevance could be extended to allow the readers an easier access to the very complex interplay of near-surface boundary layer processes that become important over patchy snow covers. I think that the manuscript would particularly benefit from a more detailed background (also including "older" studies) on wind-driven heat exchange processes, the development of internal boundary layers (.e.g. Granger et al., 2002; Essery et al., 2006) and the local advection of sensible heat (e.g. Marsh et al., 1999).

**A. We have included a more elaborate introduction of the process. The 2nd paragraph of the introduction has been changed as follows:**

*The main processes causing snow to melt are different for a patchy snow cover than a continuous snow cover. For a continuous snow cover, the surface energy balance, being responsible for snowmelt, is dominated by radiation (e.g. Male and Granger, 1981). For this type of cover, the turbulent heat fluxes are mainly driven by large-scale air mass movement affecting the ambient air temperature or moisture and wind speed, which could cause these fluxes to be significant during brief periods (Anderson et al., 2010). However, over longer periods, such as weeks, air temperature and moisture gradients near the surface are generally too low to generate significant turbulent heat fluxes (Hock, 2005). When the snow cover becomes patchy, the net radiation becomes spatially highly variable, due to variations in surface albedo and emissivity. This spatial variability can act on orders of meters, such that a highly heterogeneous surface arises of relatively warm (and possibly wet) snow free area adjacent to a relatively cold (and drier) snow patch. When a wind blows over this horizontally heterogeneous surface, downwind of the transitions internal boundary layers form, due to changes in the surface conditions (Garratt, 1990, e.g.). The heterogeneity of these internal boundary layers induces a system in which the turbulent heat fluxes can highly vary spatially, partly due to the advection of turbulent heat (Essery et al., 2006; Mott et al., 2013; Harder et al., 2017). These systems are often described as separate growing boundary layers following a power law (Granger et al., 2002, 2006; Harder et al., 2017). For the stable internal boundary layers, i.e. over snow patches, atmospheric decoupling of the air close to the snow surface from the warmer air above can occur, due to large temperature differences between both or cold-air pooling, limiting the exchange of sensible and latent heat from the atmosphere towards the snow (Fujita et al., 2010; Mott et al., 2016). Moreover, the influence of the influence of the turbulent heat fluxes on the total amount of melt increases with decreasing snow cover fractions (Harder et al., 2019; Marsh*

*et al., 1999; Schlögl et al., 2018a). It has been suggested that this process can be responsible for up to 50% of the total snowmelt (e.g. Harder et al., 2017; Mott et al., 2011). Additionally, similar processes have been found to potentially significantly contribute to snowmelt on ice fields (e.g. Mott et al., 2019) and glaciers (Sauter and Galos, 2016; Bonekamp et al., 2020; Mott et al., 2020). This stresses the potentially significant contribution of lateral transport of turbulent heat to a melting patchy snow cover, though it opens the question what the hydrological relevance on larger scales is.*

**Additionally, we have added text in the paragraph about larger scale models:**

*….Even when using complex atmospheric snow surface models, such as Alpine3D (Lehning et al., 2006) with ARPS (Xue et al., 2001), local-scale advection associated with subgrid variation of snow and bare ground is excluded. Few models, like Alpine3D and CHM (Vionnet et al., 2021), do include wind-driven processes like snow redistribution and turbulent heat fluxes, but most often parameterize the subgrid turbulent fluxes with the average temperature or moisture at the surface and lowest atmospheric layer per grid cell and do not account for subgrid spatially varying melt fluxes. As potential solutions, Harder et al. (2019) propose a simple model to include local-scale advection of turbulent heat to snowmelt using scaling laws, while Essery et al. (2006) develops a more complex approach based on integration of the energy equation as suggested by Granger et al. (2002) using mixing length theory (Weismann, 1977). Considering the subgrid heterogeneity of the melting fluxes….*

**In the paragraph about spatial observations we have provided a better overview of various observation methods, based on comments of both reviewers. See our reply on your comment on L83**
* * *
3.  The connection between the experimental and the numerical part of the manuscript is not totally clear to me. For the experimental part, the study would particularly benefit from a more detailed analysis on the spatial aspect of the process, i.e. analysis of fetch distance related snow melt and advection estimates. What is the added values of the experimental part?

**A. We use the field observations to illustrate that the processes is common and important (which of course has also been proven by previous studies) and can even be observed with relatively simple and cheap methods on relatively short timescales. We expressed this in the discussion and conclusion:**

*The melt estimates obtained with the SfM photogrammetry are in line with our expectations based on rough visual estimates from during the field campaign, whereas the estimated errors are relatively small. The errors are in the same order of magnitude as found for high-accuracy snow depth estimates obtained from time-lapse photography (e.g. Dong and Menzel, 2017; Garvelmann et al., 2013), though, it should be noted that these studies focus on somewhat larger areas. Overall, this illustrates that the influence of the turbulent heat fluxes on*

*melting snow patches is widespread and can even be observed with relatively simple and cheap methods. One of the potential limitations to this study….*

**Conclusions:**

*In this study, we examined the melt of a 50 meter long snow patch in the Finseelvi catchment, Norway, and investigated the observed melt with highly idealized simulations. The melt estimates, obtained with relatively simple and cheap structure-from-motion photogrammetry, for the upwind and downwind edge of the snow patch are feasible and the estimated errors are in line with previous studies…*

**Also, we added emphasis in the results and discussion on which processes play a role for the melt we observed in the field with the help of the simulations. In the results (section 5.3):**

*Our simulated sensible heat fluxes are approximately 500 W m−2 and at the upwind edge and 200-300 W m−2 at the downwind edge. In comparison to our field observations, at both edges these sensible heat fluxes are relatively high. At the upwind edge, the simulated sensible heat fluxes are approximately 5 times larger than the derived total contribution of the turbulent heat fluxes to the measured snowmelt. We reckon that the simulated values are large, though it should be noted that the simulations are based on highly ideal conditions for local-scale advection of the turbulent heat, whereas the conditions during the measurements were not ideal (e.g. nighttime melt is included). At the downwind edge, the measurements suggest an approximately negligible contribution of the turbulent heat fluxes to the snowmelt, whereas the simulations show at comparable snow patches a significant contribution of the sensible heat flux (~200-300 W m−2). Thus, the simulated decay of the sensible heat flux seems to be an underestimation in comparison to field observations. We expect that the comparable behaviour of the sensible heat fluxes between patches found within the idealized system, also is occurring within the Finseelvi catchment for patches within similar local conditions.*

**Also we formulated a comparison to our observations in section 5.3 when discussing the new figure 8 (for full explanation see our reply to your 1st comment):**

[revised manuscript text omitted]

**A comparison with for example average temperature, would not be valid, as we prescribe our conditions and do not have any boundary conditions that add heat (e.g. through radiation). As the fluxes at the snow patches are compensated by the bare ground fluxes, all of our average air temperatures are approximately the same. We formulate this in the discussion:**

*Though, the simulations lack the surface roughness differences for the snow and bare ground and topographical variations. Including the transition from a rough (bare ground) to smooth (snow) surface, likely would diminish the IBL growth due to increased turbulence levels and enhance the sensible heat fluxes as a consequence of the larger temperature gradients in the IBL (Garratt, 1990). It should be noted that this mostly holds for shear dominated turbulence, whereas at lower wind speeds the influence of thermal turbulence on the IBL should also be considered. Moreover, the common formation of snow patches in topographical depressions, which does not hold for Harder et al. (2017), but does to some extent for our field observations, causes atmospheric decoupling and reduced turbulent heat fluxes at low and moderate wind speeds, especially downstream of the upwind edge (Mott et al., 2016). Also, the choice of basing the numerical experiments on Harder et al. (2017) instead of our own observations, complicates a one-on-one comparison. Lastly, the exclusion of external forcings, such as radiation or large-scale advection, causes the simulations to approach equilibrium, due to the compensation between the sensible heat fluxes at the snow patches and the bare ground. This is advantageously for investigating the behaviour of the system in relation to snow patch size, but makes it more difficult to compare other characteristics (e.g. temperature) with for example Schlögl et al. (2018a), who did include external forcings, such as incoming radiation. Overall, these mechanisms summarize the largest potential flaws in direct comparison between the simulations and observation*

**Regarding the larger-scale models: a scaling relationship would make it possible to parametrize these type of processes. However, we are not certain that our relationship is appropriate for implementation, due to some limitations of our methods, such as the use of an idealized system and Reynolds number limitations. Future studies would need to look further into this relationship, especially when more data is available. We have added these considerations in the following paragraph. Also we added a sentence on the drawback of DNS, being the computational cost:**

[revised manuscript text omitted]

L 39: Warm air advection is AN important source for the energy balance but not generally the main cause of snow melt. Please change the sentence accordingly.

**A. We agree that this statement is not correct. Based on your comments, and the other review, we have adapted the text as follows:**

*The main processes causing snow to melt are different for a patchy snow cover than a continuous snow cover. For a continuous snow cover,  the surface energy balance, being responsible for snowmelt, is dominated by radiation (e.g. Male and Granger, 1981). For this type of cover, the turbulent heat fluxes are mainly  driven by large-scale air mass movement affecting the ambient air temperature or moisture and wind speed, which could cause these fluxes to be significant during brief periods (Anderson et al., 2010). However, over longer periods, such as weeks, air temperature and moisture gradients near the surface are generally too low to generate significant turbulent heat fluxes (Hock, 2005)….*
* * *
L 46-50: While the introduction to the process is correct, the sentence in L46/47 does not describe the process that is typically described as local advection of sensible heat (as by Mott et., 2018; Harder et al., 2017): Local heat advection is generally understood as a process where the mean wind that transports the warm air from snow-free towards snow-covered area. It is NOT the horizontal gradient in air temperature which initiates this process like sea breezes. The process is defined in Mott et al. (2020) as "Horizontal transport of sensible (and latent) heat with the mean flow" and can be written as wind speed*dT/dx (For further details also see Mott, R., Stiperski, I., and Nicholson, L.: Spatio-temporal flow variations driving heat exchange processes at a mountain glacier, The Cryosphere, 14, 4699–4718, https://doi.org/10.5194/tc-14-4699-2020, 2020.). The process as described in studies of Mott el. (2015,2018, 2020) and Harder et al. (2017) not refer to the advection of turbulent heat fluxes which would be defined as: U * d/dx(w'T') + U*d/dx(w'q'). As you refer to advection of turbulent heat fluxes throughout the manuscript, it is very important that you include equations on the terms you are analysing.

**A. We agree that this statement implies that the horizontal gradients initiates the process, which is not correct. We have adapted the statement in line 46-47 and combined this with your major comment on the introduction. For this see our reply, to that comment.**

**A. Also, we have included equations stating which terms we analyse. For this see, our reply to your first major comment.**
* * *
The process of advection of heat was found to be not only an important process over snow patches but also for ice fields (Mott et al., 2019) and glaciers (Sauter and Galos, 2016; Mott et al., 2020) potentially affecting snow melt processes there. Please include this in your intro to provide a more complete picture of the process and the importance for glaciers mass balance studies as well.

**A. We have added this in our intro, for this see the text in our reply to your second major comment.**
* * *
L 59: Please also mention rain on snow events as situations where TI models drastically fail.

**A. We agree with you that this should also be mentioned. Based on both reviews, we have adapted this sentence as follows:**

*However, the performance of these models decreases significantly with increasing temporal resolution, adding a spatial component to the model, application beyond the period or domain of calibration (e.g. climate change) or during rain-on-snow events (e.g. Hock, 2003).*
* * *
L 76; as you reference both Schlögl et al (2018) paper you should introduce 2018a and 2018b.

**A. We agree, we have adapted this.**
* * *
L79: this is not entirely correct as Schlögl et al. (2018b) did snow ablation and turbulence measurements over three entire ablation periods.

**A. Indeed, the statement about Schlögl et al. (2018b) was not entirely correct. We have recast this statement, in the correct way and provide examples of measurement campaigns that focus on small areas for longer periods:**

*….Other field measurement campaigns estimated the turbulent heat fluxes and the accompanied mechanisms for a single isolated snow patch (e.g. Granger et al., 2006; Harder et al., 2017). Though, all of these experiments focus on relatively brief periods of time with a maximum timespan of a day, during which the conditions for local-scale advection of the turbulent heat fluxes were often ideal. For small areas in complex terrain and longer periods, Schlögl et al. (2018b) related measured snowmelt to turbulence, while Marsh et al. (1997) estimated the role of advection of turbulent heat by comparing estimated sensible heat fluxes with and without advection. Yet, estimates of a multiple-day contribution of the turbulent heat fluxes to the melt of a single snow patch are lacking, but could provide additional insights in the role of these fluxes on longer timescales.*
* * *
L 83: "This advocates the use of spatial field observations, however, most methods for estimates on small spatial scales are relatively expensive or come with low precision and accuracy" here you should reference studies using TLS to measure high-resolution snow ablation rates (Grünewald et al., 2010; Egli et al., 2012; Schlögl et al., 2018b).

**A. Based on both reviews we have adapted this paragraph. They now are as follows:**

*To observe the melt of a snow patch over the course of multiple days, high spatial variability of melt rates complicates the observations. Single point measurements might not represent the region of interest, especially for seasonal snow covers (Sturm and Benson, 2004). This advocates the use of spatial field observations at a high temporal and spatial resolution for patchy snow covers. Ground-based methods fulfilling these requirements are amongst others terrestrial laser scanning (TLS) (Egli et al., 2012; Grünewald et al., 2010; Hojatimalekshah et al., 2021; Schlögl et al., 2018) or georectification of oblique time-lapse photography (Härer et al., 2013). Additionally, aerial platforms, such as aerial laser scanning (ALS), either manned (e.g. Deems et al., 2013; Painter et al., 2016) or unmanned (e.g. Harder et al., 2020; Jacobs et al., 2021) can be used. For both positions, application of Structure-from-Motion (SfM) photogrammetry is possible as a relatively cheap method to monitor snowmelt, of which the usage has already been explored in the 1960s (e.g Brandenberger, 1959; Hamilton, 1965), but which was sidelined due to technical constraints. Recently, due to the technical development increasing the accuracy with lower computational costs, the SfM photogrammetry has been used to successfully study seasonal snow covers with low-*

*cost imagery equipment and software for analysis (e.g. Bühler et al., 2016; Filhol et al., 2019; Girod et al., 2017; Nolan et al., 2015). Therefore, this method offers a promising way to monitor snowmelt with low costs and reasonable accuracy.*
* * *
L110: Although not being an expert on photogrammetry, I think it is not fully correct that you state that you are using SfM – you are actually using stereophotogrammetry as structure from motion implies that the 3D structure was created using camera movement.

**A. For the photogrammetry, we took images from various positions for both edges. We do realize that this can be more clearly formulated and have added the following to the text (in the methods):**

*The height maps are obtained by applying the photogrammetric principle of Structure from Motion (SfM) using MicMac (Rupnik et al., 2017). A total of 1087 pictures (610 for the upwind edge, 477 for the downwind edge)* *were taken from various positions* *for both edges spread out over the 5 days, using a Xiaomi A2 smartphone camera.*
* * *
L114: can you shortly explain how those measurements serve as a basis?

**A. We have added:**

*We use the measurements of Harder et al. (2017) on a single snow patch as a basis* *for designing our numerical experiments, and choose to focus on the sensible heat flux.*
* * *
L129: Please elaborate on how you used the meteorological data in 2.5 km distance from the actual field site. Have you applied any spatial interpolation to data to account for elevation difference or local terrain effects (e.g. wind, radiation)?

**A. For the meteorological variables, we have not applied any spatial interpolation. We are aware that these number do not exactly represent the local circumstances at the observed snow patch. We have elaborated on this:**

*When comparing the measured meteorological conditions with the snowmelt, especially the net radiation (Eq. 1 and 2**), the spatial variability of these conditions*  *should be considered.* *For this comparison, we apply the measured values without any additional computations other than time-averaging, introducing additional uncertainty. For example,*  *the snow patch is located at the bottom of a north-south orientated side valley, which causes shading at sunrise and sunset. The most prominent mountains to the east and west of the snow patch are approximately 150 to 200 m higher at 1 to 1.5 km distance from the snow patch. The meteorological measurements are done in an east-west orientated main valley, such that less shading occurs at sunrise and sunset.*
* * *
L134: Why did you not measure the spatial distribution of snow ablation over the entire snow patch? How did you determine the local wind direction? Also, was the wind fetch always constant through the measurement time period?

**A. Indeed, having a photogrammetry product covering the entire snow patch would be ideal for this study and allow for a more detailed analysis of the snowmelt. However, this would require other equipment than what was available. Still, with the equipment at hand, we try to illustrate that with relative simple and cheap methods, it is possible to come up with relatively decent snowmelt estimates. We have added in the paragraph introducing MicMac, the following for this:**

*The height maps are obtained by applying the photogrammetric principle of Structure from Motion (SfM) using MicMac (Rupnik et al., 2017). A total of 1087 pictures (610 for the upwind edge, 477 for the downwind edge) were taken from various positions for both edges spread out over the 5 days, using a Xiaomi A2 smartphone camera. By using a ground-based camera, we are not able to obtain height maps covering the entire snow patch, which does prohibits a detailed analysis of the snowmelt. Yet, using this method, we illustrate that with a simple and cheap method, it is still possible to come up with relatively decent snowmelt estimates. The method is based on Filhol et al. (2019), who studied the melt of a relatively large snowfield with 3 time-lapse cameras over the course of an ablation season. …*

**Regarding the fetch, we have added the following sentence:**

*From this snow patch (Fig. 1), daily height maps are obtained through photogrammetry over the course of 11 June until 15 June 2019 for the upwind and downwind edge of the snow patch. As the dominating wind direction experienced at the snow patch resembled the measured wind direction at the meteorological tower, which was constant throughout the field campaign (Table 4 in Sect. 4), we assume that the upwind and downwind location of the snow patch are approximately constant.*
* * *
L139: why did you not measure SWE for the entire snowpack? Doing so at different sites with different snow depths would allow a more precise information on SWE of the snowpack at the snow patch. Was the snow pack already isothermal at the start of the measurement campaign?

**A. We are aware that taking these samples only on a single location and only on one day does not reflect the potentially complex spatial (and temporal) dynamics of the snow density and SWE. However, we assume the variations occurring on these spatial and temporal scales to be relatively small compared to other uncertainties introduced to our method for computing contribution estimates of the turbulent heat fluxes to the snowmelt. We have added this consideration by adding the following text after our introduction about the snow pit:**

*The snowpit is dug in snow adjacent to the snow patch, such that the measurements are similar to the snow patch, but do not affect the photogrammetry observations. We are aware that taking these samples only on a single day and location does not reflect the potentially complex temporal and spatial dynamics of the snow density. However, we assume the variations occurring on these scales to be relatively small compared to other*

*uncertainties introduced by our analysis for computing contribution estimates of the turbulent heat fluxes to the snowmelt.*

**Regarding the snow pack, we added the following lines at the lines introducing our snowmelt calculations:**

*Furthermore, we assume that the snow*pack *is continuously*  ripe *throughout research period, given that air temperature was continuously above freezing point and the largest discharge peak had already taken place 1.5 months before the observation period.*
* * *
L159: would be nice to show the resulting snow ablation map of the snow patch retrieved from SfM.

**A. We have added the resulting height maps per day to the appendix. We have changed the original Figure A1 by including the eventually used height changes, in accordance with figure 4, and aligning the edges of the patches. The new figure looks like:**

[Figure]

***Figure A1**. **The resulting height change maps used for Figure 4 plotted over** the real-color orthoimages, which are used for distinguishing snow from bare ground, after removing isolated groups of cells and median filtering. The insets show zoomed areas of the upwind edge for better view.*
* * *
Figure 1: the small figure is not clear to me. Are white areas still snow-covered? Is this an orthofoto? How did you determine the local wind direction?

**A. Indeed the white areas are still snow-covered. Also similar to your comment on L134, we have changed the description on the wind direction. We have added this to the caption:**

*Figure 1. Overview of the Finseelvi research area with the location of the snow patch and daily averaged discharge and SCF data of the catchment. The map of Norway is obtained from https://norgeibilder.no/, the catchment area and the streams are respectively obtained through the University of Oslo and the Norwegian Water Resources and Energy Directorate (http://nedlasting.nve.no/gis/). The zoomed image is made by the Sentinel-2 satellite on 13 June 2018 with the snow patch (white areas are snow covered) and local wind direction indicated as experienced during the field campaign, which resembled the measured direction at the meteorological tower.*
* * *
L 161: what do you mean by assuming a snow albedo between 0.6 and 0.8? changed the value in time? Can you provide a reference for choosing those numbers? The albedo value has an extreme effect on your energy balance calculation and your estimated contribution from turbulent heat fluxes.

***A. We agree that it was not clear how we used these albedos. Moreover, we have included a description on the use of the longwave radiation, and that the shortwave radiation is the cause for the estimate ranges (see comment on L336) We have expressed this more clearly in this revised version:***

*To determine the snowmelt based on the net radiation, the measured incoming radiation is combined with estimates of the outgoing radiation based with the help of common snow characteristics. The outgoing shortwave radiation is calculated by assuming a snow albedo between 0.6 and 0.8. These values are based on Harding (1986), who reports albedos of approximately 0.8 for the same region in May. We use both of these values in the following calculations to account for the uncertainty in the albedo, due to spatial and temporal variations, and other potential uncertainties in the shortwave component. For the longwave radiation component, we assume it to be an appropriate estimate for the larger region.*

**Also we adapted the caption of Table 4, such that it is clear that we used these albedos for computing the ranges in net radiation and radiation-driven melt:**

*Table 4. Average meteorological measurements and calculated variables in between the photogrammetry observations. $T_{2m}$ is the air temperature measured at 2 meter, $u_{10m}$ and $u_{dir}$ are respectively the wind speed and wind direction in degrees from the north measured at 10 meter, $P_r$ is the summed precipitation during the period, SW↓ and LW↓ are respectively the incoming shortwave and longwave radiation, $RH_{2m}$ the measured relative humidity at 2 meter and $P_a$ the air pressure. The net radiation ($R_{net}$), subsequent melt ($M_R$) and specific humidity difference ($q-q_{sn}$) are computed based on a combination of the measured variables. The ranges in $R_{net}$ and $M_R$ are caused by applying two values*

*for the albedo, i.e. 0.6 and 0.8 to account for uncertainties in the shortwave radiation component (see Sect. 2).*
* * *
L174: the relative humidity was measured at the large-distance test site?

**A. Indeed the relative humidity was measured at the large-distance test site. We are aware that this does not reflect the conditions at the snow patch, however we use the computed values based on the relative humidity only as indication. We have elaborated on this at the end of this paragraph:**

*To obtain an indication of the influence of the latent heat on the melt, relative humidity is recalculated to the vapor pressure difference between the air and snow surface ($e−e_{sn}$) and, subsequently, the specific humidity difference ($q−q_{sn}$). To calculate $e − e_{sn}$, we assume the vapor pressure of the snow to be the saturated vapor pressure of air at 0°C, which is 0.613 kPa. We are aware that the used relative humidity (measured at the meteorological tower) probably does not reflect the exact conditions at the snow patch. Therefore, we only use these computed values as an indication of the contribution of the latent heat on the melt.*
* * *
L 187: local scale advection of what? Also, it is not clear how measurements of Harder have been used for the idealised system. What did you use for what exactly?

**A. We meant local-scale advection of turbulent heat. We have added this to the sentence. Also, we have added a sentence on the how we used the measurements of Harder et al. (2017) in our idealised system as an introduction and also refers to the next section where it is explained. We also add the consideration that this makes the comparison with our own observations more difficult (see reply on 4th major comment).**

*We used an idealised system (Fig. 2) to study the turbulent heat fluxes in detail and understand the behaviour observed in the205field. Instead of using our own measurements, the idealised system is based on the measurements on a single snow patch done by Harder et al. (2017), due to the availability of relatively high resolution measurements and similarity to an idealized system in which the contribution of the local-scale advection of turbulent heat to the total melt is relatively large. We are aware that this complicates the comparison between our observations and simulations. The prescribed conditions within the simulations of the idealised system are based on the observations of Harder et al. (2017) and obtained through a dimensional analysis (elaborated on in Sect. 3.2). We assume our idealised system to consist of an on average near-neutral atmosphere above a patchy surface with heterogeneous properties and can be described by the following variables:*
* * *
Figure 2: Please insert the meaning of the parameters in the figure caption. It is not easy to understand what the figure is showing.

**A. We have added the meaning of the parameters in the caption. The caption now reads:**

*Sketches of one snow patch and the adjacent bare ground, i.e. an element, (left) and an exemplary horizontal domain with indicated element and snow patches (right). The parameters represent viscosity v, thermal diffusivity κ, wind speed u, height of the atmospheric surface layer δ, average size of one snow patch element $λ_{elem}$, consisting of the length of the snow patch $λ_{sn}$ and the adjacent bare ground ($λ_{elem}≡λ_{bg}+λ_{sn}$), and the buoyancy of the snow $b_{sn}$ and bare ground $b_{bg}$.*
* * *
L 195: if theta_atm is the temperature of the atmosphere – what is theta then? The surface temperature of snow/bare ground? Please define.

**A. Theta can be any temperature. It is used to recalculate a temperature to a buoyancy. We have elaborated the sentence:**

*in which $g(m\ s^{-2})$ is the gravitational acceleration, $θ_{atm}(K)$ is the temperature of the atmosphere and θ can be any temperature to be recalculated to the buoyancy b in this equation.*
* * *
L 318: how do you define up-wind and downwind edge? is it the first grid cell? How do you deal with grid cells which become snow-free during the observation day? The daily-melt rate will be underestimated if you also consider pixels which become snow-free during a measurement day. Would be interesting to see a snow ablation rate curve depending on fetch distance.

**A. Based on your comments and the other reviewer, we have adapted section 2 stating the post-processing of the grids. Due to these changes, also the way we deal with grid cells is explained. It is now as follows:**

*Both types of grids are available for each of the 5 days and for both locations, being the upwind and downwind edge, such that we have 20 grids in total. For both locations, the following post-processing is done after obtaining the grids:*

1. *For all grids, remove isolated groups of cells which are smaller than 0.05 m²*
2. *For all grids, apply a median filter of 5 x 5 pixels to diminish the influence of noise located within the areas of interest, but maintain the sharp transitions between snow and snow free surfaces.*
3. *Compute the median height of the bare ground cells per day. The cells selected for this computation should be covered by all grids and already be bare ground on the first day.*
4. *Compute correction heights through comparing the daily median heights of the bare ground with the median height of the first day (step 3).*
5. *Remove bare ground cells out of DEM, based on orthoimages of the same day, such that only snow-covered cells remain for each day.*

*6. Apply correction height (step 4) on snow covered DEM (step 5) for each day.*

*7. For snow covered grid cells that are present on each day (step 5), we calculate height differences between the DEMs of the first day and other days (step 6). We remove absolute height differences larger than 50 cm, as larger values are highly unlikely to occur.*

*The resulting height differences over time correspond to 6.7 m2 and 30.7 m2 for respectively the upwind and downwind edge. We chose to solely use grid cells that are continuously covered by snow and have a recorded height change on each day, to reduce the chance of cells being random scatter. Additionally, this method does not include cells with relatively shallow snow depths of which the recorded melt could be affected by the presence of the bare ground below the snow.*

*We are aware that these filters have an effect on the number of analyzed grid cells and could cause an underestimation of the amount of snowmelt, especially on the upwind edge due to the varying locations of snow covered grid cells and the retreating snow line (Figure A1). For the downwind edge, the approximately constant location of the snow covered grid cells combined with the little retreat at this edge, causes this area to be significantly larger. As a disadvantage of the sizes of the covered areas, we decided to treat the edge as "point" and not consider the spatial distribution of the recorded melt.*
* * *
L336 and table 4: Please provide more precise explanation on your estimate ranges. Please also state whether any spatial interpolation is done to the meteorological variables or not.

**A. Regarding the spatial interpolation, see our reply to your comment on L129. For the estimate ranges, we have added a short additional description on the use of the albedo (See comment on L161) in the text, and also adapted the caption of table 4, to indicate where these ranges originate from.**
* * *
L337: is that the length of the sow patch?

**A. Yes indeed. We have added "in length":**

*As the snow patch was approximately 50 meter in length, the turbulent heat fluxes have likely reduced to negligible values….*
* * *
L343: I assume that you are taking the difference of snow melt due to radiation (equation2) and the actual snow melt to estimate the contribution of the turbulent heat flux. Please add more information how you exactly calculate the turbulent heat flux (latent and sensible turbulent heat flux?)

**A. We have added a description on how we computed the contribution of the turbulent heat fluxes in the method section after introducing our calculations of the radiation-driven melt:**

$$M_R = \frac{R_{net} * \Delta t}{\rho_{sn} * L_f},$$

in which $M_R$ (m) is the snowmelt due to radiation expressed as height change, $\Delta t$ (s) the time between the photogrammetry observations, $\rho_{sn}$ (kg m$^{-3}$) the snow density and $L_f$ (J kg$^{-1}$) the constant for latent heat of fusion. *Subsequently, we assume that the total melt is caused by radiation and the turbulent heat fluxes (e.g. Plüss and Mazzoni, 1994), such that the difference between the total observed melt and the computed radiation-driven melt can be attributed to the turbulent heat fluxes.*

**Also, we repeat this statement in Section 4.1:**

*When assuming this radiation to be an appropriate estimate and homogeneously spread over the patch, the estimated contribution of the turbulent fluxes at the upwind edge is 13.0 to 18.2±2.0 cm under the assumption that the residual of the difference between the observed snowmelt and radiation-driven snowmelt is caused by the turbulent heat fluxes (e.g Plüss and Mazzoni, 1994).*
* * *
L353/354: and how does this compare to the contribution at the downwind edge? As mentioned earlier it would be extremely interesting to have a fetch distance related estimate of the contribution of turbulent heat fluxes (sensible and latent). Also, if you provide a number of 60-80% contribution at the upwind edge – what does this exactly mean? Over which area? As known from other studies, the contribution strongly changes with fetch distance. These high numbers of 60-80% might be very misleading looking at the relevance for the catchment scale snow melt. It would be very interesting to see an analysis on the contribution of heat advection to total snow melt for varying snow patch sizes and snow cover fractions. Furthermore, the relative contrition of heat advection to total snow melt strongly depends on the spatial variability of snow depths as snowpacks with a high spatial variability of end of season snow depths are typically characterized by a longer time period of the patchy snow cover stage and therefore a higher importance of the heat advection process. A more detailed discussion would allow a better comparison to the study of Schlögl et al., 2018a. Please relate to results of Schlögl et al. (2018), who tried to put the local scale estimations into the catchment scale context to draw conclusions for its relevance.

**A. We agree that we need to specify what area we are dealing with. We included the areas and the explanation that we treat the upwind and downwind edge as point measurements, but are based on spatial estimates. For this see our reply to your comment on L318.**

**For the 60-80%, we do not dare to specify any exact numbers that would account for the entire catchment, as we simply do not know the exact relation with the fetch or behaviour in the catchment. We do include the reference to Schlögl et al. (2018b) (which we expect you meant to refer to instead of 2018a), of which we use Table 2 to extrapolate our findings to be maximally in the order of 10%. We have included this considerations to the results:**

*Overall, by comparing the overall melt with the melt driven by the radiation and taking the residual as turbulence-driven melt, we estimate the contribution of the turbulent heat fluxes to the snowmelt to be roughly 60 to 80 % for the upwind edge of the snow patch.*

*Extrapolating this to the entire catchment, applying the relations reported by Schlögl et al. (2018b) for two test sites in the Alps, we estimate the contribution of the fluxes to the total melt to be maximally in the order of 10 %, under the assumption that the entire catchment behaves similar.*

**Also in the conclusion we repeat this:**

*In this study, we examined the melt of a 50 meter long snow patch in the Finseelvi catchment, Norway, and investigated the observed melt with highly idealized simulations. The melt estimates, obtained with structure-from-motion photogrammetry, for the upwind and downwind edge of the snow patch are feasible and the estimated errors are in line with previous studies. The combined influence of the sensible and latent heat flux on the snowmelt at the upwind edge is estimated to be between 60 and 80%, while for the entire catchment this contribution would come down to be maximally in the order of 10% based on previous studies. This estimate is based on the difference between the recorded melt and net radiation of the snow patch determined with measurements of a meteorological tower near the catchment….*
* * *
Section 4.1: These estimations include many uncertainties (snow density differences depending on snow height, differences in shortwave radiation between snow patch and actualmeasurement location due to terrain shading, albedo). The high number of turbulent heat fluxes at the surface do not tell us how much of this turbulent heat flux originates from the higher air temperatures at the upwind edge caused by the local advection of sensible heat. Regarding the uncertainty in the net shortwave radiation the authors should consider doing radiation modelling for the area for the respective time period including high-resolution terrain information.

**A. We agree that there are many uncertainties in computing these estimates. We therefore specifically chose a relatively large range in albedo to cover the uncertainties in shortwave radiation, and we include these uncertainties in our subsequently computed melt estimates. We also state this is our reply to your comment on L161.**

**Moreover, we acknowledge the benefits of radiation modelling in the discussion:**

*… Overall, this caused not all objects to be captured from multiple perspectives, again complicating tie point identification. A possible solution to these limitations, could be to add passive control points in the snow to create more tie points for the software to connect. For more precise radiation, and thus melt estimates, this study could have benefited from radiation modelling using high-resolution terrain information (cf e.g. Silantyeva et al., 2020). Future studies could also make use of Lidar scanners, which have recently gone into mass production and hence seen the corresponding drop in cost.*

**Regarding the role of the advected heat we indeed cannot state any exact numbers on the role of the advection. Yet, we do so based on Figure 8, where we relate what we observe in the simulation to the observations. For this see, our reply to your first major comment.**
* * *
Figure 5: the authors are quite lazy with terms – please clearly state sensible turbulent heat flux as it has to be clear that this is the turbulent exchange of sensible heat.

**A. We have adapted the figure and caption following your comment:**

[Figure]

*Figure 5. Time-averaged surface sensible turbulent heat fluxes*. *Sensible turbulent heat fluxes in the P15m averaged from 2000 seconds until 2700seconds. Negative values indicate a downward flux, whereas positive values resemble an upward flux.*
* * *
L366: if I understand it correctly you describe here the turbulent sensible heat flux per grid cell and not the heat advection. These values are extremely high. You compare these with estimates of Harder et al (2017) (L273) but they are providing estimates on the advected sensible and latent heat between two points and not the advected turbulent heat flux (paper Harder et al., 2017; equation 2 and 3).

**A. Indeed this comparison was incorrect, we have adapted this. See our reply to your first major comment.**
* * *
L405: I do not fully understand the sentence here. For the local advection of sensible heat not the difference between snow surface temperature and air temperature is important but the horizontal air temperature difference (and mean horizontal wind speed).

**A. We have made some major changes to this section and also removed this sentence. See our reply to your first major comment for the changes.**
* * *
L406: the reduction in the turbulent sensible heat flux of 20% in downwind distance is most probably the result of a decreasing contribution of the advection of sensible heat with increasing fetch distance over snow.

**A. We agree with you. Now we compare these numbers to our local advection estimates (Figure 7), which is the correct comparison. See our reply to your first major comment**
* * *
L409: yes, exactly, the difference arises as you compare the simulated turbulent sensible heat flux at the surface (by your model) to an estimation of the advected sensible heat calculated by Harder et al. (2017).

**A. Indeed this comparison was incorrect, we have adapted this. See our reply to your first major comment.**

---

## Author Response (AR2)

**Dear Referee,**

**We would like to thank you for taking the time to review our paper and for all your constructive suggestions, which definitely helped to improve the quality of the manuscript. We reply to your comments below. Our response to the comments appears in bold and revised text as** *italic***.**

Minor comments:

- 110-115: It is correct that Monin-Obukhov assumptions are violated over patchy snow covers which means that models applying Monin-Obukhov will not be able to capture full dynamics leading to uncertainties in the turbulent flux estimates. However, the way this is stated in the manuscript leaves most of the readers with a big question mark. I ask the authors to give a clear explanation why Monin Obukhov assumptions are not valid over heterogeneous surfaces such as patchy snow covers – either in the introduction or in the discussion part.

    **We agree that we should indicate for the reader what these assumptions are. We have added a sentence in this paragraph on the Monin Obukhov assumptions:**
    *As a consequence of the high resolutions, these simulations do not need any turbulence parametrizations based on stability corrections and the Monin-Obukhov assumptions, in contrast to numerical atmospheric boundary layer models (Liston, 1995; Marsh et al., 1999) or Large-Eddy simulations (Mott et al., 2015; Sauter and Galos, 2016). These methods assume horizontal homogeneity and constant turbulent fluxes throughout the surface layer, which is violated for a patchy snow cover and, as such, introduce a large uncertainty (e.g. Mott et al., 2018; Schlögl et al., 2018a). Bonekamp et al. (2020) successfully showed the potential*
* * *
- L469fff: The statement that ARPS is not able to resolve the leading edge effect at all is not correct and if the authors want to state that they need to be more precise. The authors Schlögl et al clearly state that the leading edge effect is not FULLY resolved mainly due to two reasons (1) the resolution is still too coarse to resolve the very thin internal stable boundary layer and (2) Monin Obukhov assumptions are violated.
    Please revise this paragraph.

    **We agree that this statement was not precise enough. We have adapted these lines following your suggestions:**
    *When comparing the average sensible heat fluxes for all the snow patches in the domain, clear differences arise between all simulations (Figure 6). The highest sensible heat fluxes towards the snow (i.e. most negative) are found in the simulation without buoyancy effects. This also causes the total sensible heat flux for this simulation to be significantly lower than the other simulations. Furthermore, increasing snow patch size reduces the heat fluxes into the snow patches. The heat fluxes of the simulation with doubled snow patch size (P30m) decrease with approximately 15% relative to the P15m simulation. For the simulation with quadrupled snow patch size (P60m), the heat fluxes reduce with approximately 25%. This is in contrast to the results of Schlögl et al. (2018a), who reports a minor influence of snow patch size on the amount of melt. Our findings are more in line with the results of Marsh et al. (1999), who based there work on a 2D Boundary Layer Model with a regular tiled surface pattern. Potentially, the differences with Schlögl et al. (2018a) are caused by the disability of ARPS to fully resolve the leading edge effect, due to a too coarse resolution to resolve the thin internal stable boundary*

*layer formed over snow patches and the violation of the Monin-Obukhov assumptions. Both of these limitations do not apply for DNS.*
* * *
- The sentence "Especially, considering that DNS does not violate the assumptions for the Monin-Obukhov bulk formulations and is able to resolve the leading-edge effect, in contrast to modelling studies with coarser spatial resolutions, which could lead to major errors (Schlögl et al., 2017). As such, the simulations are expected to provide a more realistic behaviour of the turbulent heat fluxes." I do not fully agree as you miss the driving factors resulting in the leading edge effect which is the diurnal cycle of radiation which is heating the surface leading to instable conditions over bare ground affecting the leading edge effect which changes depending on the strength of the heating and the associated change in stability. I think the term "more realistic" is not correct as you need exact knowledge on the boundary conditions getting the realistic turbulent heat flux estimates. The unrealistically high values of turbulent heat fluxes indicate that model results are not really realistic, probably due to too extreme boundary conditions/model settings. Furthermore, in your simulations, heat advection appears to be dominant over very long distances, which is not in agreement with measurements so far. I would highlight the strength of the DNS model to capture the leading edge effect in a relative sense, i.e. the percentage of the increase in the turbulent heat flux depending on fetch distance. We also know that the leading edge effect strongly depends on wind speed – this is not addressed in the current form of the manuscript. Please revise this paragraph.

**In this final part of the paragraph, we agree with your suggestions to emphasise the general strength of DNS to capture the leading edge effect. To highlight this, we have adapted the paragraph following your comments:**

*Moreover, we identify some inaccuracies during the nondimensional scaling of the wind speed and temperature difference between the snow and atmosphere. As Harder et al. (2017) reported a wind speed of 6.4 m s$^{-1}$, this value is also considered in our dimensional analysis and related to 0.11 m s$^{-1}$, which was the average wind speed over the whole channel in the case of Moser et al. (1999). However, the reported wind speed was measured at 1.8 meter above the ground, thus implying that the average wind speed for the whole air column under consideration would be higher. Consequently this affects the leading edge effect, due to increased wind shear and, thus, also the fluxes towards the surface. Also, the temperature difference between the atmosphere and snow has been possibly overestimated, causing an increased sensible heat flux. The graphs presented by Harder et al. (2017) show the temperatures of bare ground and atmosphere to be constant near the surface, being 6.4 ∘C. However, for the dimensional analysis, the atmospheric temperature mentioned by Harder et al. (2017), i.e. 7.9 ∘C, has been used. Overall, these differences in assumptions between the simulations and the field observations make a one-to-one comparison difficult. Yet, the general behaviour found in the simulations is similar to previous literature, i.e. temperature profiles and melting patterns (e.g. Harder et al., 2017; Mott et al., 2016), and shows the potential of DNS as a modelling tool to understand the melting of a patchy snow cover. Especially, considering that DNS does not violate the assumptions for the Monin-Obukhov bulk formulations and is able to resolve the leading edge effect, in contrast to modelling studies with coarser spatial resolutions, which could lead to major errors (Schlögl et al., 2017). As such, this type of simulations is expected to provide a more realistic behaviour of the leading edge*

*effect on the turbulent heat fluxes, especially when combining with case-specific boundary conditions.*
* * *
• The authors still switch between the terms "local advection of sensible heat" and "local advection of turbulent heat". Please stick to the original definition (see first review for more details).

**Our apologies, we misinterpreted your comment in the first review. When we discussed the *local advection of turbulent heat*, we meant the advection of both sensible and latent heat. However, this usage indeed is incorrect. We have adopted this and refer now to *local advection of sensible and latent heat.* Moreover, we have added the words *vertical* and *horizontal* (or comparable) when discussing both the vertical turbulent heat fluxes and local advection, in order to create a clear distinction between the two. For example:**

*…When a wind blows over this horizontally heterogeneous surface, downwind of the transitions internal boundary layers form, due to changes in the surface conditions (e.g. Garratt, 1990). The heterogeneity of these internal boundary layers induces a system in which the turbulent heat fluxes can highly vary spatially, partly due to the advection of sensible and latent heat (Essery et al., 2006; Mott et al., 2013; Harder et al., 2017). These systems are often described as….*

*…Therefore in this study, we aim to assess the role of horizontal advection of the sensible and latent heat on snowmelt for a snow patch in a real world case and idealized environment. In the real world case, we will identify the role of the locally advected sensible and latent heat on a melting snow patch in the Finseelvi catchment through studying the vertical turbulent heat fluxes with SfM photogrammetry observations over the course of multiple days. The resulting snowmelt is compared to local meteorological measurements to put the snowmelt in perspective and extract the role of the turbulent fluxes on this melt. Subsequently, we try to uncover the behaviour of the vertical sensible heat fluxes on snowmelt, including the local-scale advection of sensible heat, in an idealized environment with DNS, allowing to extract detailed information on wind blowing over a small flat domain with a patchy snow cover….*

*Instead of using our own measurements, the idealised system is based on the measurements on a single snow patch done by Harder et al. (2017), due to the availability of relatively high resolution measurements and similarity to an idealized system in which the contribution of the local-scale advection of the sensible and latent heat to the total melt is relatively large….*

*Our simulated vertical sensible heat fluxes into the snow are approximately 500 W m$^{-2}$ and at the upwind edge and 200-300 W m$^{-2}$ at the downwind edge. In comparison to our field observations, at both edges these sensible heat fluxes are relatively high. At the upwind edge, the simulated sensible heat fluxes are approximately 5 times larger than the derived contribution of the combined turbulent heat fluxes to the measured snowmelt. We reckon that the simulated values are large, though it should be noted that the simulations are based on highly ideal conditions for turbulence-driven melt and local-scale advection of sensible and latent heat, whereas the conditions during the measurements were not ideal (e.g. nighttime*

*melt is included). At the downwind edge, the measurements suggest an approximately negligible contribution of the vertical turbulent heat fluxes to the snowmelt, whereas the simulations show at comparable snow patches a significant contribution of the vertical sensible heat flux (~ 200-300 W m$^{-2}$).*

*We used an idealised system (Fig. 2) to study the turbulent heat fluxes in detail and understand the behaviour observed in the field. As this is one of the first studies using DNS to investigate the role of the vertical turbulent heat fluxes and local heat advection in these systems, we focus on the sensible heat flux, even though MicroHH allows to include the latent heat flux (e.g. Bonekamp et al., 2020). Instead of using our own measurements…*

*Relating the meteorological circumstances (Table 4) to the measured snowmelt, allows to estimate the contribution of the vertical turbulent heat fluxes to the total amount of snowmelt.*

*As the snow patch was approximately 50 meter in length, the turbulent heat fluxes into the snow have likely reduced to negligible values at the downwind edge (e.g. when extrapolating the measurements of Harder et al., 2017)*

*Compared to Harder et al. (2017), our estimates of the horizontally advected sensible heat are relatively high. The measurements done by Harder et al. (2017) show values slightly above 400 W m$^{-2}$ for the first 3.6 m (Figure 7). In our simulations the horizontal advection of sensible heat decreases in the first 3.6 m from 577 W m$^{-2}$ to approximately 400 W m$^{-2}$. For the following 4.8 m, Harder et al. (2017) reported an average reduction in horizontally advected sensible heat to approximately 20% of values found for the first 3.6 m. In the comparable simulation with a similar dominant snow patch pattern, this reduction is not found. Yet, further downwind the advected sensible heat does reduce to values in the same order of magnitude. It should be noted that, due to setting the integration height at 2 m in Eq. 17, not all changes in the vertical temperature profile over distance from the leading edge are included, such that the horizontally advected sensible heat is underestimated. When considering a 4 m integration height, the horizontally advected sensible heat is approximately equal to the vertical sensible turbulent heat fluxes at the snow surface, especially for the first half of the patch, implying also a power of -0.35 for the decay of the horizontally advected sensible heat. This also illustrates the major contribution of the horizontally advected sensible heat to the sensible heat flux into the snow...*
* * *
- I think that the manuscript would benefit from language editing.

- Some sentences are not clear to me, e.g.: "Overall, these mechanisms are the main factors that prevent a direct match between the simulations and observation." I do not fully agree here as not considering these factors increases the uncertainty …

**We have thoroughly reread the manuscript and have adopted multiple sentences. We agree that this has made the manuscript easier to read and more clear. For example:**

*For the stable internal boundary layers, i.e. over snow patches, the air close to the snow surface can decouple from the warmer air above, either due to large temperature differences between*

*both or through cold-air pooling, eventually limiting the exchange of sensible and latent heat from the atmosphere towards the snow (Fujita et al., 2010; Mott et al., 2016).*

*As a consequence, more field observations should be performed to study its importance in various environmental settings. Additionally, these should be combined with other modelling approaches that can serve as a tool to improve our understanding of the process on small and larger scales.* (splitted sentence)

*Overall, these mechanisms greatly increase the uncertainty and make us decide not to directly compare between the simulations and observation.*

*The simulations reveal that the reducing sensible heat fluxes over distance from the leading edge are caused by the reducing temperature gradients, pointing out the major role of the horizontally advected sensible heat, which we expect to behave similarly in our field observations.*

*Moreover, the common formation of snow patches in topographical depressions causes atmospheric decoupling and reduced vertical turbulent heat fluxes at low and moderate wind speeds, especially downstream of the upwind edge (Mott et al., 2016). In the Finseelvi catchment snow patches have formed to some extent in these depressions, while this does not hold for Harder et al. (2017).* (splitted sentence)

*We expect that the important role of the horizontally advected sensible heat and the identical behaviour of the vertical sensible heat flux between patches, both found in the simulations, is also applicable to our field observations, given the probable larger-scale approximate equilibrium of the atmosphere. Though anomalies in this behaviour can be found due to varying micrometeorological conditions.* (splitted sentence)

*As multiple studies suggest the role of stability on snowmelt (e.g. Dadic et al., 2013; Essery et al., 2006), stable regions (i.e. snow patches) could have a much larger impact on the amount of wind-driven snowmelt, especially when reducing the turbulence towards the edge of collapsing.*